# String correlators on AdS$_3$: analytic structure and dual CFT

**Andrea Dei,**[a] **Lorenz Eberhardt**[b]

[a]*Jefferson Physical Laboratory, Harvard University,*
   *Cambridge, MA 02138 USA*

[b]*School of Natural Sciences, Institute for Advanced Study,*
   *Princeton, NJ 08540, USA*

 *E-mail:* adei@fas.harvard.edu, elorenz@ias.edu

ABSTRACT: We continue our study of string correlators on Euclidean AdS$_3$ with pure NS-NS flux. The worldsheet and spacetime correlators have a rich analytic structure, which we analyse completely for genus 0 four-point functions. We show that correlators exhibit a simple behaviour near their singularities. The spacetime correlators are meromorphic functions in the SL$(2,\mathbb{R})$-spins, whose pole structure is shown to agree with the prediction of a recent proposal for the dual CFT$_2$. Moreover, we also compute the residues of the spacetime correlators for some of the poles exactly and find again a perfect match with the proposal for the dual CFT$_2$, thereby checking the duality for some non-trivial four-point functions exactly. Our computations simplify drastically in the tensionless limit of AdS$_3 \times$ S$^3 \times \mathbb{T}^4$ where the behaviour near the poles gives in fact the exact answer. This paper is the third in a series with several installments.

# 1  Introduction and summary

String theory on AdS$_3$ with pure NS-NS flux is a fruitful playground to test ideas of worldsheet CFT and the AdS/CFT correspondence. It has the right amount of complexity to be physically interesting, but still computationally accessible. Since the background is supported by pure NS-NS flux, the theory is in principle straightforward to describe using perturbative string theory. For bosonic strings, the worldsheet theory involves the $SL(2, \mathbb{R})_k$ WZW model (or rather the model based on the universal cover of $SL(2, \mathbb{R})$), and superstrings can be described similarly using the RNS formalism.

The $SL(2, \mathbb{R})_k$ WZW model and its application to the description of AdS$_3$ background was studied extensively in the literature – see e.g. [1–27] for a partial list. The model is far more intricate than its compact $SU(2)_k$ counterpart, due to the appearance of so-called spectrally flowed representations. These are representations of the affine $\mathfrak{sl}(2, \mathbb{R})_k$ algebra with unbounded worldsheet energy from below. Consequently, they are much harder to study using traditional CFT techniques. There is in our view no entirely satisfactory solution to the problem of computing arbitrary correlation functions of such spectrally-flowed affine primary fields (at least in principle).[1] The problem becomes conceptually simpler for a Euclidean AdS$_3$ target space. In this case the spacetime theory is a Euclidean CFT and hence it is very natural to describe correlation functions in the usual spacetime CFT basis – namely position space. This is the so-called $x$-basis and it is the object of study of this paper.

In [7, 11] Teschner made a complete proposal for the correlation functions of the unflowed sector of the Euclidean theory, which is often called the $H_3^+$ model. We extended his result in [25, 26] to a complete proposal for the genus 0 three- and four-point functions of vertex operators involving arbitrary amounts of spectral flow. We demonstrated that knowledge of unflowed correlation functions fully determines spectrally flowed correlators via an integral transform that we review in Section 2.

---

[1]Various special cases are known. In particular the problem simplifies significantly in the so-called $m$-basis. However, somewhat confusingly this basis leads only to a subsector of the set of correlators relevant for string theory on AdS$_3$. We explained this important distinction in more detail in [25].

It is very rare to have closed form expressions for correlation functions in any curved background of string theory. It thus begs for a detailed exploration of the physical consequences. In general, it is not well-understood how to quantize the non-unitary sigma-models arising as string worldsheet theories of time-dependent backgrounds. For AdS$_3$ however, we have an extended symmetry on the worldsheet that helps guide our way, but the correlation functions have peculiar properties that we take as a signature of the time dependency of the background.

Another major motivation for the further study of string theory on AdS$_3$ is that it offers an essentially unique arena where on can explore the AdS/CFT correspondence directly from the worldsheet. In fact, the formula in [25] for the three-point functions of three spectrally flowed affine primary vertex operators directly lead to a proposal of the spacetime CFT dual to strings on AdS$_3$ with pure NS-NS flux in [27]. This proposal built on [23] and a similar proposal was made in a more restrictive setting independently in [28–30]. Since the proposal [27] will play an important role in the present paper, we will briefly explain it here. A more detailed review can be found in Section 5. Consider bosonic string theory on AdS$_3 \times X$, where $X$ is a unitary compact CFT so that the worldsheet central charge is critical.[2] Then the dual CFT is conjectured to be

$$\mathrm{Sym}^N(\mathbb{R}_Q \times X) \tag{1.1}$$

deformed by a particular non-normalisable operator $\Phi$. Here $\mathbb{R}_Q$ is a linear dilaton theory with slope $Q = \frac{k-3}{\sqrt{k-2}}$. $N$ is the number of fundamental strings in the background. Even though this is not relevant for our discussion, we should mention that the definition of this CFT at finite $N$ is not understood and the non-perturbative status of the duality is unclear. Finally, the marginal operator is given by $\Phi = \mathrm{e}^{-\sqrt{k-2}\,\phi}\sigma_2$, where $\phi$ is the linear dilaton field and $\sigma_2$ is the twist field of the twist-2 sector of the symmetric orbifold. In the correspondence, the $\mathrm{SL}(2,\mathbb{R})$-spin on the worldsheet maps to the momentum along the linear dilaton direction and spectral flow on the worldsheet maps to the twist in the symmetric orbifold. We should also emphasise that this CFT has a continuous spectrum (because of the non-compact linear dilaton direction). This is characteristic of pure NS-NS AdS$_3$ backgrounds and is reflected on the bulk side through the existence of the long string sector. From the dual CFT point of view, short string states arise in this CFT as bound states. In particular, they can be detected as poles in correlation functions of long strings [31].

In the present paper we build on our previous results [26] for the four-point functions of the worldsheet theory of strings on Euclidean AdS$_3$ and explore their physical consequences. There are a few major goals that we achieve in this paper. After reviewing known results and setting up our conventions in Section 2, we start our analysis in earnest in Section 3, where we fully analyse the singularities of worldsheet four-point functions. Let us explain the main result of the section succinctly without introducing too much of the later notation.

---

[2]Of course, the bosonic string only makes sense on genus 0 surfaces and ultimately the duality is not well-defined at higher genus. Since we discuss only genus 0 correlation functions in this paper, we can avoid the extra technical complication of discussing the superstring.

Consider the correlation function

$$\left\langle V^{w_1}_{j_1,h_1,\bar{h}_1}(x_1;z_1) V^{w_2}_{j_2,h_2,\bar{h}_2}(x_2;z_2) V^{w_3}_{j_3,h_3,\bar{h}_3}(x_3;z_3) V^{w_4}_{j_4,h_4,\bar{h}_4}(x_4;z_4) \right\rangle \tag{1.2}$$

of four spectrally flowed affine primary vertex operators. Each vertex operator is inserted at a specific worldsheet position $z_i$ and a specific target space position $x_i$. Moreover, it has an $\mathrm{SL}(2,\mathbb{R})$-spin $j_i$ and a spectral flow $w_i \in \mathbb{Z}_{\geq 0}$. Finally, it carries a spacetime conformal weight $(h_i,\bar{h}_i)$ that plays the role of the magnetic quantum number of the representation. It is then very interesting to study singularities of this correlator in the space of $x_i$'s and $z_i$'s. As in any CFT, the correlator develops a singularity whenever two vertex operators collide and in the present case this happens both in the $x_i$'s and the $z_i$'s. More interestingly though the correlator has a large number of additional singularities in the coordinate space of $x_i$'s and $z_i$'s. In a string theory language, they are associated to the existence of what might be called worldsheet instantons [17]. In a nutshell, for specific choices of the cross ratios in $x$- and $z$-space, the worldsheet can expand to the boundary of $\mathrm{AdS}_3$ at finite cost of energy. This opens up a non-compact radial direction in the path integral of the worldsheet theory (that integrates in particular over the radial direction of the worldsheet) which leads to the divergence.

We show that one can describe all possible worldsheet singularities as follows. The worldsheet correlation function has a potential singularity in the combined $x$- and $z$-space if there is a ramified holomorphic covering map mapping the worldsheet insertions $z_i$ to the target space insertions $x_i$. The map is ramified over $x_i$ with ramification indices $n_i \in \{w_i - 1, w_i, w_i + 1\}$. The simplest example of this condition is the case where all four ramification indices are equal to 1, which implies that the map is a Möbius transformation and thus there is an additional singularity when the two crossratios $x$ and $z$ agree. The correlation function then behaves as $f(x) \propto |z - x|^{2\eta} + \dots$ as $z \to x$, where the dots stand for more regular terms and $\eta$ is a critical exponent (see eq. (3.17a) for its specific form). There is also always a regular term and hence we need $\eta < 0$ for the singularity to actually show up. We determine the critical exponents $\eta$ associated to each singularity in Section 3. Whether $\eta < 0$ is possible depends on the detailed choices of the representations and spins etc. For example, in the simplest case where the ramification indices coincide with the spectral flow, $n_i = w_i$, the critical exponent takes the simple form $\eta = \sum_i j_i - k$. Thus the worldsheet correlator is singular if $\sum_i j_i < k$.

The presence of such singularities is peculiar from an axiomatic CFT standpoint, where correlators usually only have singularities when vertex operators collide. This signals a mild breakdown of locality in the model, since the condition for the location of these singularities depends simultaneously on all four vertex operators and is thus non-local. Although we will not pursue this direction here, we also want to mention that the presence of these singularities poses a challenge to the very definition of string perturbation theory. In flat space string theory, the definition of string amplitudes is inherently tied to the Deligne-Mumford compactification of (super)moduli space, see e.g. [32] for a modern discussion. The presence of extra singularities in the integrand means that the integration contour for the string correlator should be modified rather drastically. These issues are however not

too important here because all the integrals that we encounter can be defined by analytic continuation in the external parameters.

After having described the structure of worldsheet singularities in detail, we then move on and describe singularities in the string theory spacetime correlators, i.e. the correlators of the dual CFT. Of course, the dual CFT is a unitary CFT and correlators just have the usual singularities in position space. However, correlation functions are also functions of the momenta of the linear dilaton direction appearing in eq. (1.1) (that corresponds to the $\mathrm{SL}(2, \mathbb{R})$-spin on the worldsheet). Since this direction is non-compact, these momenta are continuous and there can be simple poles in the momentum dependence. These singularities are sometimes called bulk poles [31] and are what we mean by the singularities of the spacetime correlator. Some of these spacetime singularities are associated to worldsheet singularities, but the relation is not one-to-one. We work out all the locations of such bulk poles in Section 4. While their appearance from the worldsheet is intricate, the final result is rather simple and given in eq. (4.1), which is seen to be the correct answer that is predicted by the proposal (1.1). Thus our analysis provides a further check of the proposed duality.

Our tools are much sharper than what we described so far and are good enough to evaluate also the residues of the spacetime correlators on the bulk pole singularities. While this is in principle possible for all bulk pole singularities, we demonstrate it in the arguably simplest case that corresponds to the condition $\sum_i j_i = 5 - k$. Via the proposal (1.1) for the dual CFT, this condition is mapped to the momentum conservation of the linear dilaton theory. Hence this residue is computed in the dual CFT simply by the symmetric orbifold correlator, whereas residues of more complicated singularities corresponding to bulk poles could be computed by a variant of the Coulomb gas formalism that is described in [27]. We demonstrate explicitly that our formalism indeed predicts exactly the correct symmetric orbifold correlator for the residue of the simple bulk pole $\sum_i j_i = 5 - k$. This yields strong further evidence for the proposal (1.1) whose validity was so far not been tested for correlation functions that involve a non-trivial moduli space integral on the string side. Let us emphasize that our computation matches a very non-trivial four-point function on the string and CFT side to leading order in $1/N$, but exactly in $\alpha'$. As far as we are aware, such a computation is far beyond the capabilities of any formalism in any other AdS background.

Finally, we explain the application of our analysis to the tensionless limit of superstrings on $\mathrm{AdS}_3 \times \mathrm{S}^3 \times \mathbb{T}^4$, which was conjectured to be dual to the symmetric orbifold of $\mathbb{T}^4$ (without any further deformation) [33, 34]. This proposal was structurally already extensively tested [24, 35–43], but a full quantitative match of the correlation functions is still missing. Our tools are in principle strong enough to provide such a match for genus 0 correlators. Since the worldsheet theory is formulated in terms of the hybrid formalism [44], we sidestep most of the technical difficulties at the price of having a less rigorous discussion than in the rest of the paper. Most of the relevant features already appeared in the literature before, but we can make more quantitative statements. Briefly, instead of the singularities on the worldsheet of the type as discussed above, one now has $\delta$-function-like singularities [24, 37]

and hence the moduli space integral *localizes*. One can then indeed obtain correlation functions of a symmetric orbifold of a free theory in this case.

This paper is organized as follows. We start in Section 2 to review our formulae for the worldsheet three- and four-point functions from [25, 26]. We then discuss all worldsheet singularities systematically in Section 3 and all the poles in the spacetime correlator in Section 4. We finally reap the reward of our analysis in Section 5, where we discuss results and the match with the spacetime CFT. The paper is written such that one can jump directly from here to Section 5, should one not be interested in the derivations of the results. We finally conclude in Section 6 with the discussion and some open problems. Various appendices complement the main text. In Appendices A and B we review and partially extend various properties of unflowed correlators and their relation with Liouville theory. Some of these results — as the derivation of the location of poles for correlators with no spectral flow — are new and to the best of our knowledge did not appear before in the literature. In Appendix C we derive various useful identities, while in Appendix D we provide some concrete examples for computations carried out in the main text. Finally, we review symmetric orbifold correlators in Appendix E.

## 2 A short review of three- and four-point functions

In [25, 26] we proposed a closed-form formula for three- and four-point functions of discrete and continuous representations with an arbitrary amount of spectral flow for the vertex operators of the $\text{SL}(2, \mathbb{R})$ WZW model. Let us briefly review our proposal. We explained the relevant vertex operators $V_{j,h,\bar{h}}^{w}(x; z)$ at an intuitive level in the Introduction. The technical definition is given in [25], but it will not be needed in the present paper.

### 2.1 Three-point function of continuous representations

The three-point function

$$\left\langle V_{j_1,h_1,\bar{h}_1}^{w_1}(0; 0) V_{j_2,h_2,\bar{h}_2}^{w_2}(1; 1) V_{j_3,h_3,\bar{h}_3}^{w_3}(\infty; \infty) \right\rangle \tag{2.1}$$

vanishes unless [17, 24]

$$\sum_{i \neq j} w_i \geq w_j - 1 \qquad \text{for all } j \in \{1, 2, 3\} . \tag{2.2}$$

When non-vanishing, the correlator of three vertex operators corresponding to continuous representations reads

$$\left\langle V_{j_1,h_1,\bar{h}_1}^{w_1}(0; 0) V_{j_2,h_2,\bar{h}_2}^{w_2}(1; 1) V_{j_3,h_3,\bar{h}_3}^{w_3}(\infty; \infty) \right\rangle = D(j_1, j_2, j_3) \times$$
$$\times \int \prod_{i=1}^{3} \frac{\mathrm{d}^2 y_i}{\pi} y_i^{\frac{kw_i}{2}+j_i-h_i-1} \bar{y}_i^{\frac{kw_i}{2}+j_i-\bar{h}_i-1} |X_\emptyset|^{2 \sum_l j_l - 2k} \prod_{i<\ell}^{3} |X_{i\ell}|^{2 \sum_l j_l - 4j_i - 4j_\ell} , \tag{2.3a}$$

for $\sum w_i \in 2\mathbb{Z}$, while we have

$$\left\langle V^{w_1}_{j_1,h_1,\bar{h}_1}(0;0)\, V^{w_2}_{j_2,h_2,\bar{h}_2}(1;1)\, V^{w_3}_{j_3,h_3,\bar{h}_3}(\infty;\infty)\right\rangle = \mathcal{N}(j_1)D(\tfrac{k}{2}-j_1,j_2,j_3)\times$$

$$\times \int \prod_{i=1}^{3} \frac{\mathrm{d}^2 y_i}{\pi}\, y_i^{\frac{kw_i}{2}+j_i-h_i-1}\, \bar{y}_i^{\frac{kw_i}{2}+j_i-\bar{h}_i-1} |X_{123}|^{k-2\sum_l j_l} \prod_{i=1}^{3} |X_i|^{2\sum_l j_l - 4j_i - k} \quad (2.3\text{b})$$

for $\sum w_i \in 2\mathbb{Z}+1$. Let us review the definition of the various elements entering eq. (2.3). We denoted by $D(j_1,j_2,j_3)$ the three-point function with $w_1 = w_2 = w_3 = 0$, which is known in closed form [7, 11].[3] The normalisation constant $\mathcal{N}(j)$ reads

$$\mathcal{N}(j) = \frac{\nu^{\frac{k}{2}-2j}}{\gamma\left(\frac{2j-1}{k-2}\right)}\,, \quad (2.4)$$

where

$$\gamma(t) \equiv \frac{\Gamma(t)}{\Gamma(1-t)}\,. \quad (2.5)$$

The notation $\bar{t}$ denotes the corresponding right-moving quantity, not the complex conjugate, e.g. $\gamma(h) = \frac{\Gamma(h)}{\Gamma(1-h)}$, but $\gamma(j) = \frac{\Gamma(j)}{\Gamma(1-j)}$, since the left- and right-moving spacetime conformal weight can differ, but the $\mathrm{SL}(2,\mathbb{R})$-spin $j$ is shared between left and right-movers. The quantities $X_I$ for any subset $I \subseteq \{1,2,3\}$ are defined as

$$X_I(y_1,y_2,y_3) = \sum_{i\in I:\ \varepsilon_i=\pm 1} P_{\boldsymbol{w}+\sum_{i\in I}\varepsilon_i e_i} \prod_{i\in I} y_i^{\frac{1-\varepsilon_i}{2}}\,. \quad (2.6)$$

where $\boldsymbol{w} = (w_1,w_2,w_3)$ and

$$e_1 = (1,0,0)\,, \quad e_2 = (0,1,0)\,, \quad e_3 = (0,0,1)\,. \quad (2.7)$$

$P_{\boldsymbol{w}}$ is a function of $w_1, w_2, w_3$. It vanishes whenever

$$\sum_j w_j < 2 \max_{i=1,2,3} w_i \quad \text{or} \quad \sum_i w_i \in 2\mathbb{Z}+1 \quad (2.8)$$

and otherwise

$$P_{\boldsymbol{w}} = S_{\boldsymbol{w}} \frac{G\left(\frac{-w_1+w_2+w_3}{2}+1\right)G\left(\frac{w_1-w_2+w_3}{2}+1\right)G\left(\frac{w_1+w_2-w_3}{2}+1\right)G\left(\frac{w_1+w_2+w_3}{2}+1\right)}{G(w_1+1)G(w_2+1)G(w_3+1)}\,. \quad (2.9)$$

$G(n)$ denotes the Barnes G function defined for positive integers as $G(n) = \prod_{m=0}^{n-2} m!$. $S_{\boldsymbol{w}}$ is a phase and reads [27]

$$S_{\boldsymbol{w}} = (-1)^{\frac{1}{2}x(x+1)}\,, \qquad x = \frac{1}{2}\sum_{i=1}^{3}(-1)^{w_i w_{i+1}} w_i\,. \quad (2.10)$$

---

[3] See [25] for our conventions.

## 2.2 Four-point function of continuous representations

In [26] we extended the proposal (2.3) to the case of four insertion points. The four-point function

$$\left\langle V^{w_1}_{j_1,h_1,\bar{h}_1}(0;0) V^{w_2}_{j_2,h_2,\bar{h}_2}(1;1) V^{w_3}_{j_3,h_3,\bar{h}_3}(\infty;\infty) V^{w_4}_{j_4,h_4,\bar{h}_4}(x;z) \right\rangle \tag{2.11}$$

vanishes unless [17, 24]

$$\sum_{i=1}^{4} w_i \geq 2 \max_{i=1,\dots 4}(w_i) - 2 . \tag{2.12}$$

When non-vanishing, it reads

$$\left\langle V^{w_1}_{j_1,h_1,\bar{h}_1}(0;0) V^{w_2}_{j_2,h_2,\bar{h}_2}(1;1) V^{w_3}_{j_3,h_3,\bar{h}_3}(\infty;\infty) V^{w_4}_{j_4,h_4,\bar{h}_4}(x;z) \right\rangle$$
$$= \int \prod_{i=1}^{4} \frac{\mathrm{d}^2 y_i}{\pi} \, y_i^{\frac{kw_i}{2}+j_i-h_i-1} \bar{y}_i^{\frac{kw_i}{2}+j_i-\bar{h}_i-1} |X_\emptyset|^{2(j_1+j_2+j_3+j_4-k)}$$
$$\times |X_{12}|^{2(-j_1-j_2+j_3-j_4)} |X_{13}|^{2(-j_1+j_2-j_3+j_4)} |X_{23}|^{2(j_1-j_2-j_3+j_4)} |X_{34}|^{-4j_4}$$
$$\times \left\langle V^0_{j_1}(0;0) V^0_{j_2}(1;1) V^0_{j_3}(\infty;\infty) V^0_{j_4}\left(\frac{X_{23}X_{14}}{X_{12}X_{34}};z\right) \right\rangle \tag{2.13a}$$

for $\sum_i w_i \in 2\mathbb{Z}$ and

$$\left\langle V^{w_1}_{j_1,h_1,\bar{h}_1}(0;0) V^{w_2}_{j_2,h_2,\bar{h}_2}(1;1) V^{w_3}_{j_3,h_3,\bar{h}_3}(\infty;\infty) V^{w_4}_{j_4,h_4,\bar{h}_4}(x;z) \right\rangle$$
$$= \mathcal{N}(j_3) \int \prod_{i=1}^{4} \frac{\mathrm{d}^2 y_i}{\pi} \, y_i^{\frac{kw_i}{2}+j_i-h_i-1} \bar{y}_i^{\frac{kw_i}{2}+j_i-\bar{h}_i-1} |X_{123}|^{2(\frac{k}{2}-j_1-j_2-j_3-j_4)}$$
$$\times |X_1|^{2(-j_1+j_2+j_3+j_4-\frac{k}{2})} |X_2|^{2(j_1-j_2+j_3+j_4-\frac{k}{2})} |X_3|^{2(j_1+j_2-j_3+j_4-\frac{k}{2})} |X_4|^{-4j_4}$$
$$\times \left\langle V^0_{j_1}(0;0) V^0_{j_2}(1;1) V^0_{\frac{k}{2}-j_3}(\infty;\infty) V^0_{j_4}\left(\frac{X_2 X_{134}}{X_{123}X_4};z\right) \right\rangle \tag{2.13b}$$

for $\sum_i w_i \in 2\mathbb{Z}+1$. Let us spell out the various definitions entering eq. (2.13). The quantities $X_I$, defined for any subset $I \subseteq \{1,2,3,4\}$, are (suitably normalised) polynomials in $x, z, y_1, \dots, y_4$. The following definition is somewhat complicated and we state it for the sake of completeness. We invite the reader to ignore all normalisation prefactors.

$$X_I(x,z,y_1,y_2,y_3,y_4) = z^{\frac{1}{2}\delta_{\{1,4\}\subset I}} \left( (1-z)^{\frac{1}{2}}(-1)^{w_1 w_2 + w_1 w_3 + w_2 w_4 + w_3 w_4} \right)^{\delta_{\{2,4\}\subset I}}$$
$$\times \sum_{i\in I:\ \varepsilon_i = \pm 1} P_{\boldsymbol{w}+\sum_{i\in I}\varepsilon_i e_i}(x;z) \prod_{i\in I} y_i^{\frac{1-\varepsilon_i}{2}} . \tag{2.14}$$

The polynomials $P_{\boldsymbol{w}}(x;z)$ entering eq. (2.14) are defined (up to a convenient normalising prefactor) in terms of branched coverings $\gamma : \mathbb{CP}^1 \to \mathbb{CP}^1$ from the sphere to itself, with ramification indices $\boldsymbol{w} = (w_1, w_2, w_3, w_4)$,

$$P_{\boldsymbol{w}}(x;z) \equiv f(\boldsymbol{w})(1-x)^{\frac{1}{2}s(-w_1+w_2-w_3+w_4))}(1-z)^{\frac{1}{4}s((w_1+w_2-w_3-w_4)(w_1-w_2-w_3+w_4))-\frac{1}{2}w_2 w_4}$$
$$\times x^{\frac{1}{2}s(w_1-w_2-w_3+w_4)} z^{\frac{1}{4}s((w_1+w_2-w_3-w_4)(-w_1+w_2-w_3+w_4))-\frac{1}{2}w_1 w_4} \tilde{P}_{\boldsymbol{w}}(x;z) , \tag{2.15}$$

with

$$\tilde{P}_{\boldsymbol{w}}(x; z) \equiv \prod_{\gamma^{-1}} \left( z - \gamma^{-1}(x) \right) \ , \tag{2.16}$$

and

$$s(\alpha) = \begin{cases} \alpha \ , & \alpha > 0 \ , \\ 0 \ , & \alpha \leq 0 \ . \end{cases} \tag{2.17}$$

The function $f(\boldsymbol{w})$ entering eq. (2.15) is a phase. See [26] for its precise definition and for a more detailed explanation of the algorithm adopted to explicitly construct the polynomials $P_{\boldsymbol{w}}(x; z)$.

In plain English, $P_{\boldsymbol{w}}(x; z) = 0$ is exactly the condition for the existence of a branched covering map $\gamma : \mathbb{CP}^1 \to \mathbb{CP}^1$ as described above. The prefactor $\mathcal{N}(j)$ entering eq. (2.13b) is defined in eq. (2.4). Finally, we denoted by

$$\langle V_{j_1}^0(0; 0) V_{j_2}^0(1; 1) V_{j_3}^0(\infty; \infty) V_{j_4}^0(c; z) \rangle \tag{2.18}$$

the correlator of the four vertex operators in the unflowed sector. This correlator can be computed in principle via the conformal block expansion described in [11] and which we review in Appendix A.

## 2.3 Correlators of discrete representations

In the previous sections we discussed our proposal for three- and four-point correlators of *continuous* representations. Let us now review how to determine correlators with *discrete* representation insertions from these. It will be convenient in the following to adopt the prescription of [31]. This is slightly different than (but equivalent to) the prescription we used in our previous papers [25, 26]. For discrete representations of type $\mathcal{D}^+$ (respectively $\mathcal{D}^-$) the quantum numbers obey the additional integrality conditions $h - \frac{kw}{2} - j \in \mathbb{Z}_{\geq 0}$ (respectively $h - \frac{kw}{2} + j \in \mathbb{Z}_{\leq 0}$). These conditions lead to a divergence in the integral over the $y$-variables in our formulae for the correlation functions (2.3) and (2.13). These poles in the correlation functions were dubbed LSZ-poles in [31].

Correlation functions of discrete operators are then defined by extracting the corresponding residue of the pole in the spins. More precisely, set

$$h_i \to h_i + \delta h_i \ , \qquad \bar{h}_i \to \bar{h}_i + \delta h_i \ , \tag{2.19}$$

for each discrete representation in the formulae for the correlation function and extract the residue at $\delta h_i = 0$. For example, if the first insertion is a $\mathcal{D}^+$ representation we have,

$$\left\langle V_{j, h_1 \bar{h}_1}^{\mathcal{D}^+, w_1}(0; 0) \, V_{j_2, h_2, \bar{h}_2}^{\mathcal{C}, w_2}(1; 1) \, V_{j_3, h_3, \bar{h}_3}^{\mathcal{C}, w_3}(\infty; \infty) \right\rangle$$
$$\equiv \operatorname*{Res}_{\delta h_1 = 0} \left\langle V_{j_1, h_1 + \delta h_1, \bar{h}_1 + \delta h_1}^{\mathcal{C}, w_1}(0; 0) \, V_{j_2, h_2, \bar{h}_2}^{\mathcal{C}, w_2}(1; 1) \, V_{j_3, h_3, \bar{h}_3}^{\mathcal{C}, w_3}(\infty; \infty) \right\rangle \ , \tag{2.20}$$

where we added an extra label to denote the representation of each vertex operator. One can easily check that this is equivalent to our previous prescription in [25, 26]. It allows for a more uniform treatment, since we can almost exclusively discuss correlation functions of continuous vertex operators and then extract residues in the end if we are interested in the discrete case.

## 3 The analytic structure of worldsheet correlators

The four-point functions (2.13) turn out to have a very interesting singularity structure, revealing a lot about the dynamics of both the worldsheet theory and the spacetime CFT. In this section we analyse the analytic structure of worldsheet four-point functions: we identify the location of singularities in Section 3.1 and compute the precise behaviour of worldsheet correlators near these singularities in Section 3.2.

### 3.1 Location of the singularities

We are interested in singularities of the worldsheet four-point function

$$\left\langle V^{w_1}_{j_1,h_1}(0;0) V^{w_2}_{j_2,h_2}(1;1) V^{w_3}_{j_3,h_3}(\infty;\infty) V^{w_4}_{j_4,h_4}(x;z)\right\rangle , \tag{3.1}$$

for which a closed form formula is given in eq. (2.13). We already explained in the Introduction that a singularity means a subset of the 'position space' $(x,z)$-space, where the correlator diverges. There are some trivial singularities which we will not consider further, namely $x=0,\,1,\,\infty$ and $z=0,\,1,\,\infty$. Let us immediately point out one singularity that is already manifest. The factor

$$X^{j_1+j_2+j_3+j_4-k}_{\emptyset} = (P_{\boldsymbol{w}}(x;z))^{j_1+j_2+j_3+j_4-k} \tag{3.2}$$

entering the even parity correlator (2.13a), gives rise to a singularity in the $(x,z)$-space whenever $P_{\boldsymbol{w}}(x;z)=0$. Since the factor $X_{\emptyset}$ in (2.13a) does not depend on any $y_i$, it can be pulled out of the $y_i$ integrals. This singularity is hence already manifest before performing the $y_i$ integration in eq. (2.13). However, additional singularities in the $(x,z)$-space may appear (and actually do appear) after performing the $y_i$ integration. We are now going to explain how singularities of the four-point function (3.1) — i.e. singularities of the $y_i$ integral entering (2.13) — can be deduced from the singularities of its integrand.

**Singularities of the $y_i$-integrand** In order to understand when $(x,z)$-plane singularities of the integral (2.13) arise, it is useful to study singularities of the integrand, i.e. singularities in the $(y_1,y_2,y_3,y_4)$-space. Also in this case, by singularity we mean any branch surface of the chiral correlator that is present for generic choice of all the other parameters. For example, the factor $y_i^{\frac{kw_i}{2}+j_i-h_i-1}$ in (2.13) gives rise to a singularity in the $y$-space whenever $y_i=0$ or $y_i=\infty$. Note that since the singularities are all power-like, for generic values of the external parameters, the integrand tends to either zero or infinity when approaching the singularity.

We first discuss the even-parity case (2.13a). Each factor $X_{ij}$ leads to a singularity around the hypersurfaces $X_{ij}=0$. The integrand in eq. (2.13a) contains explicitly the hypersurfaces $X_{12}=0$, $X_{13}=0$, $X_{23}=0$ and $X_{34}=0$. Additional singularities appear due to the unflowed correlator

$$F(c,z) \equiv \left\langle V^0_{j_1}(0;0) V^0_{j_2}(1;1) V^0_{j_3}(\infty;\infty) V^0_{j_4}(c;z)\right\rangle , \qquad c \equiv \frac{X_{23}X_{14}}{X_{12}X_{34}} , \tag{3.3}$$

which is itself singular for $c\to 0$, $c\to 1$ or $c\to\infty$. This produces the remaining hypersurfaces of the form $X_{ij}=0$.[4] Finally, as follows from the KZ equation, the unflowed

---

[4]To study the singularity at $c=1$ we make use of the identity $1-\frac{X_{23}X_{14}}{X_{12}X_{34}}=\frac{X_{13}X_{24}}{X_{12}X_{34}}$, see [26].

correlator $F(c, z)$ can have one more singularity at $c = z$. In [26] we checked the validity of the identity

$$X_{12}X_{34}(c - z) = X_{14}X_{23} - zX_{12}X_{34} = X_\emptyset X_{1234} \qquad (3.4)$$

in numerous cases. One can see that the singularity of the unflowed correlator at $c = z$ gives rise to the additional singular hypersurface $X_{1234} = 0$. For the odd-parity case, one finds similarly that there are singularities at $X_i = 0$ and $X_{ijk} = 0$ before integration over $y_i$.

To summarise, the $y_i$ integrand in eq. (2.13) has the following singularites,[5]

$$Y_i^+ \equiv \{y_i = 0\}\,, \qquad Y_i^- \equiv \{y_i = \infty\}\,, \qquad S_I \equiv \{X_I = 0\}\,, \qquad (3.5)$$

for $i = 1, \ldots, 4$ and $I \subseteq \{1, 2, 3, 4\}$, where the parity of $I$ is equal to the parity of $\sum_i w_i$.

**Singularities of the $y_i$-integral** Let us now explain how singularities after doing the $y_i$-integral (2.13) can be deduced from the singularities of the $y_i$-integrand. In the following discussion, it is important to keep in mind that the location of the branch surfaces $S_I$ depends on $x$ and $z$, see eq. (2.14).

As it is familiar for instance from the KLT formula [45], the full non-chiral surface integral (2.13) can be decomposed into a finite sum of products of chiral integrals over chambers. Each chamber is bounded by a collection of the branch surfaces in eq. (3.5).[6] Making use of twisted homology techniques, we explained explicitly in [25] how to do this for a three-point function.

To illustrate this, let us consider a one-dimensional toy model,

$$\int_0^{z-x} \mathrm{d}y\, y^{a-1}\, (z - x - y)^{b-1} = (z - x)^{a+b-1} \frac{\Gamma(a)\Gamma(b)}{\Gamma(a + b)}\,. \qquad (3.6)$$

Eq. (3.6) is true for suitable choice of the parameters and can be extended by analytic continuation otherwise. Notice that the 'chamber' $[0, z - x]$ is bounded by the 'surfaces' $y = 0$ and $z - x - y = 0$. As we will explain in a moment, $(x, z)$ singularities of the integral may only appear when the integration chamber degenerates to zero size. This happens for $z = x$, which in fact is the only singularity, as one can directly check from the right hand side of (3.6).

Let us now focus on a chiral correlator in the decomposition over chambers we just discussed. It takes the form

$$\int_{\mathcal{C}} \prod_{i=1}^4 \mathrm{d}y_i \, \prod_{i=1}^4 y_i^{\frac{kw_i}{2}+j_i-h_i-1} \left\langle \prod_{i=1}^4 V_{j_i}^{w_i}(x_i; y_i; z_i) \right\rangle_{\text{chiral}}, \qquad (3.7)$$

---

[5]We deal with $y_i = \infty$ by projectivising these hypersurfaces and viewing them as hypersurfaces is $(\mathbb{CP}^1)^4$. We thus introduce another projective variable $u_i$ and trade the polynomials $X_I$ for polynomials that are homogeneous in $(u_i, y_i)$ for every $i$. In projective variables, $Y_i^+ = \{y_i = 0\}$, $Y_i^- = \{u_i = 0\}$ and e.g. $X_{12} = u_1 u_2 P_{\boldsymbol{w}+e_1+e_2} + y_1 u_2 P_{\boldsymbol{w}-e_1+e_2} + u_1 y_2 P_{\boldsymbol{w}+e_1-e_2} + y_1 y_2 P_{\boldsymbol{w}-e_1-e_2} = 0$. The original expressions can be recovered by setting $u_i = 1$.

[6]Here, we are making use of the fact that $(\mathbb{CP}^1)^4$ is a compact manifold. In the context of twisted homology, it is known that the set of bounded chambers is a basis of homology, i.e. that one can always express the integral (2.13) in terms of chiral integrals over bounded chambers only.

where $\mathcal{C}$ is the chamber we are integrating over. A singularity of the integral (3.7) (i.e. a branch surface in $(x, z)$-space) may only occur when the chamber $\mathcal{C}$ shrinks to zero size, i.e. when the hypersurfaces defining $\mathcal{C}$ meet in one point. In fact, if $\mathcal{C}$ has finite size, this integral converges provided that the exponents of the critical hypersurfaces appear with coefficient greater then $-1$.[7] For other values of the exponents it can be defined by analytic continuation.[8] Notice that we are not saying that when the chamber $\mathcal{C}$ shrinks to zero a $(x, z)$ singularity necessarily occurs, but that it may occur and that the integral converges otherwise. The chiral part of the above correlator may be taken to be one chiral term in a conformal block expansion.

The analysis of $(x, z)$ singularities is then reduced to investigating when integration chambers may shrink to zero size. In four-dimensional space, a chamber is bounded by at least five hypersurfaces. Notice that the location of the branch surfaces depends itself on $x$ and $z$. If the various branch surfaces were in generic position to each other for generic choices of $x$ and $z$, one would simply have to study when a chamber bounded by them shrinks to zero size, i.e. when five hypersurfaces meet at one point. However, it turns out that several of the hypersurfaces (3.5) intersect in lower codimension loci as one would generically expect. In other words, the hypersurfaces (3.5) are not in generic location. Thus, one should study for which values of $x$ and $z$ *exceptional* intersections arise. More practically, one should ask the following question. Consider five hypersurfaces that do not meet in one point for generic choices of $x$ and $z$; for which special values of $x$ and $z$ do these five hypersurfaces meet in one point?

Let us then discuss when exceptional intersections may occur. One can perform this computation directly for various choices of $\boldsymbol{w}$ and one finds that an exceptional intersection occurs precisely when $P_{\boldsymbol{w}+\boldsymbol{\varepsilon}}(x; z) = 0$, in which case

$$P_{\boldsymbol{w}+\boldsymbol{\varepsilon}}(x; z) = 0 \qquad \Longleftrightarrow \qquad S_I \cap \bigcap_{i \in I} Y_i^{\varepsilon_i} \neq \emptyset \ . \tag{3.8}$$

Here $\boldsymbol{\varepsilon} = (\varepsilon_1, \varepsilon_2, \varepsilon_3, \varepsilon_4)$ with $\varepsilon_i \in \{-1, 0, 1\}$ and for $i = 1, \ldots, 4$ we have $i \in I$ if $\varepsilon_i \neq 0$. In other words, for e.g. $P_{\boldsymbol{w}+(1,-1,0,0)}(x; z) = 0$, the surfaces $Y_1^+$, $Y_2^-$ and $S_{12}$ intersect in the points $y_1 = 0$ and $y_2 = \infty$, but would not do so if $P_{\boldsymbol{w}+(1,-1,0,0)}(x; z) \neq 0$. Note that the surfaces $Y_1^+$, $Y_2^-$ and $S_{12}$ do not depend on $y_3$ and $y_4$. Thus for generic choice of $(x, z)$, one would not expect such a triple intersection. As integration chamber $\mathcal{C}$, in this example we can hence take the chamber bounded by $S_{12}$, $Y_1^+$, $Y_2^-$ and two other surfaces, whose volume shrinks to zero for $P_{\boldsymbol{w}+(1,-1,0,0)}(x; z) \to 0$, thus indicating a possible singularity in the correlator (2.13).

The existence of this intersection (3.8) follows directly from the definition of $X_I$, see eq. (2.14). In fact, using the projectivised version of the hypersurfaces as described in footnote 5, all terms in $X_I$ vanish because either $u_i = 0$ or $y_i = 0$ for every $i$, except for a single term that vanishes because $P_{\boldsymbol{w}+\boldsymbol{\varepsilon}}(x; z) = 0$. The non-trivial statement is that these

---

[7]When considering the branch surface $Y_i^+ = \{y_i = \infty\}$, this condition amounts to requiring that the integrand decays faster than $y_i^{-1}$ at infinity.

[8]Alternatively, one can use a higher-dimension Pochhammer contour or the theory of twisted integration cycles [46].

are the only types of exceptional intersections. We have checked this claim extensively in `Mathematica`.

We should also mention that this discussion (and many related ones in this paper) is very similar to the standard Landau analysis in quantum field theory [47]. There one asks the similar question, namely given all the singularities of a loop integrand, how does one determine the singularities in the remaining scattering amplitude?

## 3.2 Worldsheet correlators near singularities

Let us now determine the leading behaviour of the worldsheet correlators close to the singularities. As we explained in the previous subsection, singularities arise from the collision of the surfaces $S_I$ and $Y_i^{\varepsilon_i}$ for $i \in I$ and $I \subseteq \{1, 2, 3, 4\}$. In the integration over $y_i$-space we should hence zoom in on the region $y_i = 0$ or $y_i = \infty$ for $i \in I$. In particular, in all non-singular factors of the integrand we can simply set $y_i = 0$ or $y_i = \infty$ for $i \in I$. As we shall see, this will simplify the integral dramatically and we will be able to perform it explicitly.

The unflowed correlator (3.3) directly enters in eq. (2.13). Let us then discuss its behaviour near the singularities listed in the previous section. Eq. (C.8) implies that the 'cross ratios' $c = \frac{X_{14} X_{23}}{X_{12} X_{34}}$ or $c = \frac{X_{134} X_2}{X_{123} X_4}$ entering respectively eqs. (2.13a) and (2.13b) approach 0, 1, $z$ or $\infty$ near the intersection of the surfaces $Y_i$ for $i \in I$ and $S_I$. Near these points, the unflowed correlator simplifies significantly.

**The worldsheet correlator near $P_w(x; z) = 0$**  Let us first explain this at the simplest case where $I = \emptyset$, which implies in particular $\sum w_i \in 2\mathbb{Z}$. We have

$$c\big|_{P_w(x;z)=0} \equiv \frac{X_{14} X_{23}}{X_{12} X_{34}}\bigg|_{P_w(x;z)=0} = z \ . \tag{3.9}$$

As discussed in Appendix A, the unflowed correlator near $c = 0$, $1$, $\infty$ or $z$ can be expressed in terms of correlators of Liouville theory. The Liouville theory in question has parameters $Q_{\mathrm{L}} = b + b^{-1}$ with $b = \frac{1}{\sqrt{k-2}}$, and as usual $c_{\mathrm{L}} = 1 + 6 Q_{\mathrm{L}}^2$. More precisely, the correlators are analytically continued Liouville-correlators with momenta $\alpha_i$ outside of the line $\mathrm{Re}(\alpha_i) = \frac{Q_{\mathrm{L}}}{2}$. Since correlation functions of Liouville theory are meromorphic functions of the momenta, it is always possible to uniquely analytically continue them and this is what the correlators $\langle \dots \rangle_{\mathrm{L}}$ denote in the following. In our case, writing $F(c, z)$ for the unflowed correlator (3.3), we find

$$\lim_{c \to z} F(c, z) = \frac{\gamma(-\frac{1}{k-2})^2 \, \gamma(k - \sum_i j_i)}{2\pi^2 \sqrt{k-2} \, \nu^{\sum_i j_i - k} \prod_{i=1}^4 \gamma(\frac{2j_i - 1}{k-2})} \langle V_{bj_1}(0) V_{bj_2}(1) V_{bj_3}(\infty) V_{bj_4}(z) \rangle_{\mathrm{L}} \tag{3.10}$$

and

$$\lim_{c \to z} |c - z|^{2(j_1 + j_2 + j_3 + j_4 - k)} F(c, z)$$

$$= |z|^{2(j_2 + j_3 - \frac{k}{2})} |1 - z|^{2(j_1 + j_3 - \frac{k}{2})} \frac{\gamma(\frac{1}{2-k})^2 \gamma(\sum_i j_i - k)}{2\pi^2 \sqrt{k-2} \, \nu^{\sum_i j_i - k}}$$

$$\times \ \langle V_{b(\frac{k}{2}-j_1)}(0)V_{b(\frac{k}{2}-j_2)}(1)V_{b(\frac{k}{2}-j_3)}(\infty)V_{b(\frac{k}{2}-j_4)}(z)\rangle_{\mathrm{L}} \ . \quad (3.11)$$

The notation means that $F(c, z)$ has two possible critical behaviours and we pick out the relevant one in the limit. For the other singularities, similar results are collected in Appendix B.3. Notice that the exponent of $|c-z|$ in eqs. (3.10) and (3.11) can alternatively be deduced by solving the KZ equation (A.1) near $c = z$. Since the KZ equation is second order, there are always two possible exponents of the singularity. Even though this is maybe simpler than the procedure we adopted to derive eqs. (3.10) and (3.11), the correspondence with Liouville correlators has one important advantage: it gives complete control over the normalisation of the unflowed correlator. This will turn out to be important in the following.

As already mentioned above, the unflowed correlator can have two different behaviours near the singularity $P_{\boldsymbol{w}}(x; z) = 0$, see eqs. (3.10) and (3.11). We note that only the behaviour of eq. (3.10) leads to a singularity in the full correlator of spectrally flowed fields. Indeed, the critical behaviour of eq. (3.11) would cancel the singularity that is manifestly present in the $X_\emptyset$ prefactor of (2.13a), compare the exponents in (3.2) and in (3.11).

Making use of eq. (3.10) and of the identities (C.1), (C.5) and (C.4) — or alternatively the ancillary `Mathematica` notebook — one shows that the correlator behaves as follows near the locus $P_{\boldsymbol{w}}(x; z) = 0$:

$$\left\langle V_{j_1,h_1}^{w_1}(0;0)V_{j_2,h_2}^{w_2}(1;1)V_{j_3,h_3}^{w_3}(\infty;\infty)V_{j_4,h_4}^{w_4}(x;z)\right\rangle\Big|_{P_{\boldsymbol{w}}(x;z)=0}$$

$$= \frac{\gamma(\frac{1}{2-k})^2\,\gamma(k-\sum_i j_i)}{2\pi^2\sqrt{k-2}\,\nu^{\sum_i j_i - k}\,\prod_{i=1}^4 \gamma(\frac{2j_i-1}{k-2})}|z-z_\gamma|^{2(j_1+j_2+j_3+j_4-k)}\,|z|^{2(-j_2-j_3+\frac{k}{2})}\,|1-z|^{2(-j_1-j_3+\frac{k}{2})}$$

$$\times \ |\Pi|^{-k}\,\langle V_{bj_1}(0)V_{bj_2}(1)V_{bj_3}(\infty)V_{bj_4}(z)\rangle_{\mathrm{L}} \prod_{i=1}^4 w_i^{2j_i-\frac{k}{2}(w_i+1)}|a_i|^{-\frac{k}{2}(w_i+1)-2j_i}$$

$$\times \ \int \prod_{i=1}^4 \frac{\mathrm{d}^2 y_i}{\pi}\,y_i^{\frac{kw_i}{2}+j_i-h_i-1}\,\bar{y}_i^{\frac{kw_i}{2}+j_i-\bar{h}_i-1}(1+a_i^{-1}y_i)^{-2j_i}(1+\bar{a}_i^{-1}\bar{y}_i)^{-2j_i} \ . \quad (3.12)$$

Here, $\gamma$ indexes the solutions to $P_{\boldsymbol{w}}(x; z) = 0$ for fixed $x$, i.e. $P_{\boldsymbol{w}}(x; z_\gamma) = 0$. To each solution $\gamma$, we have by definition of $P_{\boldsymbol{w}}(x; z)$ (explained in Section 2.2) an associated branched covering map $\gamma$. The quantities $a_i$ are the coefficients entering the Taylor expansion of the covering map $\gamma(\zeta)$ for $\zeta$ close to the ramification points $z_i$,

$$\gamma(\zeta) = x_i + a_i(\zeta - z_i)^{w_i} + \mathcal{O}\big((\zeta - z_i)^{w_i+1}\big) \ , \quad (3.13)$$

while $\Pi$ denotes the product of the residues of the covering map $\gamma(\zeta)$,

$$\Pi \equiv w_3^{-w_3-1} \prod_{a=1}^{\mathrm{N}-w_3} \Pi_a \ , \quad (3.14)$$

and $\Pi_a$ are the individual residues. We denoted by N the degree of the covering map. The product runs up to $\mathrm{N} - w_3$ since we put the third field at infinity (i.e. $x_3 = z_3 = \infty$),

which accounts for $w_3$ poles. The prefactor in (3.14) is natural since it produces symmetric formulae. We should also mention that (3.12) gives the leading term in the expansion of the correlator in $z$ around $z_\gamma$. Of course there are subleading terms that are harder to compute. It is now simple to integrate over $y_i$ in eq. (3.12). We obtain

$$
\left\langle V_{j_1,h_1}^{w_1}(0;0)V_{j_2,h_2}^{w_2}(1;1)V_{j_3,h_3}^{w_3}(\infty;\infty)V_{j_4,h_4}^{w_4}(x;z)\right\rangle\Big|_{P_{\boldsymbol{w}}(x;z)=0}
$$
$$
= \frac{\gamma(\frac{1}{2-k})^2\,\gamma(k-\sum_i j_i)}{2\pi^2\sqrt{k-2}\,\nu^{\sum_i j_i-k}}\prod_{i=1}^{4}\frac{\gamma(\frac{kw_i}{2}+j_i-h_i)\gamma(1-2j_i)}{\gamma(\frac{kw_i}{2}-j_i-h_i+1)\gamma(\frac{2j_i-1}{k-2})}
$$
$$
\times\,|z-z_\gamma|^{2(j_1+j_2+j_3+j_4-k)}\,|z|^{2(-j_2-j_3+\frac{k}{2})}\,|1-z|^{2(-j_1-j_3+\frac{k}{2})}|\Pi|^{-k}
$$
$$
\times\,\langle V_{bj_1}(0)V_{bj_2}(1)V_{bj_3}(\infty)V_{bj_4}(z)\rangle_{\mathrm{L}}\prod_{i=1}^{4}w_i^{2j_i-\frac{k}{2}(w_i+1)}a_i^{\frac{k(w_i-1)}{4}-h_i}\bar{a}_i^{\frac{k(w_i-1)}{4}-\bar{h}_i}\,. \tag{3.15}
$$

**The worldsheet correlator near $P_{\boldsymbol{w+\varepsilon}}(x;z)=0$**  A similar analysis can be carried out for the other singularities. The general formula for the correlator near the singularity $P_{\boldsymbol{w+\varepsilon}}(x;z)=0$ reads

$$
\left\langle V_{j_1,h_1}^{w_1}(0;0)V_{j_2,h_2}^{w_2}(1;1)V_{j_3,h_3}^{w_3}(\infty;\infty)V_{j_4,h_4}^{w_4}(x;z)\right\rangle\Big|_{P_{\boldsymbol{w+\varepsilon}}(x;z)=0}
$$
$$
= (z-z_\gamma)^\eta\,z^{\gamma_{14}}\,(1-z)^{\gamma_{24}}\,\Pi^{-\frac{k}{2}}\prod_{i=1}^{4}\tilde{w}_i^{j_i(1-\varepsilon_i^2)-(h_i-\frac{k\tilde{w}_i}{2})\varepsilon_i-\frac{k(\tilde{w}_i+1)}{4}}a_i^{\frac{k(\tilde{w}_i-1)}{4}-h_i}\,\times\,\mathrm{c.c.}
$$
$$
\times\,f_{|\boldsymbol{\varepsilon}|}(z)\int\prod_{i=1}^{4}\frac{\mathrm{d}^2y_i}{\pi}\,y_i^{B_i-1}\left(1-\sum_{i\in I}y_i\right)^{A-1}\prod_{i\notin I}(1-y_i)^{-2j_i}\,\times\,\mathrm{c.c.}\,. \tag{3.16}
$$

Here,

$$
\eta = -k+\sum_{i=1}^{4}\left[\varepsilon_i\left(\frac{k\tilde{w}_i}{2}-h_i\right)+j_i\left(1-\varepsilon_i^2\right)\right]\,, \tag{3.17a}
$$

$$
\gamma_{14} = \frac{k}{2}+\sum_{i=2,3}\varepsilon_i\left(h_i-\frac{k\tilde{w}_i}{2}\right)-\varepsilon_1^2 j_4-(1-\varepsilon_2^2)j_2-(1-\varepsilon_3^2)j_3-\varepsilon_4^2 j_1
$$
$$
-\varepsilon_1^2\varepsilon_4^2\left(\sum_{i=2,3}(2\varepsilon_i^2-1)\left(j_i-\frac{k}{4}\right)-j_1-j_4\right)\,, \tag{3.17b}
$$

$$
\gamma_{24} = \frac{k}{2}+\sum_{i=1,3}\varepsilon_i\left(h_i-\frac{k\tilde{w}_i}{2}\right)-(1-\varepsilon_1^2)j_1-\varepsilon_2^2 j_4-(1-\varepsilon_3^2)j_3-\varepsilon_4^2 j_2
$$
$$
-\varepsilon_2^2\varepsilon_4^2\left(\sum_{i=1,3}(2\varepsilon_i^2-1)\left(j_i-\frac{k}{4}\right)-j_2-j_4\right)\,, \tag{3.17c}
$$

$$
A = k+\sum_{i=1}^{4}\left[\left(2j_i-\frac{k}{2}\right)\left(1-\varepsilon_i^2\right)-j_i\right]+1\,, \tag{3.17d}
$$

$$B_i = \begin{cases} j_i + h_i - \frac{k\tilde{w}_i}{2} - \frac{k}{2} \,, & \varepsilon_i = -1 \,, \\ j_i - h_i + \frac{k\tilde{w}_i}{2} \,, & \varepsilon_i = 0 \,, \\ j_i - h_i + \frac{k\tilde{w}_i}{2} - \frac{k}{2} \,, & \varepsilon_i = 1 \,, \end{cases} \tag{3.17e}$$

Recall that the set $I \subseteq \{1, 2, 3, 4\}$ was defined as those $i$ for which $\varepsilon_i \neq 0$. In eq. (3.16) c.c. denotes the contribution of the right-movers, which is similar to the chiral structure. We also introduced the notation $\tilde{w}_i \equiv w_i + \varepsilon_i$. Notice that $\tilde{w}_i > 0$, as otherwise there would be no solution $z_\gamma$ for $P_{\boldsymbol{w}+\boldsymbol{\varepsilon}}(x; z_\gamma) = 0$. The quantities $a_i$ and $\Pi$, introduced respectively in eqs. (3.13) and (3.14), refer to the covering map with ramification indices $\tilde{w}_i$. The function $f_{|\boldsymbol{\varepsilon}|}(z)$ depending on $|\boldsymbol{\varepsilon}| = (|\varepsilon_1|, |\varepsilon_2|, |\varepsilon_3|, |\varepsilon_4|)$ is defined in Appendix B.3. It is always a Liouville four-point function up to some simple normalising prefactors.[9]

Let us comment on the main steps in the computation of eq. (3.16). The reader can find an explicit example in Appendix D. In arriving at eq. (3.16) we have changed the variables of integration $y_i$. In particular, we first changed $y_i \to y_i^{-1}$ in the case of $\varepsilon_i = -1$ and then we changed $y_i \to \frac{P_{\boldsymbol{w}+\boldsymbol{\varepsilon}}}{P_{\boldsymbol{w}-2e_i}} y_i$ for $\varepsilon_i \neq 0$ and $y_i \to a_i y_i$ for $\varepsilon_i = 0$. Similarly to what we explained in the case $P_{\boldsymbol{w}}(x; z) = 0$ above, we made a choice between the two possible behaviours of the unflowed correlator in the neighbourhood of $P_{\boldsymbol{w}+\boldsymbol{\varepsilon}}(x; z) = 0$. Also in this case, one of the two possibilities always leads to a regular result. Let us briefly explain how this comes about. When choosing the behaviour of the unflowed correlator differently from what we did in (3.16), the (chiral) critical exponent reads

$$\eta' = \sum_{i=1}^4 \varepsilon_i \left( \frac{k\, w_i}{2} + \varepsilon_i j_i - h_i \right) \,. \tag{3.18}$$

However, because of the change of integration variables mentioned above, the only dependence on $y_i$ for $i \in I$ is

$$\int \prod_{i \in I} \mathrm{d}^2 y_i \, y_i^{\varepsilon_i \left( \frac{kw_i}{2} + \varepsilon_i j_i - h_i \right) - 1} \, \bar{y}_i^{\varepsilon_i \left( \frac{kw_i}{2} + \varepsilon_i j_i - \bar{h}_i \right) - 1} \propto \prod_{i \in I} \delta^2 \left( \frac{kw_i}{2} + \varepsilon_i \, j_i - h_i \right) \,, \tag{3.19}$$

leading to delta functions that in turn make the critical exponent (3.18) vanish.

When performing the integration over $y_i$ in eq. (3.16) one finds that the integral over $y_i$ in the second line of (3.16) gives[10]

$$\frac{\gamma(A) \prod_{i=1}^4 \gamma(B_i) \prod_{i \notin I} \gamma(1 - 2j_i)}{\gamma \left( A + \sum_{i \in I} B_i \right) \prod_{i \notin I} \gamma(B_i + 1 - 2j_i)} \,. \tag{3.20}$$

Eqs. (3.16) and (3.20) are valid when all vertex operators sit in continuous representations. However, when discrete representations are present, it is easy to extract the corresponding result following the procedure outlined in Section 2.3.

In case that $w_i = 0$ for some $i \in \{1, 2, 3, 4\}$, we necessarily have $\varepsilon_i = 1$, since otherwise there is no singularity and eqs. (3.16) and (3.20) still apply.[11] For example, when all vertex

---

[9]Recall from Section 2 that in the parity odd case one additionally has to replace $j_3 \to \frac{k}{2} - j_3$ in $f_{|\boldsymbol{\varepsilon}|}(z)$.

[10]See e.g. [25, Appendix C] for the integral identities used here.

[11]Notice that for the string theory application we discuss later unflowed vertex operators must sit in discrete representations, as the mass-shell condition is otherwise violated. As in the generic case, the result for discrete correlators can be extracted following the procedure outlined in Section 2.3.

operators are unflowed, we must have $\varepsilon_i = 1$ for all $i \in \{1, 2, 3, 4\}$. The relevant covering map is the identity, i.e. $z_\gamma = x$. Thus we have

$$\left\langle V_{j_1}^0(0;0) V_{j_2}^0(1;1) V_{j_3}^0(\infty;\infty) V_{j_4}^0(x;z) \right\rangle \Big|_{x \sim z} \propto |z - x|^{2(k - j_1 - j_2 - j_3 - j_4)} , \qquad (3.21)$$

in agreement with the worldsheet analysis of Maldacena and Ooguri in [17].

**Further simplifications**  If the spins $j_i$ satisfy special relations, the appearing Liouville correlator can be written explicitly in terms of free boson correlators through the Coulomb gas formalism. These special relations will be very interesting in the string theory context. Let us explain this for the case $P_{\boldsymbol{w}}(x;z) = 0$, since we will make use of it later. Let us suppose that the spins are chosen such that

$$\epsilon \equiv \sum_{i=1}^4 j_i - k + 1 \sim 0 \qquad (3.22)$$

is close to zero. This translates to

$$\sum_{i=1}^4 \alpha_i - Q_{\mathrm{L}} \sim 0 \qquad (3.23)$$

in the Liouville language, where we recall that $\alpha_i = b j_i$ and $b^{-2} = k - 2$. Since this condition corresponds to momentum-conservation in the linear dilaton theory, Liouville correlators reduce to free field correlators and we have

$$\left\langle V_{bj_1}(0) V_{bj_2}(1) V_{bj_3}(\infty) V_{bj_4}(z) \right\rangle_{\mathrm{L}} \Big|_{\epsilon \equiv \sum_i j_i - k + 1 \sim 0} = \frac{\sqrt{k-2}}{\epsilon} |z|^{-\frac{4 j_1 j_4}{k-2}} |z - 1|^{-\frac{4 j_2 j_4}{k-2}} , \qquad (3.24)$$

see eq. (A.31). Hence, eq. (3.10) reduces to

$$F(c, z) \Big|_{\substack{c \sim z, \\ \epsilon \equiv \sum_i j_i - k + 1 \sim 0}} = \frac{\gamma(\frac{1}{2-k})^2}{2\pi^2 \, \nu^{k-2} \prod_{i=1}^4 \gamma(\frac{2 j_i - 1}{k-2})} |z|^{-\frac{4 j_1 j_4}{k-2}} |z - 1|^{-\frac{4 j_2 j_4}{k-2}} . \qquad (3.25)$$

A similar analysis of the limiting behaviour of $F(c, z)$ can be carried out for the singularities at $c \sim 0, 1, \infty$. We collect the results in Appendix B.4. For the special choice of spins in (3.22), eq. (3.25) leads to the following formula for the singular behaviour of the four-point function near $P_{\boldsymbol{w}}(x;z) = 0$

$$\left\langle V_{j_1,h_1}^{w_1}(0;0) V_{j_2,h_2}^{w_2}(1;1) V_{j_3,h_3}^{w_3}(\infty;\infty) V_{j_4,h_4}^{w_4}(x;z) \right\rangle \Bigg|_{\substack{P_{\boldsymbol{w}}(x;z)=0, \\ \epsilon = \sum_i j_i - k + 1}}$$

$$= \frac{\gamma(\frac{1}{2-k})^2 \nu^{2-k}}{2\pi^2 (k-2)^4} |z - z_\gamma|^{2(\epsilon - 1)} |z|^{-\frac{(2 j_1 + 2 - k)(2 j_4 + 2 - k)}{k-2}} |1 - z|^{-\frac{(2 j_2 + 2 - k)(2 j_4 + 2 - k)}{k-2}}$$

$$\times |\Pi|^{-k} \prod_{i=1}^4 R(j_i, h_i, \bar{h}_i) \, w_i^{2 j_i - \frac{k}{2}(w_i + 1)} a_i^{\frac{k(w_i - 1)}{4} - h_i} \bar{a}_i^{\frac{k(w_i - 1)}{4} - \bar{h}_i} , \qquad (3.26)$$

where

$$R(j, h, \bar{h}) = \frac{(k-2) \, \nu^{1-2j} \, \gamma\left(h - \frac{kw}{2} + j\right)}{\gamma\left(\frac{2j-1}{k-2}\right) \gamma\left(h - \frac{kw}{2} + 1 - j\right) \gamma(2j)} \qquad (3.27)$$

is the reflection coefficient entering two-point functions.

## 4 The analytic structure of string correlators

From a string theory perspective, we are ultimately only interested in spacetime singularities and we will discuss them now. As mentioned in the Introduction, by spacetime singularities, we mean poles in the spacetime correlation function as a function of the spins $j_i$. There are two sources of such singularities. Either the singularities are already present in the unflowed correlator that enters the integrands of (2.13a) and (2.13b) or they arise through integration over the $y_i$-coordinates and the cross-ratio $z$.

The final result for the location of the spacetime singularities is simple, even though the discussion that leads to it is slightly messy and involves several case distinctions. We will find that the spacetime correlator (with only continuous representations) has a pole whenever

$$\sum_i j_i = (n+a)(k-2) + m + 3 \tag{4.1}$$

with $m, n \in \mathbb{Z}_{\geq 0}$ together with all the reflected possibilities where we replace any $j_i$ by $1 - j_i$. The parameter $a$ is

$$a = \begin{cases} -1 \,, & \sum_i w_i \geq 2\max_i(w_i) + 2 \quad \text{and} \quad \sum_i w_i \in 2\mathbb{Z} \,, \\ -\frac{1}{2} \,, & \sum_i w_i \geq 2\max_i(w_i) + 1 \quad \text{and} \quad \sum_i w_i \in 2\mathbb{Z} + 1 \,, \\ 0 \,, & \sum_i w_i = 2\max_i(w_i) \,, \\ \frac{1}{2} \,, & \sum_i w_i = 2\max_i(w_i) - 1 \,, \\ 1 \,, & \sum_i w_i = 2\max_i(w_i) - 2 \,. \end{cases} \tag{4.2}$$

These are the so-called bulk poles. As we mentioned in Section 2.3, there are also the LSZ-type poles that arise from the presence of discrete representations. They are already manifestly present in the worldsheet correlators in the $h$-basis and thus we will focus our attention on the bulk poles. In this section, we will only determine the location of the spacetime poles and not the residues themselves. Apart from a few exceptions that we discuss in Section 5, these are quite complicated.

We also want to mention that the structure of (3.16) and (4.2) is heavily constrained by the identification of discrete representations with different amounts of spectral flow as $[\mathcal{D}_j^+]^w = [\mathcal{D}_{\frac{k}{2}-j}^-]^{w+1}$. Indeed, this identification predicts that a spacetime correlator

$$\left\langle\!\!\left\langle V_{j_1,h_1}^{w_1}(x_1) \cdots V_{j_n,h_n}^{w_n}(x_n) \right\rangle\!\!\right\rangle \tag{4.3}$$

has the same singularities as the correlator with $w_i \to w_i + 1$ and $j_i \to \frac{k}{2} - j_i$.[12] We use here and in the following double brackets for string correlators.

In the rest of this Section we explain the derivation of this result. Readers only interested in its physical implications are invited to skip ahead to Section 5.

---

[12]This can fail because the relevant correlator with discrete representions can vanish. This leads to the appearance of the special cases in (4.2).

### 4.1 Generalities

Let us first explain some generalities. The combined integral over the $y_i$'s and the modulus $z$ is for the purposes of spacetime singularities of hypergeometric type. This means that all the singularities of the integrand are of the form

$$\left| \prod_i F_i(y_1, y_2, y_3, y_4, z)^{a_i} \right|^2 \tag{4.4}$$

for some polynomials $F_i$ and some complex exponents $a_i$. The integrand is of course not literately of this form because the unflowed correlator that enters formulae (2.13a) and (2.13b) is more complicated. However, we have seen that for any singular locus of the integrand, the unflowed correlator simplifies and has this rational behaviour, see e.g. the formulae in Appendix B.3. Thus, for the purpose of analysing singularities, we can assume that the integrand is of this form.

This general behaviour immediately tells us that the spacetime correlator will be a meromorphic function in all the external variables (at least for the genus 0 correlators that we are considering here). This follows from the general theory of hypergeometric integrals, see e.g. [46]. Indeed, singularities in the spacetime correlator come from regions in the $(y_i, z)$-space where one or several of the $F_i$'s go to zero.

Let's consider the case where some number of $F_i$'s go to zero, say $F_1, \ldots, F_n$ on a codimension $m$ subvariety of the $(y_i, z)$-space. We can also assume that all zeros are simple, since for a $r$-fold zero we could replace the corresponding $F_i$ by its $r$-th root, which would be well-defined near the singularity. This tells us that if one of the $F_i$'s has a $r$-th order zero, the effective exponent will be $r a_i$ instead of $a_i$. Since we are only interested in determining the possibility of a pole and not the residue itself, we may suppress all the other regular factors that the integrand contains. Let us now choose local holomorphic coordinates of the $(y_i, z)$-space such that the codimension $m$ subvariety is described by the coordinates $t_1, \ldots, t_{5-m}$, whereas the normal space is described by the coordinates $s_1, \ldots, s_m$. In other words, the critical subvariety is locally described by $s_1 = \cdots = s_m = 0$. We then have

$$F_i \sim \sum_{p=1}^{m} f_{ip}(t_1, \ldots, t_{5-m}) s_p \tag{4.5}$$

to first order near the subvariety. It is finally convenient to choose real polar coordinates for the normal space $(s_1, \ldots, s_m)$ whose radial direction we denote by $\rho$ and the angular direction by $\Omega_{2m-1}$. The integrand locally behaves as

$$\int \prod_{\ell=1}^{5-m} \mathrm{d}^2 t_\ell \prod_{p=1}^{m} \mathrm{d}^2 s_p \left| \prod_i F_i(y_1, y_2, y_3, y_4, z)^{a_i} \right|^2$$

$$\sim \int \prod_{\ell=1}^{5-m} \mathrm{d}^2 t_\ell \, \mathrm{d}^{2m-1} \Omega_{2m-1} \, \mathrm{d}\rho \, \rho^{2m-1} \rho^{\sum_i a_i + \sum_i \bar{a}_i} f(t_\ell, \Omega_{2m-1}) \tag{4.6}$$

for some regular function $f(t_\ell, \Omega_{2m-1})$. In this form, only the integral over $\rho$ can lead to a pole. It can be done explicitly and leads to the condition

$$\sum_i a_i + \sum_i \bar{a}_i \in \mathbb{Z}_{\leq -2m} \tag{4.7}$$

for a singularity. Indeed, the leading singularity $\sum_i a_i + \sum \bar{a}_i = -2m$ is manifest in the integral over $\rho$. If we would expand the integral also to subleading powers in $\rho$ then $\sum_i a_i$ would be replaced by $\sum_i a_i + $ positive integer, thereby leading to the whole tower of poles.

### 4.2 Poles arising from integration

Let us analyse the poles of spacetime correlators that arise from the integral over the $y_i$'s and $z$. Further singularities, which may already be present in the unflowed four-point function *before* any integral is carry out, will be considered below. Since we know that the final correlation function is reflection symmetric, it suffices to look for poles that treat all the four points symmetrically and hence put conditions on $\sum_i j_i$. We organise the discussion according to the codimension $m = 1, \ldots, 5$ of the singular locus.

**Codimension 1** At codimension 1 there are two candidate singular loci,

$$X_\emptyset = 0 \qquad \text{and} \qquad X_{1234} = 0 . \tag{4.8}$$

Both appear in the parity even case. In the first case, since $X_\emptyset \equiv P_{\boldsymbol{w}}(x; z)$, poles of the spacetime correlator follow from integrating eq. (3.12). We encounter a pole whenever $\boldsymbol{w}$ admits a covering map and

$$\sum_i j_i - k \in \mathbb{Z}_{\leq -1} . \tag{4.9}$$

We will discuss the first singularity in this series extensively in Section 5. The second locus in eq. (4.8) originates from the unflowed correlator in the integrand. For $X_{1234} \to 0$, the generalized cross ratio $c$ in the unflowed correlator tends to $z$ and we can make use of eq. (B.15f). Omitting regular prefactors, we have

$$\left\langle V_{j_1}^0(0;0) V_{j_2}^0(1;1) V_{j_3}^0(\infty;\infty) V_{j_4}^0(c;z) \right\rangle \overset{c \to z}{\sim} |c - z|^{2(k - \sum_i j_i)}$$
$$\gamma\left(\textstyle\sum_i j_i - k\right) \left\langle V_{b(\frac{k}{2}-j_1)}(0) V_{b(\frac{k}{2}-j_2)}(1) V_{b(\frac{k}{2}-j_3)}(\infty) V_{b(\frac{k}{2}-j_4)}(z) \right\rangle_{\mathrm{L}} . \tag{4.10}$$

The combination in the second line is regular for $k - \sum_i j_i \in \mathbb{Z}_{\leq -1}$ and hence integration over $z$ gives rise to a pole whenever

$$k - \sum_i j_i \in \mathbb{Z}_{\leq -1} . \tag{4.11}$$

**Codimension 2** There is one codimension 2 condition that is symmetric in all the spins, namely

$$X_I = 0 , \qquad \text{for each } I \subset \{1, 2, 3, 4\} \text{ with } |I| = 3 . \tag{4.12}$$

Even though these look like four conditions, one can check that only two are independent. The generalised cross ratio entering the unflowed correlator is regular and we only have to

look at the prefactors in (2.13b). The only singular term is $X_{123}$ and from eq. (4.7) we find poles at

$$\frac{k}{2} - \sum_i j_i \in \mathbb{Z}_{\leq -2} \ . \tag{4.13}$$

**Codimension 3**   At codimension 3 the unique symmetric possibility is

$$X_I = 0 \ , \qquad \text{for each } I \subset \{1,2,3,4\} \text{ with } |I| = 2 \text{ or } |I| = 4 \ . \tag{4.14}$$

These seem like 7 conditions, but they again only lead to a codimension 3 singularity. The generalised cross ratio is again generic and the location of poles can again be read off from the prefactors. Upon summing all the exponents of the $X_{ij}$ prefactors in (2.13a) and using (4.7) we find

$$-\sum_i j_i \in \mathbb{Z}_{\leq -3} \ . \tag{4.15}$$

**Codimension 4**   Here the natural candidate is

$$X_I = 0 \ , \qquad \text{for each } I \subset \{1,2,3,4\} \text{ with } |I| = 1 \text{ or } |I| = 3 \ . \tag{4.16}$$

One may check again that there are only 4 independent conditions in this case. Also in this case the generalised cross ratio is generic. One can also check that $X_{ij\ell}$ goes to zero at twice the speed compared to $X_i$. It hence contributes twice as much to the critical exponent than $X_i$. Summing up all the exponents of the prefactors in eq. (2.13b) with a weight 2 for $X_{ij\ell}$ leads to poles at

$$-\sum_i j_i - \frac{k}{2} \in \mathbb{Z}_{\leq -4} \ . \tag{4.17}$$

**Codimension 5**   It should by now be clear that the codimension 5 singularity condition is

$$X_I = 0 \ , \qquad \text{for each } I \subset \{1,2,3,4\} \text{ with } |I| = 0 \ , |I| = 2 \text{ or } |I| = 4 \ . \tag{4.18}$$

Strictly speaking these are still only four independent conditions, since we only added one condition compared to the codimension 3 singularity. However if $X_\emptyset = 0$, the $X_{ij}$'s factor fall into two irreducible factors (see eq. (C.5)) and we can require that both factors vanish separately. In other words, we impose that the $X_{ij}$'s vanish to second order. It then turns out that $X_{1234}$ vanishes even to third order. The argument of the unflowed correlator is again generic in this limit. Thus we obtain the condition for singularities by summing all the exponents of the prefactors in eq. (2.13a) weighted with their respective order of vanishing. Overall, poles emerge for

$$-\sum_i j_i - k \in \mathbb{Z}_{\leq -5} \ . \tag{4.19}$$

## 4.3 Poles from the unflowed correlators

The unflowed correlator enters the formulae (2.13a) and (2.13b) and may itself have poles before any integration is carried out. These poles obviously propagate to the full string correlator. We analysed in Appendix B.1 precisely when this happens, which is for

$$\sum_i j_i = (n+2)(k-2) + m + 3 \quad \text{or} \quad \sum_i j_i = -n(k-2) - m + 1 . \tag{4.20}$$

Remembering that for $\sum_i w_i \in 2\mathbb{Z} + 1$ we have to replace $j_3$ by $\frac{k}{2} - j_3$ (see eq. (2.13b)), the unflowed correlator has the following singularities in $\sum_i j_i$

$$\sum_i j_i = \left(n + \frac{3}{2}\right)(k-2) + m + 3 \quad \text{or} \quad \sum_i j_i = -\left(n - \frac{1}{2}\right)(k-2) - m + 1 . \tag{4.21}$$

Notice that the singularities that we found in the previous section nicely combine with the series of singularities in eqs. (4.20) and (4.21). In fact, for the correlator with $\sum_i w_i \in 2\mathbb{Z}$, the full set of singularities is

$$\sum_i j_i = (n-1)(k-2) + m + 3 \quad \text{or} \quad \sum_i j_i = -(n-1)(k-2) - m + 1 , \tag{4.22}$$

with $m, n \in \mathbb{Z}_{\geq 0}$. Instead, for $\sum_i w_i \in 2\mathbb{Z} + 1$ we obtain

$$\sum_i j_i = \left(n - \frac{1}{2}\right)(k-2) + m + 3 \quad \text{or} \quad \sum_i j_i = -\left(n - \frac{1}{2}\right)(k-2) - m + 1 , \tag{4.23}$$

where as always $m, n \in \mathbb{Z}_{\geq 0}$. As a consistency check, we note that these conditions are reflection symmetric. Eqs. (4.22) and (4.23) respectively reproduce the first and second cases in eq. (4.2).

## 4.4 Edge cases

There is one last fine print that we need to take care of. The issue is that for some cases close to the bound (2.12)

$$\sum_i w_i \geq 2 \max_i (w_i) - 2 , \tag{4.24}$$

which we call 'the edge' in the following, some of the singularities we just encountered are actually absent.

**Two units from the edge** The first case in which something interesting happens is for

$$\sum_i w_i = 2 \max_i (w_i) . \tag{4.25}$$

In this case, $X_\emptyset$ becomes trivial (i.e. it does not have any zeros in $z$ except for possibly $z = 0$, $z = 1$ and $z = \infty$) and thus the codimension 1 singularity with $X_\emptyset = 0$ and the codimension 5 singularity that we discussed above are absent. Consequently, the list of singularities of the spacetime correlator is modified to

$$\sum_i j_i = n(k-2) + m + 3 , \tag{4.26}$$

together with all the reflected images. This reproduces the third possibility in eq. (4.2).

**One unit from the edge**  Moving closer to the edge, we look at the case of

$$\sum_i w_i = 2\max_i(w_i) - 1 \ . \tag{4.27}$$

In this case the parity is necessarily odd. The analysis of the codimension 2 singularity is unchanged. However, since some of the $X_I$ with $|I| = 1$ are $y_i$-independent, the corresponding codimension 4 singularity does not exist. This means that these poles are absent in the spacetime correlator and correspondingly all the poles that would be related by reflection symmetry to these poles in the correlator are also absent.[13] We thus conclude that the set of poles in this case reads

$$\sum_i j_i = \left( n + \frac{1}{2} \right)(k-2) + m + 3 \ , \tag{4.28}$$

together with all the reflected images. We hence obtain the third possibility in eq. (4.2).

**On the edge**  Finally, we look at the most extreme possibility, i.e.

$$\sum_i w_i = 2\max_i(w_i) - 2 \ . \tag{4.29}$$

In this case, the formula (2.13a) needs to be interpreted correctly. Indeed, $X_\emptyset = 0$ in this case and thus the formula is seemingly nonsensical. It was explained in [26] how to understand the formula. Essentially, $X_\emptyset = 0$ forces the generalized crossratio of the unflowed correlator to be pinned at $c = z$. One then needs to pick the correct critical behaviour of eq. (B.15f) to cancel the $X_\emptyset$ prefactor. In fact, for the edge case, the following formula was derived in [26]:

$$\left\langle V^{w_1}_{j_1,h_1,\bar{h}_1}(0;0) V^{w_2}_{j_2,h_2,\bar{h}_2}(1;1) V^{w_3}_{j_3,h_3,\bar{h}_3}(\infty;\infty) V^{w_4}_{j_4,h_4,\bar{h}_4}(x;z) \right\rangle \tag{4.30}$$

$$= \int \prod_{i=1}^{4} \frac{\mathrm{d}^2 y_i}{\pi} \, y_i^{\frac{kw_i}{2}+j_i-h_i-1} \bar{y}_i^{\frac{kw_i}{2}+j_i-\bar{h}_i-1} |X_{13}|^{2(-j_1+j_2-j_3+j_4)} |X_{23}|^{2(j_1-j_2-j_3+j_4)}$$

$$\times |X_{12}|^{4j_3-2k} |X_{34}|^{2(j_1+j_2+j_3-j_4-k)} |X_{1234}|^{2(-j_1-j_2-j_3-j_4+k)} f_{(1,1,1,1)}(z) \ , \tag{4.31}$$

where $f_{(1,1,1,1)}(z)$ is defined as the limit of the unflowed correlator as $x$ approaches $z$ as in eq. (B.15f).

One then performs exactly the same analysis as before. The codimension 1 singularity caused by $X_\emptyset = 0$ obviously does not exist since $X_\emptyset = 0$ generically. The other codimension 1 singularity $X_{1234} = 0$ however does exist and leads to the poles

$$\sum_i j_i = k + m + 1 \ . \tag{4.32}$$

---

[13]The absence of the reflected poles is not obvious since they are contained in the unflowed correlator. Thus they must be cancelled by a zero in the integration over $(y_i, z)$-space.

The analysis for the codimension 3 singularity on the other hand is completely analogous. Finally, the codimension 5 singularity also does not exist since again $X_\emptyset = 0$ generically. Thus the only singularities for the integral are those for

$$\sum_i j_i = m + 3 \ . \tag{4.33}$$

To analyse the singularities of the unflowed correlator, we can use eq. (B.15f). The Liouville correlator on the right hand side has a pole whenever

$$\sum_i j_i = (n+1)(k-2) + m + 3 \quad \text{or} \quad \sum_i j_i = -(n+1)(k-2) + 1 - m \ . \tag{4.34}$$

Additionally there are further zeros and poles contained in the prefactor $\gamma(j_1+j_2+j_3+j_4-k)$ that appears in eq. (B.15f). We get poles for

$$\sum_i j_i = k - m \tag{4.35}$$

and zeros for

$$\sum_i j_i = k + m + 1 \ . \tag{4.36}$$

We can now collect everything together. The zeros of the $\gamma$-factor cancel the poles of the codimension 1 singularity $X_{1234} = 0$. Thus we just have to add the poles to the Liouville correlator that come from the codimension 3 singularity and the poles from the $\gamma$-prefactor.

Hence, the actual poles of the spacetime correlator are located at

$$\sum_i j_i = (n+1)(k-2) + m + 3 \ , \tag{4.37}$$

together with all the reflected images.[14] This reproduces the last possibility in eq. (4.2) and hence putting everything together we obtain eqs. (4.1) and (4.2).

## 5 String correlators and spacetime CFT

In the previous section, we extensively analysed the location of spacetime singularities. In this section, we explore one of these poles in more detail and explain the relation to the spacetime CFT. The hard technical work is done at this point and the discussion in this section becomes physically much more interesting.

### 5.1 Residues at singularities

The simplest spacetime pole occurs for $m = n = 0$ in (4.1), which corresponds to

$$\sum_i j_i = 5 - k \ . \tag{5.1}$$

---

[14]There are again some poles that are cancelled by a zero in the integration. One can uniquely fix this answer by making use of reflection symmetry.

We recall that this singularity comes from the integration over $z$ and $y_i$. It is the codimension 5 singularity discussed above and hence the worldsheet integral fully localizes.

We also note that this condition is related via reflection symmetry to the condition $\sum_i j_i = k-1$, which is the special case analysed at the end of Section 3.2. Hence instead of recomputing the string correlator for this case, we use the formula (3.26) to obtain it directly. We saw that for $\sum_i j_i = k-1+\epsilon$, the correlation function behaves as $|z-z_\gamma|^{2(\epsilon-1)}$, where we recall that $z_\gamma$ is a solution of $P_{\boldsymbol{w}}(x; z_\gamma) = 0$ for fixed $x$. This means that for $\epsilon \to 0$, the integral over the cross ratio is divergent and the divergent pieces are localized at $z = z_\gamma$. In fact, we have the formula

$$\operatorname*{Res}_{\epsilon=0} |z - z_\gamma|^{2(\epsilon-1)} = \pi \delta^2(z - z_\gamma) \ . \tag{5.2}$$

Thus from eq. (3.26) we arrive at the following formula for the string correlator

$$\operatorname*{Res}_{\sum_i j_i = k-1} \left\langle\!\!\left\langle V_{j_1,h_1}^{w_1}(0)\, V_{j_2,h_2}^{w_2}(1)\, V_{j_3,h_3}^{w_3}(\infty)\, V_{j_4,h_4}^{w_4}(x) \right\rangle\!\!\right\rangle$$

$$= C_{\mathrm{S}^2} \frac{\nu^{2-k}\, \gamma(\tfrac{1}{2-k})^2}{2\pi(k-2)^4} \sum_\gamma |z_\gamma|^{-\frac{(k-2j_1-2)(k-2j_4-2)}{k-2}} |1-z_\gamma|^{-\frac{(k-2j_2-2)(k-2j_4-2)}{k-2}}$$

$$\times\ |\Pi|^{-k} \prod_{i=1}^{4} R(j_i, h_i, \bar{h}_i) w_i^{2j_i - \frac{k}{2}(w_i+1)} a_i^{\frac{k(w_i-1)}{4} - h_i} \bar{a}_i^{\frac{k(w_i-1)}{4} - \bar{h}_i} \ . \tag{5.3}$$

Here we denoted the string correlator by a double bracket. It is obtained by integrating the worldsheet correlator over the crossratio $z$ and inserting the prefactor $C_{\mathrm{S}^2}$ that accounts for the normalisation of the gravitational path integral on the sphere. See [48] for a standard discussion of this factor in the context of flat space string theory. The sum in (5.3) runs over all solutions of $P_{\boldsymbol{w}}(x; z_\gamma) = 0$ for fixed $x$. As before, the $a_i$'s denote the leading Taylor coefficients around the ramification points and $\Pi$ is the product of all residues of the relevant covering map, see eqs. (3.13) and (3.14).

Since the worldsheet theory is reflection symmetric, the same has to be true after integration over the cross ratio, i.e. for the string correlator. We may hence get the residue near $\sum_i j_i = 5-k$ by cancelling the reflection coefficients present in (5.3) and replacing spins by their reflected counterparts. See [25] for a discussion of reflection symmetry in the flowed sector. This leads to the simple final formula

$$\operatorname*{Res}_{\sum_i j_i = 5-k} \left\langle\!\!\left\langle V_{j_1,h_1}^{w_1}(0)\, V_{j_2,h_2}^{w_2}(1)\, V_{j_3,h_3}^{w_3}(\infty)\, V_{j_4,h_4}^{w_4}(x) \right\rangle\!\!\right\rangle$$

$$= C_{\mathrm{S}^2} \frac{\nu^{k-4}\, \gamma(\tfrac{1}{2-k})^2}{2\pi(k-2)^4} \sum_\gamma |z_\gamma|^{-\frac{(2j_1+k-4)(2j_4+k-4)}{k-2}} |1-z_\gamma|^{-\frac{(2j_2+k-4)(2j_4+k-4)}{k-2}}$$

$$\times\ |\Pi|^{-k} \prod_{i=1}^{4} w_i^{2-2j_i - \frac{k}{2}(w_i+1)} a_i^{\frac{k(w_i-1)}{4} - h_i} \bar{a}_i^{\frac{k(w_i-1)}{4} - \bar{h}_i} \ . \tag{5.4}$$

## 5.2 The spacetime CFT

The correlator (5.4) looks exactly like a correlation function of a symmetric orbifold CFT (see Appendix E for a short review) and can be understood directly from the spacetime

CFT. To explain this, let us first review in more detail the proposal of [27] for the dual spacetime CFT. We already mentioned the qualitative idea in the Introduction.

Let us consider bosonic string theory on $\mathrm{AdS}_3 \times X$, where $X$ stands for any compact unitary CFT with central charge $c(X) = 26 - \frac{3k}{k-2}$. This ensures that the total worldsheet theory will be critical. The construction of the dual CFT starts with the simple symmetric orbifold

$$\mathrm{Sym}^N(\mathbb{R}_Q \times X) \ . \tag{5.5}$$

The linear dilaton slope is chosen to be $Q = \frac{k-3}{\sqrt{k-2}}$,[15] so that the central charge of the 'seed' theory of the symmetric orbifold is the Brown-Henneaux central charge [49]

$$1 + 6Q^2 + c(X) = 6k \ . \tag{5.6}$$

A linear dilaton theory on its own does not define a consistent unitary CFT. However, one can promote this to a consistent unitary interacting CFT by turning on a particular non-normalisable marginal operator. This is very similar to the Coulomb gas construction of Liouville theory starting from a linear dilaton theory [50]. The marginal operator comes from the twist-2 sector and takes the form

$$\Phi = \mathrm{e}^{-\frac{\phi}{b}} \sigma_2 \ , \tag{5.7}$$

where $b = (k-2)^{-\frac{1}{2}}$, $\phi$ is the linear dilaton direction and $\sigma_2$ is the twist-2 field of the symmetric orbifold. We denote the coupling constant of this marginal operator by $\mu$. After adding this operator to the theory, the delta-functions from the momentum conservation of the linear dilaton theory get smeared out to poles. These poles reproduce the analytic structure of the string correlators that we discussed. Indeed, the momentum conservation of the linear dilaton theory predicts poles at $\ell$-th order in conformal perturbation theory for[16]

$$\sum_i p_i = Q + \frac{\ell}{2b} + mb \ , \qquad m \in \mathbb{Z}_{\geq 0} \ , \tag{5.8}$$

and similarly for any $p_i$ replaced by $Q - p_i$. This is similar to the situation for Liouville theory explained in Appendix A.5. A more complete discussion can also be found in [27]. The map between the linear dilaton momenta and the worldsheet spins takes the form

$$p_i = \frac{j_i + \frac{k}{2} - 2}{\sqrt{k-2}} \ , \tag{5.9}$$

so that the condition for singularities can be rewritten as

$$\sum_i j_i = \left(\frac{\ell}{2} - 1\right)(k-2) + m + 3 \ . \tag{5.10}$$

This agrees with (4.1) and (4.2) once we realize that not all orders $\ell$ of conformal perturbation theory can contribute to a given correlator. Indeed, correlation functions in the

---

[15]Note that $Q \neq Q_{\mathrm{L}} = \frac{k-1}{\sqrt{k-2}}$ that we used earlier.

[16]We use $p_i$ for the momenta of the dual CFT in order to avoid confusion with the Liouville momenta $\alpha_i$ that appeared earlier in the paper.

symmetric orbifold theory are computed by lifting them to appropriate covering spaces. The condition for a covering map to exist can be read off from Hurwitz' formula for the covering map. We have (for the case that both the base and the covering surface is a sphere)

$$2 = 2\text{N} - \sum_i (w_i - 1) - \ell \ , \tag{5.11}$$

where the additional $-\ell$ comes from the insertion of the $\ell$ marginal twist-2 operators and N is the degree of the covering map. From this we learn that

$$\ell = \sum_i w_i \bmod 2 \ . \tag{5.12}$$

Moreover, we also have by construction $\text{N} \geq w_i$ for all $i$, which leads to the constraint

$$\ell \geq 2 \max_i (w_i) + 2 - \sum_i w_i \ . \tag{5.13}$$

Given these two constraints, one then recovers (4.1) and (4.2). We are not aware of any simple way to see from the dual CFT that correlation functions with $\sum_i w_i < 2 \max_i (w_i) - 2$ have to vanish identically (as is the case for the string worldsheet theory).

For the sake of completeness, let us also explain the map of the remaining parameters. The worldsheet theory has generically only the string coupling constant $g_{\text{string}}$ as a further parameter. It is mapped to the combination

$$g_{\text{string}} \sim N^{-\frac{1}{2}} \mu^{\frac{k-3}{k-2}} \ , \tag{5.14}$$

up to irrelevant normalisation factors. The dual CFT does not depend separately on $N$ and $\mu$, but only on this combination, at least in a $\frac{1}{N}$ expansion.

We can finally compare our computation of the residue of the four-point function with the expectation from the spacetime CFT. As noticed already above, the condition $\sum_i j_i = 5 - k$ where the residue of the spacetime correlation function was easy to compute corresponds to the 0th-order in conformal perturbation theory, see eq. (5.10) with $\ell = 0$ and $m = 0$. Hence, the four-point function (5.4) is expected to directly reproduce a correlation function of the symmetric orbifold, up to a different normalisation of the vertex operators in the string description and the dual CFT. More precisely, we expect [27]

$$\operatorname*{Res}_{\sum_i j_i = 5-k} \left\langle\!\!\left\langle V_{j_1,h_1}^{w_1}(0)\, V_{j_2,h_2}^{w_2}(1)\, V_{j_3,h_3}^{w_3}(\infty)\, V_{j_4,h_4}^{w_4}(x) \right\rangle\!\!\right\rangle = \frac{1}{\pi N} \sum_\gamma |z_\gamma|^{-4p_1p_4}\, |1 - z_\gamma|^{-4p_2p_4}$$

$$\times\, |\Pi|^{-k} \prod_{i=1}^4 N(w_i, j_i)\, w_i^{-\frac{k}{2}(w_i+1)+\frac{1}{2}}\, a_i^{\frac{k(w_i-1)}{4} - h_i}\, \bar{a}_i^{\frac{k(w_i-1)}{4} - \bar{h}_i} \ , \tag{5.15}$$

where $N(w_i, j_i)$ accounts for the different normalisation of vertex operators in the bulk and in the boundary CFT. The relation between $j_i$ and $p_i$ is given in eq. (5.9). The right-hand side of this formula is simply the answer one would get for the connected contribution in the

symmetric orbifold.[17] For the benefit of the reader, we have summarised some facts about correlation functions of symmetric orbifold CFTs in Appendix E. One can now verify from eq. (5.4) that we indeed find a full match between the string and the dual CFT prediction, with the identification of parameters

$$\frac{C_{\mathrm{S}^2}}{\prod_{i=1}^4 N(w_i, j_i)} = \frac{\pi\,(k-2)^4\,\nu^{4-k}\,\gamma(\frac{k-1}{k-2})^2}{2N}\prod_{i=1}^4 w_i^{2j_i-\frac{3}{2}}\;. \tag{5.16}$$

This is a very strong check on the proposal, as it amounts to matching a four-point function *exactly* in $\alpha'$, something that seems impossible in other instances of the AdS/CFT correspondence.

The holographic match of three-point functions performed in [27] implies

$$\frac{C_{\mathrm{S}^2}}{\prod_{i=1}^3 N(w_i, j_i)} = \frac{2\pi(k-2)^2\,\nu^{2-\frac{k}{2}}\,\gamma(\frac{k-1}{k-2})}{\sqrt{N}}\prod_{i=1}^3 w_i^{2j_i-\frac{3}{2}} \tag{5.17}$$

and allows to solve for the normalisation parameters[18]

$$C_{\mathrm{S}^2} = \frac{128\,\pi\,N\,\nu^{k-4}}{(k-2)^4\,\gamma(\frac{k-1}{k-2})^2}\;, \qquad \text{and} \qquad N(w_i, j_i) = \frac{4\sqrt{N}\,\nu^{\frac{k}{2}-2}\,w_i^{\frac{3}{2}-2j_i}}{(k-2)^2\,\gamma(\frac{k-1}{k-2})}\;. \tag{5.18}$$

We should also note that the four-point function (5.4) has no LSZ-type poles (i.e. poles of the form $h_i - \frac{kw_i}{2} - j_i \in \mathbb{Z}_{\geq 0}$ or $h_i - \frac{kw_i}{2} + j_i \in \mathbb{Z}_{\leq 0}$). Hence following the prescription described in Section 2.3, this correlation function is only non-vanishing if all external operators are continuous representations. From a dual CFT point of view, this is expected – 0th order conformal perturbation theory cannot capture the discrete representations that arise as bound states. However, they become visible at higher orders in conformal perturbation theory. For example, it follows from (5.10) that the string correlator (5.3) could be obtained at 4th order in conformal perturbation theory in the dual CFT (although we did not check this explicitly). It indeed has all the LSZ poles and is hence non-vanishing with any number of external discrete representations. This explains the emergence of discrete representations as bound states in the full interacting dual CFT.

## 5.3 The tensionless limit

While the stringy correlation function (5.4) only describes a very special case in the generic duality, it is actually essentially the full answer in the tensionless limit. The tensionless

---

[17]Connected means here that the covering map that appears in the symmetric orbifold is connected. It does not mean that this correlator is connected in usual sense of the QFT. Indeed, as was pointed out in [51], the worldsheet four-point function does not have right cluster properties for a connected correlator, because the spacetime identity can appear as an intermediate state in the four-point function. For this reason it was argued in [51] that the dual CFT should be interpreted as a grand canonical ensemble. This does however not play an important role for the correlation function we consider.

[18]This is a bit different from the values given in [27]. The proposed values for these constants in [27] were based on a matching of the two- and three-point functions. Since the normalisation of two-point functions in string theory is very subtle [17, 52] it is better to use the matching of the four-point function. The values given in [27] correspond to a non-standard normalisation of the moduli space integral.

limit denotes the string background $\mathrm{AdS}_3 \times \mathrm{S}^3 \times \mathbb{T}^4$ of type IIB superstrings with one unit of NS-NS flux.

If we describe the superstring in the RNS-formalism, the tensionless limit corresponds to $k = 3$ in the above formulae. Moreover, it was shown in [34] that in this limit only $\mathfrak{sl}(2, \mathbb{R})_k$-representations with spin $j_i = \frac{1}{2}$ survive, due to the appearance of extra null-vector constraints on the worldsheet. This seems to spell doom on the spacetime correlators, since

$$\sum_i j_i = 2 = 5 - k = k - 1 \tag{5.19}$$

is always satisfied and thus any correlation function would sit on the pole in (5.4). What was instead shown in various papers [23, 35, 37, 39] is that our formula (2.13) is modified drastically. Indeed, (2.13) was derived by solving various consistency conditions on the worldsheet that come all in the form of differential equations. One can however note that when $\sum_i j_i = k - 1$, there is another solution to these constraints. It is rigid in that it cannot be deformed away to a situation where this condition is not satisfied. It is obtained by replacing the prefactors $|X_\emptyset|^{2(j_1+j_2+j_3+j_4-k)} = |X_\emptyset|^{-2}$ in (2.13) by $\pi \delta^2(X_\emptyset)$.[19] Indeed, the two distributions $|z|^{-2}$ and $\pi \delta^2(z)$ satisfy the same differential equations $z \partial_z f(z, \bar{z}) = -f(z, \bar{z})$, which guarantees that we indeed get a valid solution of all the constraints. It is clear that this distributional solution is incorrect for the full bosonic $\mathrm{SL}(2, \mathbb{R})_k$ WZW model, since it can only be defined for $\sum_i j_i = k-1$ and not be deformed continuously away from this point. However, it is the correct solution for the tensionless string [37]. While before we could expand around the points in moduli space where $X_\emptyset = 0$, the solution is now pinned to these points and thus the expansion becomes exact. We have in particular the exact formula analogous to (3.26)

$$\left\langle V_{j_1,h_1}^{w_1}(0;0) V_{j_2,h_2}^{w_2}(1;1) V_{j_3,h_3}^{w_3}(\infty;\infty) V_{j_4,h_4}^{w_4}(x;z) \right\rangle^{(2)}$$
$$= \frac{\gamma(\frac{1}{2-k})^2 \nu^{2-k}}{2\pi(k-2)^4} \sum_\gamma \delta^2(z - z_\gamma) |z_\gamma|^{-\frac{(2j_1+2-k)(2j_4+2-k)}{k-2}} |1 - z_\gamma|^{-\frac{(2j_2+2-k)(2j_4+2-k)}{k-2}}$$
$$\times |\Pi|^{-k} \prod_{i=1}^4 R(j_i, h_i, \bar{h}_i) w_i^{2j_i - \frac{k}{2}(w_i+1)} a_i^{\frac{k(w_i-1)}{4} - h_i} \bar{a}_i^{\frac{k(w_i-1)}{4} - \bar{h}_i}. \tag{5.20}$$

The superscript (2) is supposed to remind us that this corresponds to the alternative $\delta$-function solution of the constraints on the worldsheet. The integral over moduli space is now trivial. It just removes the $\delta$-function prefactor. Contrary to the generic solution, this alternative solution hence leads to a finite result when $\sum_i j_i = k - 1$. One then sees from (5.20) that a string worldsheet model based on this alternative solution of the worldsheet constraints will be dual to an ordinary symmetric orbifold without the need to add a deformation.

This is exactly what happens for the tensionless limit. To make this a bit more quantitative, it is convenient to use the hybrid formalism that is based on a $\mathfrak{psu}(1,1|2)_k$

---
[19]We include a factor of $\pi$ because $\mathrm{Res}_{\epsilon=0} |z|^{2(\epsilon-1)} = \pi \delta^2(z)$ and thus this is arguably the natural normalisation.

current algebra. In particular, for the tensionless limit we have $k = 1$ and there is a $\mathfrak{sl}(2,\mathbb{R})_1 \subset \mathfrak{psu}(1,1|2)_1$ conformal subalgebra. Using the hybrid formalism has the advantage that it makes picture-changing operators simpler. Indeed it was explained in [37] that picture changing is achieved by shifting some of the spins down by one unit, i.e. $j_i \to j_i - 1$, so that the condition $\sum_i j_i = k - 1$ is preserved. Thus the hybrid formalism predicts that we should compute the correlation function with $j_1 = \frac{1}{2}$, $j_2 = \frac{1}{2}$, $j_3 = -\frac{1}{2}$, $j_4 = -\frac{1}{2}$ in the $\mathrm{SL}(2,\mathbb{R})_1$ WZW model in order to reproduce the symmetric orbifold correlators. Physical state conditions of the hybrid formalism are quite complicated and are discussed in [41, 43, 53–55]. There are in particular a couple of technicalities that we are neglecting here. They lead to overall constants which is why we will omit constant prefactors in the following formulae. Thus, the following discussion will be a bit more sketchy than the rest of the paper. We should also note that these formulae are strictly speaking only correct for odd $w_i$, since otherwise the $\mathrm{SU}(2)_1$ WZW model sits in the spinor representation and gives additional contributions to the correlator. With these qualifications in mind, we simply get

$$\left\langle\!\!\left\langle V_{h_1}^{w_1}(0)V_{h_2}^{w_2}(1)V_{h_3}^{w_3}(\infty)V_{h_4}^{w_4}(x)\right\rangle\!\!\right\rangle^{(2)} \sim \sum_\gamma |\Pi|^{-1}\prod_{i=1}^4 a_i^{\frac{w_i-1}{4}-h_i}\bar{a}_i^{\frac{w_i-1}{4}-\bar{h}_i} \qquad (5.21)$$

for a string correlator in the tensionless limit of the hybrid formalism. We omitted the spin labels because they are no longer free. In principle, the conformal weights $h_i$ are of course determined by the mass-shell condition. We could however imagine to dress up the vertex operators $V_{h_i}^{w_i}$ with a primary vertex operator of the internal CFT. Comparing with eq. (E.3), we see that this agrees with the formula for the correlation functions of twist fields of a symmetric orbifold with central charge 6 (up to an overall normalisation and the relative normalisation of vertex operators that we did not analyse). In particular, our analysis improves on the previous results [24, 37] in that it fixes the full coordinate dependence of the correlation functions up to a single overall constant.

## 6 Conclusions, discussion and open questions

In this paper, we analysed string correlation functions on $\mathrm{AdS}_3$ with pure NS-NS flux in detail. We focused on genus 0 four-point functions, where we showed that string correlators are effectively calculable. In particular, we determined the analytic structure of both the worldsheet correlators and the spacetime correlators. We then compared our results to the predictions of the proposed dual CFT (1.1) and found perfect agreement in all cases.

### 6.1 Discussion

It is interesting that this instance of the AdS/CFT correspondence is under such good control that various (non-protected) quantities can be matched non-perturbatively in $\alpha'$. The duality gives two completely different descriptions of string theory on $\mathrm{AdS}_3$. At this point, the two descriptions have very different strengths and we can use one to learn more about the other and vice-versa. We now discuss the complementary strengths of the two descriptions.

The worldsheet description is technically and conceptually much more complicated than the description in terms of the deformed symmetric product orbifold, since it involves the additional data of worldsheet moduli and $\mathrm{SL}(2, \mathbb{R})$-spins that are in the end integrated over and fixed by the mass-shell condition, respectively. The representation theory of the affine $\mathfrak{sl}(2, \mathbb{R})_k$ algebra is also quite complicated to deal with. Hence the worldsheet description typically yields much more involved intermediate results which often simplify in the end for physically interesting computations such as the ones described in this paper. We should also stress that the approach that we advocated in this paper and in [25, 26] is completely algebraic. We systematically exploited various constraints for the worldsheet theory to derive the formula for the worldsheet four-point function eq. (2.13) and the appearance of geometric data in it remained mysterious. The worldsheet gives a formula for the string four-point function that is in principle valid for all choices of external parameters. Thus, it allows one to prove various formal properties of the correlation functions such as their reflection symmetry or the analyticity properties of the correlators as a function of the spins. It is moreover simple to derive bounds on the spectral flow that ensure that the string correlator is non-vanishing. These are called spectral flow violation in previous literature and are essentially the fusion rules of the model.

The dual CFT description on the other hand is conceptually much easier since it boils down to computing integrated correlators in a symmetric orbifold theory at a given order in conformal perturbation theory. However, conformal perturbation theory only gives one access to the poles of spacetime correlators and there is no known way to define directly the correlation functions for any choice of momenta. This problem is similar to the definition of Liouville theory. The dual CFT description is also entirely geometrical and can be formulated in terms of ramified covering maps. Matching of the two formalism thus implies a number of surprising identities between the worldsheet quantities and covering map quantities (such as the ones listed in Appendix C). There also does not seem to be a simple way to derive the reflection symmetry or the fusion rules of correlation functions in the symmetric orbifold description. Overall, the dual CFT description is hence usually simpler but more restrictive. It also remains a hard challenge to even define the dual CFT properly.

## 6.2 Open questions

Let us mention some further observations and possible future research directions.

**Higher points and genera**  Our results are generalisable in a variety of ways. We exclusively dealt with tree-level four-point functions of affine primary vertex operators of the bosonic string. We expect that these restrictions can be loosened as follows. There surely exist higher point and higher-genus versions of our formula for the four-point function (2.13). From a dual CFT point of view, it is trivial to generalise the prediction for the corresponding residue of the string correlator (5.4) to any number of points. For an $n$-point function, the analogous condition reads $\sum_i j_i = n - 1 - \frac{n-2}{2}(k-2)$ and the formula for (5.4) is essentially identical, except that the four-point free boson correlator is replaced by an $n$-point free-bosonc correlator and the sum runs over covering maps that are appropriately

ramified over $n$ points. This simplicity should be mirrored by a corresponding simplicity on the worldsheet. Essentially the same is true for higher genus correlators from the dual CFT point of view, where the free boson correlator in (5.4) has to be replaced by the corresponding higher genus free boson correlator.

**Multi-particle states**   A conceptually interesting generalization concerns multi-particle states. In the dual CFT, we also have twist operators of the form $\sigma_{w_1, w_2, \ldots, w_n}$, where $w_1 \geq w_2 \geq \ldots$ labels a partition and hence a conjugacy class of the symmetric group. In this paper we only discussed correlation functions of single-particle states that correspond to single-cycle twist-fields. In the symmetric orbifold and its deformation, we can easily include multi-twist fields as well and one still has to sum over covering maps with the appropriate branch profile. This was considered e.g. in [56, 57]. However, on the string side this generalisation is non-trivial. Since such a multi-cycle twist operator can be obtained by a (non-singular) limit where several single-cycle twist fields approach, the same should be true on the worldsheet. This would lead to multi-local vertex operators of the form $V_{j_1, \ldots, j_n, h}^{w_1, \ldots, w_n}(x; z_1, \ldots, z_n)$ that are dual to multi-cycle twist fields. We hence see that a complete match with the dual CFT necessitates the inclusion of multi-local vertex operators in the worldsheet CFT. This would constitute a further breakdown of locality of the worldsheet theory beyond the peculiar singularity structure discussed in this paper.

**Other backgrounds**   Another interesting direction for further exploration is the consideration of different locally AdS backgrounds, such as the Euclidean BTZ black hole. These backgrounds can be constructed on the worldsheet as an orbifold CFT of the $\mathrm{SL}(2, \mathbb{R})_k$ model, as explained in [35, 58–60]. One then expects string correlation functions to match correlation functions of the dual CFT on the respective conformal boundary of the locally AdS space.

**More systematic treatment**   We have directly checked that the residue of the string correlator matches the dual CFT correlator in a special case. Our derivation uses many non-trivial identities listed in Appendix C that were surprising from our point of view and we could only check numerically. It is thus clear that our methods, while powerful, are at the moment not able to lead to a clear conceptual understanding of the underlying mathematics of the matching of the string worldsheet with the dual CFT. It would thus be very desirable to develop a more systematic understanding.

**Relation with the $H_3^+$/Liouville correspondence**   The duality analysed in this paper relates string correlation functions of the (analytically continued) $\mathrm{SL}(2, \mathbb{R})_k$ WZW model to correlation functions of a deformed symmetric orbifold. There is a similar relation of correlation functions to Liouville correlators known as the $H_3^+$/Liouville correspondence [19–21, 61, 62].[20] It is somewhat puzzling that this relation did not play a role in our analysis. It would be very interesting to understand the relation between these two statements.

---

[20]There is also yet another curious relation between the four-point function of the $H_3^+$ and a special Liouville five-point function that is conceptually very mysterious to us [63, 64].

**Spacetime OPE**   We have studied the worldsheet theory of string theory on AdS$_3$ and its conjectured dual CFT in detail. This setup is ideally suited to study various structural aspects of the dual CFT. Most importantly in our opinion, the dual CFT has an operator product expansion, whose appearance is very non-trivial from a worldsheet point of view. It suggests that there should be a corresponding OPE on the worldsheet of the schematic form (suppressing possible structure constants etc.)[21]

$$V_{j_1,h_1}^{w_1}(x_1;z_1)V_{j_2,h_2}^{w_2}(x_2;z_2) \overset{x_1 \to x_2}{\sim} \sum_j V_{j,h_1+h_2}^{w_1,w_2}(x_2;z_1,z_2) + \sum_{j,h,w} V_{j,h}^w(x_2;z_2) + \dots , \quad (6.1)$$

where terms in the dots are subleading when $x_1 \to x_2$. This spacetime OPE should contain both bi-local and local vertex operators because both single- and double-twist operators would appear in the dual CFT. The emergence of a spacetime OPE from the worldsheet OPE was previously discussed in [12, 65, 66].

**SL(2, ℝ) Chern-Simons theory**   While our motivation for studying AdS$_3$ string theory was mainly drawn from a desire to understand string theory on this background and the associated AdS/CFT correspondence, we should also mention that the model is very interesting from the point of view of Chern-Simons theory. Contrary to the compact case, there is no known relationship between non-compact WZW models and Chern-Simons theories in three dimensions. It is not even known in principle how to quantise the latter. However, one may hope that a study of the SL(2, ℝ) WZW model gives a useful hint how to proceed for SL(2, ℝ) Chern-Simons theory.

## Acknowledgments

We would like to thank Nick Agia, Davide Bufalini, Matthias Gaberdiel, Indranil Halder, Daniel Jafferis, David Kolchmeyer, Juan Maldacena, Emil Martinec, Sylvain Ribault, Alessandro Sfondrini, Cumrun Vafa, Edward Witten and Xi Yin for stimulating discussions. We are very grateful to Sylvain Ribault for useful correspondence and for his comments on an early draft of this paper. The work of A.D. is funded by the Swiss National Science Foundation via the Early Postdoc.Mobility fellowship. L.E. gratefully acknowledges support from the grant DE-SC0009988 from the U.S. Department of Energy.

## A    Limits of the unflowed four-point function

In this appendix, we derive various technical statements about the relation of limits of the unflowed correlation functions and Liouville correlation functions. These are closely related to the $H_3^+$/Liouville correspondence [20, 21]. But since they are not difficult to derive independently and in order to make sure that overall constants are correctly normalised in our conventions, we will rederive these identities. Our discussion is inspired by [67]. Earlier works analysing similar identities include [68, 69].

---

[21]Potentially, further BRST-exact terms might appear on the right-hand side that disappear after the moduli space integral.

### A.1  Conformal blocks

Let us start by reviewing some standard properties of conformal blocks for the unflowed sector of the $\mathrm{SL}(2,\mathbb{R})_k$ WZW model (or rather its analytic continuation, the $H_3^+$ model). Conformal blocks are special solutions of the KZ-equation, which takes the form [11]

$$\partial_z \mathcal{F}_j(x,z) = \frac{1}{k-2}\left(\frac{\mathcal{P}}{z} + \frac{\mathcal{Q}}{z-1}\right)\mathcal{F}_j(x,z)\,, \tag{A.1}$$

where $\mathcal{P}$ and $\mathcal{Q}$ are differential operators with respect to $x$,

$$\begin{aligned}
\mathcal{P} &= x^2(x-1)\frac{\partial^2}{\partial x^2} - \left((\kappa-1)\,x^2 + 2\,j_1\,x - 2\,j_4\,x\,(x-1)\right)\frac{\partial}{\partial x}\\
&\quad - 2\,\kappa\,j_4\,x - 2\,j_1\,j_4\,, 
\end{aligned} \tag{A.2}$$

$$\begin{aligned}
\mathcal{Q} &= -(1-x)^2 x\frac{\partial^2}{\partial x^2} - 2\,\kappa\,j_4\,(1-x) - 2\,j_2\,j_4\\
&\quad + \left((\kappa-1)(1-x)^2 + 2\,j_2\,(1-x) - 2\,j_4\,x\,(x-1)\right)\frac{\partial}{\partial x}\,, 
\end{aligned} \tag{A.3}$$

and

$$\kappa \equiv j_3 - j_1 - j_2 - j_4\,. \tag{A.4}$$

Here $j$ labels different solutions of the KZ-equation. We sometimes write $\mathcal{F}_j^{j_1,j_2,j_3,j_4}(x,z)$ to emphasize the dependence on the other spins. Correlators of continuous representations are given by a conformal block expansion (in the s-channel)

$$\langle V_{j_1}^0(0;0)V_{j_2}^0(1;1)V_{j_3}^0(\infty;\infty)V_{j_4}^0(x;z)\rangle = \int_{\frac{1}{2}+i\mathbb{R}}\mathrm{d}j\,\frac{D(j_1,j_4,j)D(j_2,j_3,j)}{B(j)}\mathcal{F}_j(x,z)\mathcal{F}_j(\bar{x},\bar{z})\,. \tag{A.5}$$

It is convenient to expand s-channel conformal blocks in a small parameter. In the present context, there are several natural choices.

**Small $z$ expansion**  For small values of $z$, we expand conformal blocks as follows,

$$\mathcal{F}_j(x,z) = z^{\Delta(j)-\Delta(j_1)-\Delta(j_4)}x^{j-j_1-j_4}\sum_{n=0}^{\infty}f_{j,n}(x)z^n\,. \tag{A.6}$$

Plugging this ansatz into the KZ-equation leads to two linearly independent solutions for $f_{j,0}(x)$:

$$f_{j,0}(x) = {}_2F_1(j+j_2-j_3, j-j_1+j_4; 2j; x)\,, \tag{A.7a}$$

$$f_{j,0}(x) = x^{1-2j}\,{}_2F_1(1-j+j_2-j_3, 1-j-j_1+j_4; 2-2j; x)\,. \tag{A.7b}$$

From these initial conditions, one can determine the solution for $f_{j,n}(x)$ recursively [11]. This procedure is unique if one assumes that the leading power of $f_{j,n}(x)$ in $x$ is integer, as is appropriate for a conformal block. The two solutions in eq. (A.7) are related by reflection symmetry $j \to 1-j$. Hence, it suffices to keep one of the solutions, but later integrate over all $j \in \frac{1}{2} + i\mathbb{R}$ in the conformal block expansion (A.5). Thus, conformal blocks are normalised by picking the first solution with unit coefficient.

**Small $x$ expansion**   Another useful expansion is an expansion in small $x$, where we can write

$$\mathcal{F}_j(x,z) = x^{\Delta(j)-\Delta(j_1)-\Delta(j_4)+j-j_1-j_4} \left(\frac{z}{x}\right)^{\Delta(j)-\Delta(j_1)-\Delta(j_4)} \sum_{n=0}^{\infty} g_{j,n} \left(\frac{z}{x}\right) x^n \ . \tag{A.8}$$

The leading order term takes again a simple form

$$g_{j,0}(u) = {}_2F_1 \left( j_1 + j_4 - j, j_2 + j_3 - j; k - 2j; u \right) \ . \tag{A.9}$$

We chose the normalisation to agree with the small $z$-expansion.

## A.2   Symmetries of the conformal block

The conformal blocks satisfy several highly non-obvious symmetries that follow from the existence of the degenerate field with $j = \frac{k}{2}$. In [26, 70], it was shown that one can swap an even number of spins $j_i \to \frac{k}{2} - j_i$ in the correlator and leave the result unchanged. For example, we have

$$\langle V_{j_1}^0(0;0) V_{j_2}^0(1;1) V_{j_3}^0(\infty;\infty) V_{j_4}^0(x;z) \rangle = \mathcal{N}(j_1)\mathcal{N}(j_2)\mathcal{N}(j_3)\mathcal{N}(j_4)$$

$$\times \left| x^{-j_1+j_2+j_3-j_4}(1-x)^{j_1-j_2+j_3-j_4} z^{j_1+j_4-\frac{k}{2}}(1-z)^{j_2+j_4-\frac{k}{2}}(x-z)^{k-j_1-j_2-j_3-j_4} \right|^2$$

$$\times \left\langle V_{\frac{k}{2}-j_1}^0(0;0) V_{\frac{k}{2}-j_2}^0(1;1) V_{\frac{k}{2}-j_3}^0(\infty;\infty) V_{\frac{k}{2}-j_4}^0(x;z) \right\rangle \ . \tag{A.10}$$

Apart from the $x$- and $z$-prefactors, the rule is simply that for every swapped spin $j_i$ we have to include a prefactor $\mathcal{N}(j_i)$ (that is defined in (2.4)). Upon combining this with the conformal block expansion (A.5) and using that

$$\mathcal{N}(j_1)\mathcal{N}(j_4)D(\tfrac{k}{2}-j_1, \tfrac{k}{2}-j_4, j) = D(j_1, j_4, j) \ , \tag{A.11}$$

we learn that the conformal block satisfies the identity

$$\mathcal{F}_j^{j_1,j_2,j_3,j_4}(x,z) = x^{-j_1+j_2+j_3-j_4}(1-x)^{j_1-j_2+j_3-j_4} z^{j_1+j_4-\frac{k}{2}}(1-z)^{j_2+j_4-\frac{k}{2}}$$

$$\times (x-z)^{k-j_1-j_2-j_3-j_4} \mathcal{F}_j^{\frac{k}{2}-j_1,\frac{k}{2}-j_2,\frac{k}{2}-j_3,\frac{k}{2}-j_4}(x,z) \ . \tag{A.12}$$

There are similar identities that only swap some of the spins. To make the present discussion more readable, we collect them in Appendix B.2.

## A.3   Singular behaviour for $x \to z$

From eq. (A.12) we see that the conformal blocks are singular as $x \to z$. The two possible behaviours are

$$\mathcal{F}_j(x,z) \sim f_j^{(1)}(z) + \cdots \ , \tag{A.13}$$

$$\mathcal{F}_j(x,z) \sim (x-z)^{k-j_1-j_2-j_3-j_4} f_j^{(2)}(z) + \cdots \ , \tag{A.14}$$

where the dots denote higher powers in $(x-z)$. The claim of [67] is that for $x \to z$, the conformal blocks (A.13) and (A.14) are related to usual Virasoro conformal blocks.

Let us focus first on the regular possibility. From the small $x$-expansion, we find that

$$f_j^{(1)}(z) = \frac{\Gamma(k-2j)\Gamma(k-j_1-j_2-j_3-j_4)}{\Gamma(k-j_1-j_4-j)\Gamma(k-j_2-j_3-j)} z^{\Delta(j)+j-\Delta(j_1)-j_1-\Delta(j_4)-j_4} + \cdots , \quad \text{(A.15)}$$

where the dots denote higher orders in $z$. Let us define

$$b^2 = \frac{1}{k-2} . \quad \text{(A.16)}$$

Then we note that

$$\Delta(j) + j = \alpha(Q_{\mathrm{L}} - \alpha) , \quad \text{(A.17)}$$

where $\alpha = bj$ and $Q_{\mathrm{L}} = b + b^{-1}$, as usual in the parametrization of Virasoro conformal blocks. This motivates the identification

$$f_j^{(1)}(z) = \frac{\Gamma(k-2j)\Gamma(k-j_1-j_2-j_3-j_4)}{\Gamma(k-j_1-j_4-j)\Gamma(k-j_2-j_3-j)} \mathcal{F}_{\alpha=bj}^{\mathrm{Vir}}(\alpha_i = bj_i; z) \quad \text{(A.18)}$$

with a Virasoro conformal block, since the two have the same small $z$ expansion. Based on monodromy considerations, this identity was proven in [67]. The other singular behaviour can be readily deduced by using the symmetry (A.12)

$$f_j^{(2)}(z) = z^{j_2+j_3-\frac{k}{2}}(1-z)^{j_1+j_3-\frac{k}{2}} \frac{\Gamma(k-2j)\Gamma(j_1+j_2+j_3+j_4-k)}{\Gamma(j_1+j_4-j)\Gamma(j_2+j_3-j)}$$
$$\times \mathcal{F}_{\alpha=bj}^{\mathrm{Vir}}\left(\alpha_i = b\left(\tfrac{k}{2}-j_i\right); z\right) . \quad \text{(A.19)}$$

## A.4 Review of the DOZZ formula

Before continuing, let us comment on the relation of the $\mathrm{SL}(2,\mathbb{R})$ structure constants and the DOZZ formula for Liouville [71, 72]. The two are very closely related. The DOZZ formula reads

$$D_{\mathrm{L}}(\alpha_1, \alpha_2, \alpha_3) = \langle V_{\alpha_1}(0)V_{\alpha_2}(1)V_{\alpha_3}(\infty)\rangle_{\mathrm{L}} = \left(\nu\, b^{2-2b^2}\right)^{b^{-1}(Q_{\mathrm{L}}-\sum_i \alpha_i)}$$
$$\times \frac{\Upsilon_0 \Upsilon(2\alpha_1)\Upsilon(2\alpha_2)\Upsilon(2\alpha_3)}{\Upsilon(\alpha_1+\alpha_2+\alpha_3-Q_{\mathrm{L}})\Upsilon(\alpha_1+\alpha_2-\alpha_3)\Upsilon(\alpha_2+\alpha_3-\alpha_1)\Upsilon(\alpha_1+\alpha_3-\alpha_2)} . \quad \text{(A.20)}$$

We defined the cosmological constant $\mu$ slightly differently to what is customary. We have

$$\nu = \pi\,\mu_{\mathrm{DOZZ}}\,\gamma(b^2) , \quad \text{(A.21)}$$

where $\mu_{\mathrm{DOZZ}}$ is the value that follows naturally from the path integral. We also have $\Upsilon_0 = \frac{\mathrm{d}}{\mathrm{d}x}\big|_{x=0}\Upsilon(x)$. $\Upsilon(x) \equiv \Upsilon_b(x)$ is very closely related to the Barnes double Gamma function that appears in the three-point function of the $H_3^+$ model. We have

$$\mathrm{G}_k(j) = \frac{b^{-b^2 j(j+1+b^{-2})}}{\Upsilon(-bj)} \quad \text{(A.22)}$$

and hence

$$D(j_1, j_2, j_3) = -\frac{b\,\nu^{-b^{-2}}\gamma(-b^2)\gamma(k-j_1-j_2-j_3)}{2\pi^2 \prod_{i=1}^3 \gamma\left(\frac{2j_i-1}{k-2}\right)} D_{\mathrm{L}}(bj_1, bj_2, bj_3) , \quad \text{(A.23)}$$

where $b^{-2} = k - 2$. This implies that

$$\frac{D(j_1, j_4, j)D(j_2, j_3, j)}{B(j)} = D_{\mathrm{L}}(bj_1, bj_4, bj)\, D_{\mathrm{L}}(bj_2, bj_3, Q_{\mathrm{L}} - bj)$$

$$\times \frac{b^2\, \nu^{2-k}\, \gamma(-b^2)^2\, \gamma(k - j - j_1 - j_4)\, \gamma(k - j - j_2 - j_3)}{4\pi^3 \gamma(k - 2j) \prod_{i=1}^{4} \gamma\left(\frac{2j_i - 1}{k - 2}\right)} \ . \quad \text{(A.24)}$$

## A.5 Bulk poles of Liouville theory

We also want to review the pole structure of Liouville correlators, since we use it several times in the present paper. The poles that we discuss are the so-called bulk-poles [31]. They are simplest to understand from the functional integral of Liouville theory:[22]

$$\left\langle \prod_i V_{\alpha_i}(z_i) \right\rangle_{\mathrm{L}}$$

$$= \int \mathcal{D}\phi \ \exp\left(-\frac{1}{4\pi} \int \mathrm{d}^2 z \sqrt{g}\, (g^{ab}\partial_a\phi\partial_b\phi + Q_{\mathrm{L}}R\,\phi + \nu\, \mathrm{e}^{2b\phi}) + \sum_i 2\alpha_i\phi(z_i)\right) \ . \quad \text{(A.25)}$$

There is a standard argument [73] (known as KPZ scaling) that determines the dependence of a correlation function on $\nu$. For this, we note that $\nu$ can be entirely removed from the path integral by shifting $\phi = \phi_0 - \frac{1}{2b}\log(\nu)$. Most terms on the right-hand side are invariant under this shift, but we get the overall factor

$$\nu^{\frac{1}{2b}(Q_{\mathrm{L}}\chi - 2\sum_i \alpha_i)} \ , \quad \text{(A.26)}$$

where $\chi$ is the Euler characteristic that enters via the Gauss-Bonnet theorem. This matches with the $\nu$-dependence that we get from the DOZZ formula (A.20).

The path integral also tells us the location of the poles. For this, one has to analyse the convergence of the path integral. For $\phi \to \infty$, the action is dominated by the term $\mathrm{e}^{2b\phi}$ and assuming $\nu > 0$, the action becomes very large in this regime, which leads to a converging answer. The dangerous regime is $\phi \to -\infty$. Assume that $\phi \sim \phi_0$, where $\phi_0$ is constant. Then the path integral over the constant mode becomes

$$\int \mathrm{d}\phi_0 \ \mathrm{e}^{-Q_{\mathrm{L}}\chi\phi_0 + \sum_i 2\alpha_i\phi_0} \ , \quad \text{(A.27)}$$

where we used again the Gauss-Bonnet theorem. This is only convergent for $\sum_i \mathrm{Re}(\alpha_i) > \frac{Q_{\mathrm{L}}\chi}{2}$. In particular, this condition is true for the operators $\alpha_i \in \frac{Q_{\mathrm{L}}}{2} + i\mathbb{R}$ that are part of the spectrum of the Liouville theory (except for the sphere two-point function and the torus partition function). Once we analytically continue Liouville correlators, we encounter a pole if this inequality is saturated, which can be seen by performing the integral over $\phi_0$ in the region $\phi_0 \to -\infty$ explicitly. However, this is not the full story. Since we can expand the term $\mathrm{e}^{2b\phi}$ in the action, we get subleading corrections

$$\int \mathrm{d}\phi_0 \ \mathrm{e}^{-Q_{\mathrm{L}}\chi\phi_0 + \sum_i 2\alpha_i\phi_0 + 2nb\phi_0} \ , \quad \text{(A.28)}$$

---

[22]Strictly speaking the coupling constant should read $\mu_{\mathrm{DOZZ}}$ instead of $\nu$. The two are related by (A.21) and this does not matter for the following discussion.

which suggests that the Liouville correlators have poles whenever

$$\sum_i \alpha_i = \frac{Q_{\rm L}\,\chi}{2} - nb \tag{A.29}$$

for $n \in \mathbb{Z}_{\geq 0}$. All other singularities may be found by invoking reflection symmetry and the duality symmetry $b \leftrightarrow b^{-1}$ in Liouville theory. This shows that Liouville correlators have (at least the) singularities

$$\sum_i \alpha_i = \frac{Q_{\rm L}\,\chi}{2} - nb - mb^{-1} \tag{A.30}$$

for $m, n \in \mathbb{Z}_{\geq 0}$, together with all reflected singularities that are obtained by replacing any number of $\alpha_i$'s by $Q_{\rm L} - \alpha_i$.

If we construct Liouville theory from the Coulomb gas formalism, we would treat $\mathrm{e}^{2b\phi}$ as a marginal operator that deforms the theory away from a linear dilaton theory. The 'simplest' singularity appears for $\sum_i \alpha_i = \frac{Q_{\rm L}\,\chi}{2}$. This condition corresponds exactly to the charge conservation of the undeformed theory and any insertion of the marginal operator violates charge conservation. For this reason, one has [72]

$$\operatorname*{Res}_{\sum_i \alpha_i = \frac{Q_{\rm L}\,\chi}{2}} \left\langle \prod_i V_{\alpha_i}(z_i) \right\rangle_{\rm L} = \left\langle \prod_{i=1}^n \mathrm{e}^{2\alpha_i z_i} \right\rangle'_{\text{linear dilaton}} = \prod_{i<j} |z_i - z_j|^{-4\alpha_i\alpha_j} \;, \tag{A.31}$$

where the prime means that we are omitting the momentum conserving delta-function in the linear dilaton theory and the last equality holds for sphere correlation functions. There could be in principle a universal normalisation constant that relates the Liouville correlators to the free boson correlators. This constant is 1 by construction of the DOZZ formula, which was first derived from the path integral. One can also check that this equality indeed holds for the DOZZ formula (where the right-hand side is equal to one for the three-point function). We will use this fact in the following.

## A.6   Singular behaviour of the correlation function

We are interested in the regular part of the limit

$$\lim_{x \to z} \left\langle V_{j_1}^0(0;0) V_{j_2}^0(1;1) V_{j_3}^0(\infty;\infty) V_{j_4}^0(x;z) \right\rangle \;. \tag{A.32}$$

This will in fact turn out to be useful in the main text. Let us assume that we are considering a four-point function of four vertex operator corresponding to continuous representations. Then the conformal block decomposition of the four-point function reads

$$\left\langle V_{j_1}^0(0;0) V_{j_2}^0(1;1) V_{j_3}^0(\infty;\infty) V_{j_4}^0(x;z) \right\rangle = \int_{\frac{1}{2}+i\mathbb{R}} \mathrm{d}j\, \mathcal{C}(j) \mathcal{F}_j(x,z) \mathcal{F}_j(\bar{x},\bar{z}) \;, \tag{A.33}$$

where we defined

$$\mathcal{C}(j) = \frac{D(j_1, j_4, j)\, D(j_2, j_3, j)}{B(j)} \tag{A.34}$$

with

$$B(j) \equiv \frac{k-2}{\pi} \frac{\nu^{1-2j}}{\gamma\left(\frac{2j-1}{k-2}\right)} \;. \tag{A.35}$$

As we already mentioned, the two solutions of the KZ equation (A.1) are related by $j \to 1-j$ and integrating over $j = \frac{1}{2} + is$ with $s \in \mathbb{R}$ amounts to including both solutions. Including both solutions is necessary in order for the four point function to be monodromy invariant [17]. In the following it will be useful to move the integration contour. In doing so, one should pay attention not to cross any poles. We now explain very carefully where all these pole are situated in the $j$-plane. The conclusion will however be that none of these poles matter for the final result, which in the limit $x \to z$ expresses the correlator of the $H_3^+$-model in terms of a Liouville theory correlator.

As we will see momentarily, the only relevant poles for us will lie in the region $\frac{1}{2} < \mathrm{Re}(j) < \frac{k-1}{2}$. The conformal blocks $\mathcal{F}_j(x,z)$ do not have any poles in this regime and thus we may focus on the poles originating from $\mathcal{C}(j)$. The three-point function $D(j_1, j_4, j)$ has poles whenever

$$J = n + m(k-2) \qquad \text{or} \qquad J = -(n+1) - (m+1)(k-2) \tag{A.36}$$

for $m, n \in \mathbb{Z}_{\geq 0}$, where $J$ is either of

$$1 - j_1 - j_4 - j , \qquad j_1 - j_4 - j , \qquad j_4 - j_1 - j , \qquad j - j_1 - j_4 . \tag{A.37}$$

Since $\mathrm{Re}(j_1) = \mathrm{Re}(j_4) = \frac{1}{2}$, it is easy to see that the only such poles for which $\frac{1}{2} < \mathrm{Re}(j) < \frac{k-1}{2}$ is the case

$$j = j_1 + j_4 + n \tag{A.38}$$

for $n \in \mathbb{Z}_{\geq 0}$ chosen such that $j < \frac{k-1}{2}$.

Thus, we can deform the contour of integration as follows:

$$\langle V_{j_1}^0(0;0) V_{j_2}^0(1;1) V_{j_3}^0(\infty;\infty) V_{j_4}^0(x;z) \rangle = \int_{\mathfrak{C}} dj \, \mathcal{C}(j) \mathcal{F}_j(x,z) \mathcal{F}_j(\bar{x}, \bar{z}) , \tag{A.39}$$

where the contour $\mathfrak{C}$ together with the dangerous poles is depicted in Figure 1. We deformed the contour such that the bulk of the contour runs on the axis $\mathrm{Re}(j) = \frac{k-1}{2}$, except for two arcs – one that encircles the poles that we have identified and the other encircles the region of the poles, but with $j \to k-1-j$. It is simple to see that this operation does not pick up further poles.

Equivalently, we can write

$$\langle V_{j_1}^0(0;0) V_{j_2}^0(1;1) V_{j_3}^0(\infty;\infty) V_{j_4}^0(x;z) \rangle$$
$$= \frac{1}{2} \int_{\mathfrak{C}} dj \Big( \mathcal{C}(j) \mathcal{F}_j(x,z) \mathcal{F}_j(\bar{x}, \bar{z}) + \mathcal{C}(k-1-j) \mathcal{F}_{k-1-j}(x,z) \mathcal{F}_{k-1-j}(\bar{x}, \bar{z}) \Big) , \tag{A.40}$$

where we used that the contour $\mathfrak{C}$ is invariant under $j \to k-1-j$. Using the functional equations of the Barnes G-function, one can show that [17]

$$\mathcal{C}(k-1-j) = -\frac{\gamma(k-2j)^2 \gamma(j_1 + j_4 + j - k + 1) \gamma(j_2 + j_3 + j - k + 1)}{(k-2j-1)^2 \gamma(j_1 + j_4 - j) \gamma(j_2 + j_3 - j)} \mathcal{C}(j) \tag{A.41}$$

so that in the $x \to z$ limit, making use of eqs. (A.13) and (A.18), we find

$$\lim_{x \to z} \langle V_{j_1}^0(0;0) V_{j_2}^0(1;1) V_{j_3}^0(\infty;\infty) V_{j_4}^0(x;z) \rangle$$

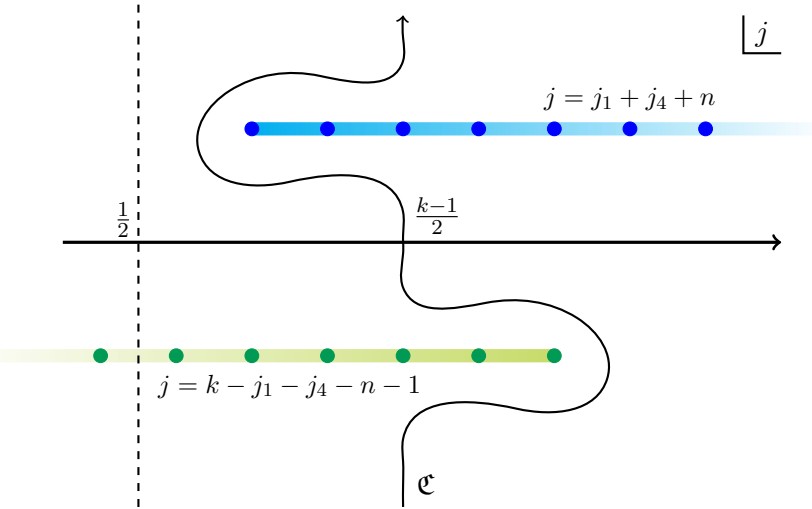

**Figure 1**. The deformed contour $\mathfrak{C}$. The green poles are not present in the original integral (A.39), but are added in after we symmetrise the integrand in $j \to k - 1 - j$ in eq. (A.40).

$$
= \frac{1}{2} \int_{\mathfrak{C}} dj\, \mathcal{C}(j) \left[ \frac{\Gamma(k-2j)^2 \Gamma(k-j_1-j_2-j_3-j_4)^2}{\Gamma(k-j_1-j_4-j)^2 \Gamma(k-j_2-j_3-j)^2} \right.
$$
$$
- \frac{\gamma(k-2j)^2 \gamma(j_1+j_4+j-k+1) \gamma(j_2+j_3+j-k+1)}{(k-2j-1)^2 \gamma(j_1+j_4-j) \gamma(j_2+j_3-j)}
$$
$$
\left. \times \frac{\Gamma(-k+2+2j)^2 \Gamma(k-j_1-j_2-j_3-j_4)^2}{\Gamma(-j_1-j_4+j+1)^2 \Gamma(-j_2-j_3+j+1)^2} \right]
$$
$$
\times \mathcal{F}^{\mathrm{Vir}}_{\alpha=bj}(\alpha_i = bj_i; z) \mathcal{F}^{\mathrm{Vir}}_{\alpha=bj}(\alpha_i = bj_i; \bar{z}) . \qquad (A.42)
$$

In deriving (A.42) we used that the Virasoro conformal blocks depend just on the conformal dimension $\alpha(Q_{\mathrm{L}} - \alpha)$ so that

$$
\mathcal{F}^{\mathrm{Vir}}_{\alpha=bj}(\alpha_i = bj_i; z) = \mathcal{F}^{\mathrm{Vir}}_{\alpha=b(k-1-j)}(\alpha_i = bj_i; z) . \qquad (A.43)
$$

Making use of the identities

$$
\gamma(t) = \frac{1}{\gamma(1-t)} , \quad \gamma(t+1) = -t^2 \gamma(t) , \quad \Gamma(t)^2 = \frac{\pi \gamma(t)}{\sin(\pi t)} , \qquad (A.44)
$$

valid for $t = \bar{t}$, the term in the square brackets in eq. (A.42) can be rewritten as

$$
\frac{\gamma(k-2j)\gamma(k-j_1-j_2-j_3-j_4)}{\gamma(k-j_1-j_4-j)\gamma(k-j_2-j_3-j)} \left[ \frac{\sin(\pi(-j+j_1+j_4))\sin(\pi(j-j_2-j_3))}{\sin(\pi(k-2j))\sin(\pi(k-j_1-j_2-j_3-j_4))} \right.
$$
$$
\left. - \frac{\sin(\pi(k-j_1-j_4-j))\sin(\pi(-k+j_2+j_3+j))}{\sin(\pi(k-2j))\sin(\pi(k-j_1-j_2-j_3-j_4))} \right]
$$
$$
= \frac{\gamma(k-2j)\gamma(k-j_1-j_2-j_3-j_4)}{\gamma(k-j_1-j_4-j)\gamma(k-j_2-j_3-j)} , \qquad (A.45)
$$

where we made use of the identity

$$-\sin\theta_1\sin\theta_3 + \sin\theta_2\sin\theta_4 = \sin(\theta_1+\theta_2)\sin(\theta_1+\theta_4)\ , \qquad \text{for } \theta_1+\theta_2+\theta_3+\theta_4 = 0\ . \quad \text{(A.46)}$$

Putting everything together and making use of eq. (A.24) we have

$$\lim_{x\to z}\langle V_{j_1}^0(0;0)V_{j_2}^0(1;1)V_{j_3}^0(\infty;\infty)V_{j_4}^0(x;z)\rangle$$

$$= \frac{1}{2}\int_{\mathfrak{C}}\mathrm{d}j\,\mathcal{C}(j)\frac{\gamma(k-2j)\,\gamma(k-j_1-j_2-j_3-j_4)}{\gamma(k-j_1-j_4-j)\,\gamma(k-j_2-j_3-j)}\mathcal{F}_{bj}^{\mathrm{Vir}}(bj_i;z)\,\mathcal{F}_{bj}^{\mathrm{Vir}}(bj_i;\bar{z}) \qquad \text{(A.47)}$$

$$= \frac{\nu^{2-k}\gamma(-\frac{1}{k-2})^2\,\gamma(k-j_1-j_2-j_3-j_4)}{8\pi^3(k-2)\prod_{i=1}^4\gamma(\frac{2j_i-1}{k-2})}\times$$

$$\times \int_{\mathfrak{C}}\mathrm{d}j\,D_{\mathrm{L}}(bj_1,bj_4,bj)\,D_{\mathrm{L}}(bj_2,bj_3,Q_{\mathrm{L}}-bj)\,\mathcal{F}_{bj}^{\mathrm{Vir}}(bj_i;z)\,\mathcal{F}_{bj}^{\mathrm{Vir}}(bj_i;\bar{z})\ . \qquad \text{(A.48)}$$

As a last step, we now relate this to a Liouville correlator with external Liouville momenta $\alpha_i = bj_i$. However, note that $\mathrm{Re}(bj_i) = \frac{b}{2}$, which is not the usual value of $\frac{Q_{\mathrm{L}}}{2}$ that we have in Liouville theory. Rather, we will actually relate the right-hand-side of eq. (A.48) to an analytically continued Liouville correlator.

To get an honest Liouville correlator, we now start analytically continuing $j_i$ from $\mathrm{Re}(j_i) = \frac{1}{2}$ to $\mathrm{Re}(j_i) = \frac{Q_{\mathrm{L}}}{2b} = \frac{1}{2}(1+b^{-2}) = \frac{k-1}{2}$. This has the effect of moving the blue poles in Figure 1 to the right and the green poles to the left (and no new poles enter the picture). Thus after the analytic continuation we can straighten the integration contour and simply integrate over $j \in \frac{k-1}{2} + i\mathbb{R}$. Thus we have for $j \in \frac{k-1}{2} + i\mathbb{R}$:

$$\lim_{x\to z}\langle V_{j_1}^0(0;0)V_{j_2}^0(1;1)V_{j_3}^0(\infty;\infty)V_{j_4}^0(x;z)\rangle = \frac{\nu^{2-k}\gamma(\frac{1}{2-k})^2\,\gamma(k-\sum_i j_i)}{8\pi^3(k-2)\prod_{i=1}^4\gamma(\frac{2j_i-1}{k-2})}$$

$$\times \int_{\frac{k-1}{2}+i\mathbb{R}}\mathrm{d}j\,D_{\mathrm{L}}(bj_1,bj_4,bj)\,D_{\mathrm{L}}(bj_2,bj_3,Q_{\mathrm{L}}-bj)\,\mathcal{F}_{bj}^{\mathrm{Vir}}(bj_i;z)\,\mathcal{F}_{bj}^{\mathrm{Vir}}(bj_i;\bar{z})\ . \quad \text{(A.49)}$$

In Liouville CFT, we have the following conformal block expansion for four-point functions, see e.g. [74][23]

$$\langle V_{\alpha_1}(0)V_{\alpha_2}(1)V_{\alpha_3}(\infty)V_{\alpha_4}(z)\rangle_{\mathrm{L}}$$

$$= \frac{1}{4\pi}\int_{\frac{Q_{\mathrm{L}}}{2}+i\mathbb{R}}\mathrm{d}\alpha\,D_{\mathrm{L}}(\alpha_1,\alpha_4,\alpha)\,D_{\mathrm{L}}(\alpha_2,\alpha_3,Q_{\mathrm{L}}-\alpha)\,\mathcal{F}_{\alpha}^{\mathrm{Vir}}(\alpha_i;z)\,\mathcal{F}_{\alpha}^{\mathrm{Vir}}(\alpha_i;\bar{z})\ . \quad \text{(A.50)}$$

Hence, we finally obtain

$$\lim_{x\to z}\langle V_{j_1}^0(0;0)V_{j_2}^0(1;1)V_{j_3}^0(\infty;\infty)V_{j_4}^0(x;z)\rangle$$

$$= \frac{\nu^{2-k}\,\gamma(\frac{1}{2-k})^2\,\gamma(k-\sum_i j_i)}{2\pi^2\sqrt{k-2}\prod_{i=1}^4\gamma(\frac{2j_i-1}{k-2})}\langle V_{bj_1}(0)V_{bj_2}(1)V_{bj_3}(\infty)V_{bj_4}(z)\rangle_{\mathrm{L}}\ . \quad \text{(A.51)}$$

---

[23]The extra factor of $\frac{1}{4\pi}$ is needed to ensure the correct normalisation of the conformal block expansion because $D_{\mathrm{L}}(\alpha_1,\alpha_2,\alpha_3)\to 2\pi\delta(\alpha_1+\alpha_2-Q_{\mathrm{L}})+$ reflected term for $\alpha_3\to 0$.

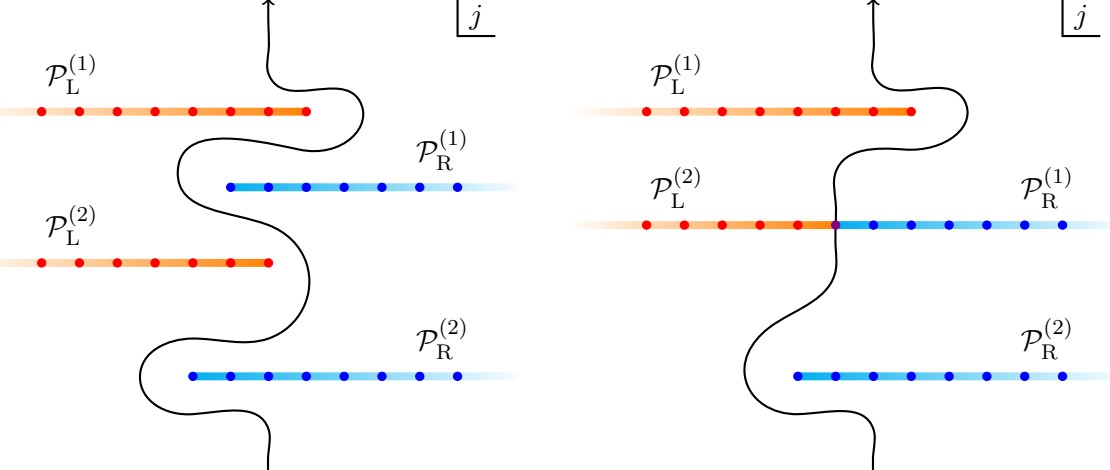

**Figure 2**. For generic values of the spins $j_i$ the integration contour in eq. (A.5) can be deformed to avoid poles (left figure). For some special values of the spins $j_i$ the integration contour in eq. (A.5) gets pinched (right figure) and the four-point function develops a pole.

This identity holds for $j_i \in \frac{k-1}{2} + i\mathbb{R}$. However, other values are uniquely determined by analytic continuation and thus the relation holds for all $j_i \in \mathbb{C}$.

Making use of the identities of Appendix B.2 one can derive similar relations for $x \to 0, 1, \infty$. For example, taking the $x \to 1$ limit of eq. (B.6) and making use of (A.51) we find

$$
\lim_{x \to 1} \langle V_{j_1}^0(0;0) V_{j_2}^0(1;1) V_{j_3}^0(\infty;\infty) V_{j_4}^0(x;z) \rangle
$$
$$
= |z|^{2j_4} \frac{\gamma(\frac{1}{2-k})^2 \gamma(j_1 - j_2 + j_3 - j_4)}{2\pi^2 \sqrt{k-2}\, \nu^{k-2} \gamma\left(\frac{2j_2-1}{k-2}\right) \gamma\left(\frac{2j_4-1}{k-2}\right)}
$$
$$
\times \; \langle V_{b(\frac{k}{2}-j_1)}(0) V_{bj_2}(1) V_{b(\frac{k}{2}-j_3)}(\infty) V_{bj_4}(z) \rangle_{\mathrm{L}} \;, \quad \text{(A.52)}
$$

which after analytic continuation again holds for all possible spins. We collect similar identities in Appendix B.3.

# B  Identities for unflowed four-point functions

In this appendix we derive and collect various identities for unflowed correlators.

## B.1  Poles of the unflowed correlators

Let us derive the location of all the poles of the unflowed four-point correlator as a function of the external spins $j_i$. For this we use the conformal block expansion (A.5) of the $H_3^+$ model, from which one can read off singularities of the four-point function. The integrand has various poles due to the non-trivial pole structure of the structure constants. Let us denote in the following

$$
\mathcal{S} = \{n + m(k-2) \,|\, m,\, n \in \mathbb{Z}_{\geq 0}\} \subset \mathbb{R}_{\geq 0} \;. \quad \text{(B.1)}
$$

The structure constant $D(j_1, j_2, j_3)$ has a pole whenever (compare with eq. (A.37))

$$j_1 + j_2 + j_3 \in k + \mathcal{S} \ , \tag{B.2a}$$

$$j_1 + j_2 - j_3 \in k - 1 + \mathcal{S} \ , \tag{B.2b}$$

$$j_1 - j_2 - j_3 \in \mathcal{S} \ , \tag{B.2c}$$

$$-j_1 - j_2 - j_3 \in -1 + \mathcal{S} \ , \tag{B.2d}$$

where we used hopefully obvious notation for the translates of the set $\mathcal{S}$. It is always understood that these conditions include also all the poles obtained by permuting the roles of $j_1$, $j_2$ and $j_3$ such as $j_1 - j_2 + j_3 \in k - 1 + \mathcal{S}$. Hence the product $D(j_1, j_4, j)D(j_2, j_3, j)$ of structure constants leads to an intricate pole structure for the integrand of the conformal block expansion. Once we start analytically continuing the external spins $j_i$, these poles start to move around in the $j$-plane. The resulting integral will have a singularity whenever the contour gets trapped between two poles and cannot be deformed to avoid the singularity, see Figure 2.

We can analyse this systematically as follows. As a function of $j$, $D(j_1, j_4, j)$ has four families of poles $\mathcal{P}_{\mathrm{L}}^{(1)}$ that are to the left of $\mathrm{Re}(j) = \frac{1}{2}$ for $\mathrm{Re}(j_1) = \mathrm{Re}(j_4) = \frac{1}{2}$ and four families of poles $\mathcal{P}_{\mathrm{R}}^{(1)}$ to the right. They are

$$\mathcal{P}_{\mathrm{L}}^{(1)} = (1 - j_1 - j_4 - \mathcal{S}) \cup (j_1 - j_4 - \mathcal{S}) \cup (j_4 - j_1 - \mathcal{S}) \cup (j_1 + j_4 + 1 - k - \mathcal{S}) \tag{B.3a}$$

$$\mathcal{P}_{\mathrm{R}}^{(1)} = (j_1 + j_4 + \mathcal{S}) \cup (k - j_1 - j_4 + \mathcal{S}) \cup (j_1 - j_4 + k - 1 + \mathcal{S}) \cup (j_4 - j_1 + k - 1 + \mathcal{S}) \ . \tag{B.3b}$$

We also denote by $\mathcal{P}_{\mathrm{L}}^{(2)}$ and $\mathcal{P}_{\mathrm{R}}^{(2)}$ the corresponding quantities for the second structure constant $D(j_2, j_3, j)$. The contour can only become trapped between poles of a left- and a right-type, since we could for example just move the contour right when two poles of left-type coincide. Thus the four-point function has a singularity whenever

$$\mathcal{P}_{\mathrm{L}}^{(1)} \cap \mathcal{P}_{\mathrm{R}}^{(2)} \neq 0 \quad \text{or} \quad \mathcal{P}_{\mathrm{L}}^{(2)} \cap \mathcal{P}_{\mathrm{R}}^{(1)} \neq 0 \ . \tag{B.4}$$

This leads immediately to the singularities

$$j_1 + j_2 + j_3 + j_4 \in 2k - 1 + \mathcal{S} \ , \tag{B.5a}$$

$$j_1 + j_2 + j_3 - j_4 \in k + \mathcal{S} \ , \tag{B.5b}$$

$$j_1 + j_2 - j_3 - j_4 \in k - 1 + \mathcal{S} \ , \tag{B.5c}$$

$$j_1 - j_2 - j_3 - j_4 \in \mathcal{S} \ , \tag{B.5d}$$

$$-j_1 - j_2 - j_3 - j_4 \in -1 + \mathcal{S} \ , \tag{B.5e}$$

together with all the singularities obtained by permuting the labels of the spins. As a consistency check, we notice that this set of poles reduces to the set of poles for the three-point function when we set $j_4 = 0$. As a further consistency check we note that this is consistent with all the flip identities derived in Appendix B.2 that allow us to simultaneously replace two spins $j_i$ by $\frac{k}{2} - j_i$.

## B.2 Flip identities

We are interested in identities similar to

$$\left\langle V_{j_1}^0(0;0)V_{j_2}^0(1;1)V_{j_3}^0(\infty;\infty)V_{j_4}^0(x;z)\right\rangle$$
$$= \mathcal{N}(j_1)\,\mathcal{N}(j_3)\,|x|^{-4j_4}\,|z|^{2j_4}\left\langle V_{\frac{k}{2}-j_1}^0(0;0)V_{j_2}^0(1;1)V_{\frac{k}{2}-j_3}^0(\infty;\infty)V_{j_4}^0\left(\frac{z}{x};z\right)\right\rangle , \quad \text{(B.6)}$$

which has been derived in [26, 70] . We will refer to these identities as *flip identities*. They can be obtained by composing (B.6) with global Ward identities. For example, we have

$$\left\langle V_{j_1}^0(0;0)V_{j_2}^0(1;1)V_{j_3}^0(\infty;\infty)V_{j_4}^0(x;z)\right\rangle$$
$$= \left\langle V_{j_2}^0(0;0)V_{j_1}^0(1;1)V_{j_3}^0(\infty;\infty)V_{j_4}^0(1-x;1-z)\right\rangle \tag{B.7}$$
$$= \mathcal{N}(j_2)\,\mathcal{N}(j_3)\,|1-x|^{-4j_4}\,|1-z|^{2j_4}$$
$$\times \left\langle V_{\frac{k}{2}-j_2}^0(0;0)V_{j_1}^0(1;1)V_{\frac{k}{2}-j_3}^0(\infty;\infty)V_{j_4}^0\left(\frac{1-z}{1-x};1-z\right)\right\rangle \tag{B.8}$$
$$= \mathcal{N}(j_2)\,\mathcal{N}(j_3)\,|1-x|^{-4j_4}\,|1-z|^{2j_4}$$
$$\times \left\langle V_{j_1}^0(0;0)V_{\frac{k}{2}-j_2}^0(1;1)V_{\frac{k}{2}-j_3}^0(\infty;\infty)V_{j_4}^0\left(\frac{z-x}{1-x};z\right)\right\rangle , \tag{B.9}$$

where we used global Ward identities in the first and third equality and eq. (B.6) in the second. Similar identities read

$$\left\langle V_{j_1}^0(0;0)V_{j_2}^0(1;1)V_{j_3}^0(\infty;\infty)V_{j_4}^0(x;z)\right\rangle$$
$$= \mathcal{N}(j_1)\,\mathcal{N}(j_2)\,|z|^{2j_4}\,|1-z|^{2j_4}\,|x-z|^{-4j_4}$$
$$\times \left\langle V_{\frac{k}{2}-j_1}^0(0;0)V_{\frac{k}{2}-j_2}^0(1;1)V_{j_3}^0(\infty;\infty)V_{j_4}^0\left(\frac{z(x-1)}{x-z};z\right)\right\rangle , \tag{B.10}$$
$$= \mathcal{N}(j_1)\,\mathcal{N}(j_4)\,|z|^{2(j_1+j_4-\frac{k}{2})}\,|1-z|^{2j_2}|x|^{2(j_2+j_3-j_1-j_4)}\,|1-x|^{2(j_1-j_2+j_3+j_4-k)}$$
$$\times |x-z|^{-2(j_1+j_2+j_3+j_4-k)}\left\langle V_{\frac{k}{2}-j_1}^0(0;0)V_{j_2}^0(1;1)V_{j_3}^0(\infty;\infty)V_{\frac{k}{2}-j_4}^0\left(\frac{x-z}{x-1};z\right)\right\rangle , \tag{B.11}$$
$$= \mathcal{N}(j_2)\,\mathcal{N}(j_4)\,|1-x|^{2(j_1-j_2+j_3-j_4)}x^{2(-j_1+j_2+j_3+j_4-k)}\,|1-z|^{2(j_2+j_4-\frac{k}{2})}\,|z|^{2j_1}$$
$$\times |x-z|^{-2(j_1+j_2+j_3+j_4-k)}\left\langle V_{j_1}^0(0;0)V_{\frac{k}{2}-j_2}^0(1;1)V_{j_3}^0(\infty;\infty)V_{\frac{k}{2}-j_4}^0\left(\frac{z}{x};z\right)\right\rangle , \tag{B.12}$$
$$= \mathcal{N}(j_3)\,\mathcal{N}(j_4)\,|x|^{2(j_2+j_3-j_1-j_4)}\,|1-x|^{2(j_1+j_3-j_2-j_4)}\,|x-z|^{2(j_4-j_1-j_2-j_3)}$$
$$\times |1-z|^{2j_2}\,|z|^{2j_1}\left\langle V_{j_1}^0(0;0)V_{j_2}^0(1;1)V_{\frac{k}{2}-j_3}^0(\infty;\infty)V_{\frac{k}{2}-j_4}^0\left(\frac{z(x-1)}{x-z};z\right)\right\rangle , \tag{B.13}$$
$$= \mathcal{N}(j_1)\mathcal{N}(j_2)\mathcal{N}(j_3)\mathcal{N}(j_4)|x|^{2(-j_1+j_2+j_3-j_4)}|1-x|^{2(j_1-j_2+j_3-j_4)}$$
$$\times |z|^{2(j_1+j_4-\frac{k}{2})}|1-z|^{2(j_2+j_4-\frac{k}{2})}|x-z|^{2(k-j_1-j_2-j_3-j_4)}$$
$$\times \left\langle V_{\frac{k}{2}-j_1}^0(0;0)V_{\frac{k}{2}-j_2}^0(1;1)V_{\frac{k}{2}-j_3}^0(\infty;\infty)V_{\frac{k}{2}-j_4}^0(x;z)\right\rangle . \tag{B.14}$$

## B.3 Collision identities

Following the procedure outlined in Appendix A.6 one can derive identities similar to eqs. (A.51) and (A.52). We collect them here and refer to them as *collision identities*. We

also introduce the notation $f_{\boldsymbol{\varepsilon}}(z)$ with $\boldsymbol{\varepsilon} = (\varepsilon_1, \varepsilon_2, \varepsilon_3, \varepsilon_4)$, which turns out to be useful in Section 3.

$$f_{(0,1,0,0)}(z) \equiv f_{(0,1,1,0)}(z) \equiv \lim_{x \to 0} \langle V_{j_1}^0(0;0) V_{j_2}^0(1;1) V_{j_3}^0(\infty;\infty) V_{j_4}^0(x;z) \rangle$$

$$= |1 - z|^{2j_4} \frac{\gamma(\frac{1}{2-k})^2 \gamma(-j_1 + j_2 + j_3 - j_4)}{2\pi^2 \sqrt{k-2}\, \nu^{k-2} \gamma(\frac{2j_1-1}{k-2}) \gamma(\frac{2j_4-1}{k-2})}$$

$$\times \langle V_{bj_1}(0) V_{b(\frac{k}{2}-j_2)}(1) V_{b(\frac{k}{2}-j_3)}(\infty) V_{bj_4}(z) \rangle_{\mathrm{L}} \,, \tag{B.15a}$$

$$f_{(1,0,0,1)}(z) \equiv f_{(1,0,1,1)}(z) \equiv \lim_{x \to 0} |x|^{2(j_1-j_2-j_3+j_4)} \langle V_{j_1}^0(0;0) V_{j_2}^0(1;1) V_{j_3}^0(\infty;\infty) V_{j_4}^0(x;z) \rangle$$

$$= |z|^{2(\frac{k}{2}-j_2-j_3)} |1 - z|^{2j_2} \frac{\gamma(\frac{1}{2-k})^2 \gamma(j_1 - j_2 - j_3 + j_4)}{2\pi^2 \sqrt{k-2}\, \nu^{k-2} \gamma(\frac{2j_2-1}{k-2}) \gamma(\frac{2j_3-1}{k-2})}$$

$$\times \langle V_{b(\frac{k}{2}-j_1)}(0) V_{bj_2}(1) V_{bj_3}(\infty) V_{b(\frac{k}{2}-j_4)}(z) \rangle_{\mathrm{L}} \,, \tag{B.15b}$$

$$f_{(1,0,0,0)}(z) \equiv f_{(1,0,1,0)}(z) \equiv \lim_{x \to 1} \langle V_{j_1}^0(0;0) V_{j_2}^0(1;1) V_{j_3}^0(\infty;\infty) V_{j_4}^0(x;z) \rangle$$

$$= |z|^{2j_4} \frac{\gamma(\frac{1}{2-k})^2 \gamma(j_1 - j_2 + j_3 - j_4)}{2\pi^2 \sqrt{k-2}\, \nu^{k-2} \gamma(\frac{2j_2-1}{k-2}) \gamma(\frac{2j_4-1}{k-2})}$$

$$\times \langle V_{b(\frac{k}{2}-j_1)}(0) V_{bj_2}(1) V_{b(\frac{k}{2}-j_3)}(\infty) V_{bj_4}(z) \rangle_{\mathrm{L}} \,, \tag{B.15c}$$

$$f_{(0,1,0,1)}(z) \equiv f_{(0,1,1,1)}(z) \equiv \lim_{x \to 1} |1 - x|^{2(-j_1+j_2-j_3+j_4)} \langle V_{j_1}^0(0;0) V_{j_2}^0(1;1) V_{j_3}^0(\infty;\infty) V_{j_4}^0(x;z) \rangle$$

$$= |z|^{2j_1} |1 - z|^{2(\frac{k}{2}-j_1-j_3)} \frac{\gamma(\frac{1}{2-k})^2 \gamma(-j_1 + j_2 - j_3 + j_4)}{2\pi^2 \sqrt{k-2}\, \nu^{k-2} \gamma(\frac{2j_1-1}{k-2}) \gamma(\frac{2j_3-1}{k-2})}$$

$$\times \langle V_{bj_1}(0) V_{b(\frac{k}{2}-j_2)}(1) V_{bj_3}(\infty) V_{b(\frac{k}{2}-j_4)}(z) \rangle_{\mathrm{L}} \,, \tag{B.15d}$$

$$f_{(0,0,1,0)}(z) \equiv f_{(0,0,0,0)}(z) \equiv \lim_{x \to z} \langle V_{j_1}^0(0;0) V_{j_2}^0(1;1) V_{j_3}^0(\infty;\infty) V_{j_4}^0(x;z) \rangle$$

$$= \frac{\gamma(\frac{1}{2-k})^2 \gamma(k - j_1 - j_2 - j_3 - j_4)}{2\pi^2 \sqrt{k-2}\, \nu^{k-2} \prod_{i=1}^{4} \gamma(\frac{2j_i-1}{k-2})} \langle V_{bj_1}(0) V_{bj_2}(1) V_{bj_3}(\infty) V_{bj_4}(z) \rangle_{\mathrm{L}} \,, \tag{B.15e}$$

$$f_{(1,1,0,1)}(z) \equiv f_{(1,1,1,1)}(z) \equiv \lim_{x \to z} |x - z|^{2(j_1+j_2+j_3+j_4-k)} \langle V_{j_1}^0(0;0) V_{j_2}^0(1;1) V_{j_3}^0(\infty;\infty) V_{j_4}^0(x;z) \rangle$$

$$= |z|^{2(j_2+j_3-\frac{k}{2})} |1 - z|^{2(j_1+j_3-\frac{k}{2})} \frac{\gamma(\frac{1}{2-k})^2 \gamma(j_1 + j_2 + j_3 + j_4 - k)}{2\pi^2 \sqrt{k-2}\, \nu^{k-2}}$$

$$\times \langle V_{b(\frac{k}{2}-j_1)}(0) V_{b(\frac{k}{2}-j_2)}(1) V_{b(\frac{k}{2}-j_3)}(\infty) V_{b(\frac{k}{2}-j_4)}(z) \rangle_{\mathrm{L}} \,, \tag{B.15f}$$

$$f_{(0,0,0,1)}(z) \equiv f_{(0,0,1,1)}(z) \equiv \lim_{x \to \infty} |x|^{2(j_1+j_2-j_3+j_4)} \langle V_{j_1}^0(0;0) V_{j_2}^0(1;1) V_{j_3}^0(\infty;\infty) V_{j_4}^0(x;z) \rangle$$

$$= |z|^{2j_1} |1 - z|^{2j_2} \frac{\gamma(\frac{1}{2-k})^2 \gamma(-j_1 - j_2 + j_3 + j_4)}{2\pi^2 \sqrt{k-2}\, \nu^{k-2} \gamma(\frac{2j_1-1}{k-2}) \gamma(\frac{2j_2-1}{k-2})}$$

$$\times \langle V_{bj_1}(0) V_{bj_2}(1) V_{b(\frac{k}{2}-j_3)}(\infty) V_{b(\frac{k}{2}-j_4)}(z) \rangle_{\mathrm{L}} \,, \tag{B.15g}$$

$$f_{(1,1,0,0)}(z) \equiv f_{(1,1,1,0)}(z) \equiv \lim_{x \to \infty} |x|^{4j_4} \langle V_{j_1}^0(0;0) V_{j_2}^0(1;1) V_{j_3}^0(\infty;\infty) V_{j_4}^0(x;z) \rangle$$

$$= |z|^{2j_4} |1 - z|^{2j_4} \frac{\gamma(\frac{1}{2-k})^2 \gamma(j_1 + j_2 - j_3 - j_4)}{2\pi^2 \sqrt{k-2}\, \nu^{k-2} \gamma(\frac{2j_3-1}{k-2}) \gamma(\frac{2j_4-1}{k-2})}$$

$$\times \langle V_{b(\frac{k}{2}-j_1)}(0) V_{b(\frac{k}{2}-j_2)}(1) V_{bj_3}(\infty) V_{bj_4}(z) \rangle_{\mathrm{L}} \,. \tag{B.15h}$$

## B.4 Unflowed correlators for special combinations of the spins

Let us collect here identities similar to eq. (3.25). They can be derived following the procedure of Section 3.2. Denoting the unflowed correlator as $F(c, z)$, see eq. (3.3), we find

$$F(c,z)\Big|_{\substack{c\sim 0, \\ \epsilon\equiv j_1-j_2-j_3+j_4+1\sim 0}} = \frac{\gamma(\frac{1}{2-k})^2\,|z|^{-\frac{4j_1j_4}{k-2}}|z-1|^{-\frac{4(1-j_2)j_4}{k-2}}}{2\pi^2\,\nu^{k-2}\gamma(\frac{2j_1-1}{k-2})\gamma(\frac{2j_4-1}{k-2})}\ , \tag{B.16a}$$

$$F(c,z)\Big|_{\substack{c\sim 0, \\ \epsilon\equiv -j_1+j_2+j_3-j_4+1\sim 0}} = \frac{\gamma(\frac{1}{2-k})^2\,|c|^{-2}|z|^{-\frac{4(1-j_1)(1-j_4)}{k-2}}|z-1|^{-\frac{4j_2(1-j_4)}{k-2}}}{2\pi^2\,\nu^{k-2}\gamma(\frac{2j_2-1}{k-2})\gamma(\frac{2j_3-1}{k-2})}\ , \tag{B.16b}$$

$$F(c,z)\Big|_{\substack{c\sim 1, \\ \epsilon\equiv -j_1+j_2-j_3+j_4+1\sim 0}} = \frac{\gamma(\frac{1}{2-k})^2\,|z|^{-\frac{4(1-j_1)j_4}{k-2}}|z-1|^{-\frac{4j_2j_4}{k-2}}}{2\pi^2\,\nu^{k-2}\gamma(\frac{2j_2-1}{k-2})\gamma(\frac{2j_4-1}{k-2})}\ , \tag{B.16c}$$

$$F(c,z)\Big|_{\substack{c\sim 1, \\ \epsilon\equiv -j_1+j_2+j_3-j_4+1\sim 0}} = \frac{\gamma(\frac{1}{2-k})^2\,|c-1|^{-2}|z|^{-\frac{4j_1(1-j_4)}{k-2}}|z-1|^{-\frac{4(1-j_2)(1-j_4)}{k-2}}}{2\pi^2\,\nu^{k-2}\gamma(\frac{2j_1-1}{k-2})\gamma(\frac{2j_3-1}{k-2})}\ , \tag{B.16d}$$

$$F(c,z)\Big|_{\substack{c\sim z, \\ \epsilon\equiv \sum_i j_i-k+1\sim 0}} = \frac{\gamma(\frac{1}{2-k})^2}{2\pi^2\,\nu^{k-2}\prod_{i=1}^4\gamma(\frac{2j_i-1}{k-2})}|z|^{-\frac{4j_1j_4}{k-2}}|z-1|^{-\frac{4j_2j_4}{k-2}}\ , \tag{B.16e}$$

$$F(c,z)\Big|_{\substack{c\sim z, \\ \epsilon\equiv -\sum_i j_i+k+1\sim 0}} = \frac{\gamma(\frac{1}{2-k})^2|z-c|^{-2}|z|^{-\frac{4(1-j_1)(1-j_4)}{k-2}}|z-1|^{-\frac{4(1-j_2)(1-j_4)}{k-2}}}{2\pi^2\,\nu^{k-2}}\ , \tag{B.16f}$$

$$F(c,z)\Big|_{\substack{c\sim \infty, \\ \epsilon\equiv j_1+j_2-j_3-j_4+1\sim 0}} = \frac{\gamma(\frac{1}{2-k})^2|c|^{-2}|z|^{-\frac{4j_1(1-j_4)}{k-2}}|z-1|^{-\frac{4j_2(1-j_4)}{k-2}}}{2\pi^2\,\nu^{k-2}\gamma(\frac{2j_1-1}{k-2})\gamma(\frac{2j_2-1}{k-2})}\ , \tag{B.16g}$$

$$F(c,z)\Big|_{\substack{c\sim \infty, \\ \epsilon\equiv -j_1-j_2+j_3+j_4+1\sim 0}} = \frac{\gamma(\frac{1}{2-k})^2|c|^{-4j_4}|z|^{-\frac{4j_4(1-j_1)}{k-2}}|z-1|^{-\frac{4j_4(1-j_2)}{k-2}}}{2\pi^2\,\nu^{k-2}\gamma(\frac{2j_3-1}{k-2})\gamma(\frac{2j_4-1}{k-2})}\ . \tag{B.16h}$$

## C   Identities for the polynomials $P_w(x; z)$

In this appendix we collect various identities satisfied by the polynomials $P_{\boldsymbol{w}}(x; z)$, see eq. (2.15) for their definition.

### C.1   Shifted $P_{\boldsymbol{w}+\boldsymbol{\delta}}(x; z)$ evaluated at $P_{\boldsymbol{w}}(x; z) = 0$

The first class of identities concerns the polynomial $P_{\boldsymbol{w}+\boldsymbol{\delta}}(x; z)$ when evaluated on the locus $P_{\boldsymbol{w}}(x; z) = 0$. On this locus, a covering map with ramification indices $\boldsymbol{w} = (w_1, w_2, w_3, w_4)$ exists and hence the quantities $a_i$ and $\Pi$ have meaning (see eqs. (3.13) and eqs. (3.14) for their definition). It turns out that $P_{\boldsymbol{w}+\boldsymbol{\delta}}(x; z)$ can be evaluated in terms of these quantities, at least for specific choices of $\boldsymbol{\delta} \equiv (\delta_1, \delta_2, \delta_3, \delta_4)$. The identity takes the form

$$P_{\boldsymbol{w}+\boldsymbol{\delta}}(x; z)\Big|_{P_{\boldsymbol{w}}(x;z)=0} = A_{\boldsymbol{w},\boldsymbol{\delta}}\, z^{p(|\boldsymbol{\delta}|)}\,(1-z)^{q(|\boldsymbol{\delta}|)}\,\Pi^{\frac{1}{2}}\prod_{i=1}^4 a_i^{\frac{w_i+2\delta_i+1}{4}}\ . \tag{C.1}$$

Of course, this identity is valid only when a covering map for $\boldsymbol{w}$ exists. Moreover, we require that $P_{\boldsymbol{w}+\boldsymbol{\delta}}(x;z)$ is not identically zero. In eq. (C.1), $|\boldsymbol{\delta}| = (|\delta_1|, |\delta_2|, |\delta_3|, |\delta_4|)$. The exponents $p(|\boldsymbol{\delta}|)$ and $q(|\boldsymbol{\delta}|)$ depend on the choice of $\boldsymbol{\delta}$ and we have

$$p(0,0,0,2) = -\frac{1}{2}\,, \quad p(0,0,1,1) = 0\,, \quad p(0,0,2,0) = \frac{1}{2}\,, \quad p(0,1,0,1) = 0\,, \quad \text{(C.2a)}$$

$$p(0,1,1,0) = \frac{1}{2}\,, \quad p(0,2,0,0) = \frac{1}{2}\,, \quad p(1,0,0,1) = 0\,, \quad p(1,0,1,0) = 0\,, \quad \text{(C.2b)}$$

$$p(1,1,0,0) = 0\,, \quad p(1,1,1,1) = 0\,, \quad p(2,0,0,0) = -\frac{1}{2}\,, \quad \text{(C.2c)}$$

while $q(|\delta_1|, |\delta_2|, |\delta_3|, |\delta_4|) = p(|\delta_2|, |\delta_1|, |\delta_3|, |\delta_4|)$. The constant prefactor $A_{\boldsymbol{w},\boldsymbol{\delta}}$ takes the form

$$A_{\boldsymbol{w},\boldsymbol{\delta}} = \text{phase}(\boldsymbol{w}, |\boldsymbol{\delta}|)\,(-1)^{\frac{1}{2\max_i |\delta_i|}\sum_i \delta_i}$$

$$\times \prod_{i=1}^{4} w_i^{\frac{w_i+1-2|\delta_i|}{4}} \begin{cases} 1\,, & \sum_i |\delta_i| < 4\,, \\ 1 + \frac{1}{2}\sum_{i=1}^{4}\delta_i w_i\,, & \sum_i |\delta_i| = 4\,. \end{cases} \quad \text{(C.3)}$$

The identity only holds for $\boldsymbol{\delta}$'s such that $|\boldsymbol{\delta}|$ appears in the list (C.2). We have checked in Mathematica that this holds for all allowed $\boldsymbol{w}$ such that $\sum_i w_i \leq 12$. We haven't tried to fully fix the phase entering eq. (C.3), since we don't need its dependence on $|\boldsymbol{\delta}|$ and $\boldsymbol{w}$.

## C.2 Derivative of $P_{\boldsymbol{w}}(x;z)$ evaluated at $P_{\boldsymbol{w}}(x;z) = 0$

We also need the first derivative of $P_{\boldsymbol{w}}(x;z)$ evaluated on the vanishing locus of $P_{\boldsymbol{w}}(x;z)$. It essentially has the same form as the identity (C.1) for $\boldsymbol{\delta} = 0$ and takes the form

$$\partial_z P_{\boldsymbol{w}}(x;z)\Big|_{P_{\boldsymbol{w}}(x;z)=0} = \text{phase} \times z^{-\frac{1}{2}}(1-z)^{-\frac{1}{2}}\Pi^{\frac{1}{2}}\prod_{i=1}^{4}(a_i w_i)^{\frac{w_i+1}{4}}\,. \quad \text{(C.4)}$$

We checked this in the ancillary Mathematica notebook for all allowed values of $\boldsymbol{w}$ with $\sum_i w_i \leq 12$.

## C.3 $X_{ij}$ evaluated at $P_{\boldsymbol{w}}(x;z) = 0$

When evaluated on the vanishing locus of $P_{\boldsymbol{w}}(x;z)$, also the 'generalised differences' $X_{ij}$ simplify:

$$X_{ij}\Big|_{P_{\boldsymbol{w}}(x;z)=0} = d_{ij}\,(1 - a_i^{-1}y_i)(1 - a_j^{-1}y_j)\,P_{\boldsymbol{w}+e_i+e_j}(x;z)\Big|_{P_{\boldsymbol{w}}(x;z)=0}\,, \quad \text{(C.5)}$$

where

$$d_{12} = d_{13} = d_{23} = d_{34} = 1\,, \qquad d_{14} = \sqrt{z} \quad \text{(C.6a)}$$

$$d_{24} = (-1)^{w_1 w_2 + w_1 w_3 + w_2 + w_4 + w_3 w_4}\sqrt{1-z}\,, \quad \text{(C.6b)}$$

$e_i$ are the canonical basis four-vectors and $i,j \in \{1,2,3,4\}$. (C.5) is a direct consequence of (C.1) and the definition (2.14).

## C.4 The 'cross ratio' $c$ evaluated at $P_{w+\varepsilon}(x; z) = 0$

The 'cross ratio' $c$ defined as

$$c = \begin{cases} \frac{X_{14}X_{23}}{X_{12}X_{34}} , & \sum_i w_i \in 2\mathbb{Z} , \\ \frac{X_{134}X_2}{X_{123}X_4} , & \sum_i w_i \in 2\mathbb{Z}+1 , \end{cases} \tag{C.7}$$

entering respectively eqs. (2.13a) and (2.13b) approaches 0, 1, $z$ or $\infty$ near the singularities $P_{w+\varepsilon}(x; z) = 0$. It follows from the identity (C.1) that

$$c\Big|_{P_{w+\varepsilon}(x;z)=0} = 0 \qquad \text{for } |\varepsilon| = (0,1,0,0), (1,0,0,1), (0,1,1,0), (1,0,1,1) , \tag{C.8a}$$

$$c\Big|_{P_{w+\varepsilon}(x;z)=0} = 1 \qquad \text{for } |\varepsilon| = (1,0,0,0), (1,0,1,0), (0,1,0,1), (0,1,1,1) , \tag{C.8b}$$

$$c\Big|_{P_{w+\varepsilon}(x;z)=0} = z \qquad \text{for } |\varepsilon| = (0,0,0,0), (0,0,1,0), (1,1,0,1), (1,1,1,1) , \tag{C.8c}$$

$$c\Big|_{P_{w+\varepsilon}(x;z)=0} = \infty \qquad \text{for } |\varepsilon| = (0,0,0,1), (1,1,0,0), (0,0,1,1), (1,1,1,0) . \tag{C.8d}$$

## D The worldsheet correlator near $P_{w+(1,0,0,1)}(x; z) = 0$

In this appendix we explain by an example how to compute the worldsheet four-point function

$$\left\langle V_{j_1,h_1}^{w_1}(0;0) V_{j_2,h_2}^{w_2}(1;1) V_{j_3,h_3}^{w_3}(\infty;\infty) V_{j_4,h_4}^{w_4}(x;z) \right\rangle \tag{D.1}$$

near the singularity $P_{w+\varepsilon}(x; z) = 0$. We derive eq. (3.16) explicitly for $\varepsilon = (1,0,0,1)$.

Let us start by discussing how the unflowed correlator entering eq. (2.13a) behaves in the vicinity of $P_{w+(1,0,0,1)}(x; z) = 0$. Near this locus, the 'cross ratio'

$$c = \frac{X_{14}X_{23}}{X_{12}X_{34}} \tag{D.2}$$

tends to 0 — see eq. (C.8a) — and from eq. (B.15b) we find that the unflowed correlator behaves as[24]

$$\langle V_{j_1}^0(0;0) V_{j_2}^0(1;1) V_{j_3}^0(\infty;\infty) V_{j_4}^0(x;z) \rangle\Big|_{P_{w+(1,0,0,1)}(x;z)=0} = |c|^{2(-j_1+j_2+j_3-j_4)} f_{(1,0,0,1)}(z) . \tag{D.3}$$

The worldsheet correlator then reads

$$\left\langle V_{j_1,h_1}^{w_1}(0;0) V_{j_2,h_2}^{w_2}(1;1) V_{j_3,h_3}^{w_3}(\infty;\infty) V_{j_4,h_4}^{w_4}(x;z) \right\rangle\Big|_{P_{w+(1,0,0,1)}(x;z)=0}$$

$$= f_{(1,0,0,1)}(z) \int \prod_{i=1}^{4} \frac{d^2 y_i}{\pi} y_i^{\frac{kw_i}{2}+j_i-h_i-1} \left( X_\emptyset^{j_1+j_2+j_3+j_4-k} X_{12}^{-2j_2} X_{13}^{-j_1+j_2-j_3+j_4} \right.$$

$$\left. \times X_{34}^{j_1-j_2-j_3-j_4} X_{14}^{-j_1+j_2+j_3-j_4} \right)\Big|_{P_{w+(1,0,0,1)}(x;z)=0} \times \text{c.c.} , \tag{D.4}$$

---

[24] As also discussed in the main text, one could in principle also consider the other solution of the KZ equation, i.e. use eq. (B.15a) in place of eq. (B.15b). We will see in a moment that this would not yield a singularity.

where c.c. stands for complex conjugate. Let us now perform the change of variables

$$y_1 \to P_{w+(1,0,0,1)}\, y_1 \ , \quad \text{and} \quad y_4 \to P_{w+(1,0,0,1)}\, y_4 \tag{D.5}$$

and compute the various 'generalised differences' $X_I$ entering (D.4). Making use of the identity eq. (C.1), we find

$$X_\emptyset\big|_{P_{w+\varepsilon}=0} = P_w\big|_{P_{w+\varepsilon}=0} = P_{w+\varepsilon+(-1,0,0,-1)}\big|_{P_{w+\varepsilon}=0} \tag{D.6}$$

$$= \tilde{w}_1^{-\frac{1}{2}}\, \tilde{w}_4^{-\frac{1}{2}}\, \Pi^{\frac{1}{2}}\, a_1^{-\frac{1}{2}}\, a_4^{-\frac{1}{2}}\, \prod_{i=1}^{4}(\tilde{w}_i\, a_i)^{\frac{\tilde{w}_i+1}{4}} \tag{D.7}$$

and

$$X_{12}\left(x, z, P_{w+\varepsilon}\, y_1, y_2, y_3, P_{w+\varepsilon}\, y_4\right)\big|_{P_{w+\varepsilon}=0}$$

$$= \left(P_{w+\varepsilon+(0,1,0,-1)} + P_{w+\varepsilon+(0,-1,0,-1)}\, y_2\right)\big|_{P_{w+\varepsilon}=0}$$

$$= \tilde{w}_2^{-\frac{1}{2}}\, \tilde{w}_4^{-\frac{1}{2}}\, \Pi^{\frac{1}{2}}\, a_2^{\frac{1}{2}}\, a_4^{-\frac{1}{2}}\, (1 - a_2^{-1}y_2)\, \prod_{i=1}^{4}(\tilde{w}_i\, a_i)^{\frac{\tilde{w}_i+1}{4}} \ , \tag{D.8}$$

where $\tilde{w}_i = w_i + \varepsilon_i$. Similarly,

$$X_{13}\left(x, z, P_{w+\varepsilon}\, y_1, y_2, y_3, P_{w+\varepsilon}\, y_4\right)\big|_{P_{w+\varepsilon}=0}$$

$$= \tilde{w}_3^{-\frac{1}{2}}\, \tilde{w}_4^{-\frac{1}{2}}\, \Pi^{\frac{1}{2}}\, a_3^{\frac{1}{2}}\, a_4^{-\frac{1}{2}}\, (1 - a_3^{-1}y_3)\, \prod_{i=1}^{4}(\tilde{w}_i\, a_i)^{\frac{\tilde{w}_i+1}{4}} \ , \tag{D.9a}$$

$$X_{34}\left(x, z, P_{w+\varepsilon}\, y_1, y_2, y_3, P_{w+\varepsilon}\, y_4\right)\big|_{P_{w+\varepsilon}=0}$$

$$= \tilde{w}_1^{-\frac{1}{2}}\, \tilde{w}_3^{-\frac{1}{2}}\, \Pi^{\frac{1}{2}}\, a_1^{-\frac{1}{2}}\, a_3^{\frac{1}{2}}\, (1 - a_3^{-1}y_3)\, (1-z)^{\frac{1}{2}}\, \prod_{i=1}^{4}(\tilde{w}_i\, a_i)^{\frac{\tilde{w}_i+1}{4}} \ , \tag{D.9b}$$

$$X_{14}\left(P_{w+\varepsilon}\, y_1, y_2, y_3, P_{w+\varepsilon}\, y_4\right)\big|_{P_{\tilde{w}}=0}$$

$$= z^{\frac{1}{2}}\, P_{w+\varepsilon}\left(1 + P_{w+\varepsilon+(-2,0,0,0)}\, y_1 + P_{w+\varepsilon+(0,0,0,-2)}\, y_4\right)\big|_{P_{w+\varepsilon}=0} \tag{D.9c}$$

Putting everything together, making the further change of variables

$$y_1 \to \left(P_{w+\varepsilon+(-2,0,0,0)}\right)^{-1} y_1 \ , \quad \text{and} \quad y_4 \to \left(P_{\tilde{w}+(0,0,0,-2)}\right)^{-1} y_4 \tag{D.10}$$

and using once more eq. (C.1) we find

$$\left\langle V^{w_1}_{j_1,h_1}(0;0) V^{w_2}_{j_2,h_2}(1;1) V^{w_3}_{j_3,h_3}(\infty;\infty) V^{w_4}_{j_4,h_4}(x;z)\right\rangle\bigg|_{P_{w+(1,0,0,1)}(x;z)=0}$$

$$= f_{(1,0,0,1)}(z)\bigg[(z - z_\gamma)^{\frac{k(\tilde{w}_1+\tilde{w}_4-2)}{2}-h_1-h_4+j_2+j_3}(1-z)^{-\frac{k(\tilde{w}_1-1)}{2}+h_1-j_2-j_3}$$

$$\times\ \Pi^{-\frac{k}{2}}\, \tilde{w}_1^{\frac{k\tilde{w}_1}{2}-h_1}\, \tilde{w}_2^{j_2}\, \tilde{w}_3^{j_3}\, \tilde{w}_4^{\frac{k\tilde{w}_4}{2}-h_4}\, \prod_{i=1}^{4}a_i^{\frac{k(\tilde{w}_i-1)}{4}-h_i}\, \tilde{w}_i^{-\frac{k(\tilde{w}_i+1)}{4}}\ \times\ \text{c.c.}\bigg]$$

$$\times \int \prod_{i=1}^{4} \frac{\mathrm{d}^2 y_i}{\pi} \, y_i^{\frac{kw_i}{2}+j_i-h_1-1} \, (1-y_2)^{-2j_2} \, (1-y_3)^{-2j_3} \, (1-y_1-y_4)^{-j_1+j_2+j_3-j_4} \, \times \text{ c.c. } .$$

$$\tag{D.11}$$

One can check through a similar computation that using the behaviour

$$\langle V_{j_1}^0(0;0) V_{j_2}^0(1;1) V_{j_3}^0(\infty;\infty) V_{j_4}^0(x;z)\rangle\Big|_{P_{\boldsymbol{w}+(1,0,0,1)}(x;z)=0} = f_{(0,1,1,0)}(z) \, , \tag{D.12}$$

of the unflowed correlator leads to a regular result for the full correlation function. Hence this behaviour does not contribute to the critical behaviour of the full correlator.

# E   Correlators of the symmetric product orbifold

In Section 5, we compared our spacetime correlators with correlators of symmetric product orbifold twist fields. In this appendix we briefly review the definition of these correlators. The relevant single cycle correlators are

$$\langle \sigma_{w_1}(0) \sigma_{w_2}(1) \sigma_{w_3}(x) \sigma_{w_4}(\infty)\rangle \tag{E.1}$$

in the symmetric product orbifold $\mathrm{Sym}^N(X)$, where $X$ is a CFT with central charge $c$. We will take $\sigma_w$ to be generic primary fields in the respective twisted sector, whose conformal weights can be parametrized by

$$h = \frac{c(w^2-1)}{24w} + \frac{\tilde{h}}{w} \, . \tag{E.2}$$

Here, $\tilde{h}$ can be thought of as the conformal weight of the field in the covering space. For unit-normalised two-point functions, the connected large $N$ contribution to the four-point function takes the form [56, 75, 76][25]

$$\langle \sigma_{w_1}(0) \sigma_{w_2}(1) \sigma_{w_3}(x) \sigma_{w_4}(\infty)\rangle_{\mathrm{c}} = N^{-1} \sum_{\gamma} \prod_{i=1}^{4} w_i^{-\frac{c(w_i+1)}{12}} a_i^{\frac{c}{24}(w_i-1)-h_i} \bar{a}_i^{\frac{c}{24}(w_i-1)-h_i}$$

$$\times |\Pi|^{-\frac{c}{6}} \langle \tilde{\sigma}(0) \tilde{\sigma}(1) \tilde{\sigma}(z) \tilde{\sigma}(\infty)\rangle \, . \tag{E.3}$$

Here, $\tilde{\sigma}$ are the respective fields lifted up to the covering space with conformal weights $\tilde{h}_i$. (Despite the notation, we do not assume those fields to be all equal.) Here we used the convention (3.14) for the product of the residues of the covering map. The prefactor is inserted so as to keep the two point function canonically normalised. It can be derived in a variety of ways, either by using the covering space method of Lunin & Mathur [75, 76], or by imposing factorisation of the result [56].

---

[25]This correlator is connected in the same sense as in footnote 17.

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
