# Peer review of "String correlators on AdS3: analytic structure and dual CFT"

_SciPost Physics_

## Round 1 · Referee Report · Sylvain Ribault (Referee 1) · 2022-5-10

Report

This article is the third in a series on string correlators in AdS3, which studies the AdS3/CFT2 relation in unprecedented detail. The previous two articles have established explicit formulas for three- and four-point functions from the worldsheet CFT: starting from these formulas, the task is now to study their singularities, and compare them with singularities of the corresponding spacetime CFT correlators.

The formulas for the string correlators have rich geometrical structures, and their detailed study requires a wealth of technical results, many of which are given in the appendices. The main text focuses on the general arguments while minimizing technical details. The resulting article is quite clear and concise. The price to pay is that checking any one of the many nontrivial results can require quite a lot of unpacking. It would not be realistic for a reader or reviewer to check all the results, and neither would it be realistic for the authors to write all their derivations in detail. The question is rather whether the motivated reader has enough indications for understanding the gist of the calculations, and for being able to reconstruct any one argument should the need arise. The submitted text is already quite good in this respect, although I think that some improvements are possible and desirable.

Before making specific suggestions, let me discuss one of the article's main results: the formula (4.1) for the poles of the spacetime four-point function. This very simple result has a complicated derivation, which involves considering quite a few different cases. To find a simpler derivation (or at least a test), one might remember that the vertex operators Vwj,h,ˉh(x;z) should in principle be linear combinations of the fields of the WZW model. These fields are in one-to-one correspondence with states in the spectrum, and they do not depend on x: this x may be interpreted as determining the direction in which we perform spectral flow, but we might choose it to be the same for all fields. Then we might use [17], which relates our N-point function with a correlation function of Liouville theory of the type
Ni=1Vαi(zi)N2+rj=1V12b(yj)
where αi=b(1ji)+12b, and the spectral flow violation r=wi obeys |r|N2.
From standard Coulomb gas arguments, this Liouville correlation function is singular if iαiN2+r2b+mb+nb1=b+b1 for m,nZ0. In terms of the AdS3 spins and level, this condition is
Ni=1ji=(n+a)(k2)+m+N1
where a=r2 is half-integer such that |a|N22. For N=4 this agrees with Eq. (4.1), with the only difference that a is now half the spectral flow violation instead of the more elaborate (4.2). The relaxation of spectral flow violation due to what is in principle just a change of bases is a mysterious feature of the results of [23, 24], maybe due to the fact that we are not dealing with a bona fide change of bases -- in other words, not just with algebra, but with analysis. In any case, this relaxation takes a rather simple form. It might be possible to guess the generalization of (4.1) to N-point functions on arbitrary Riemann surfaces, with the help of [19].
However, the fact that (4.1) now seems to emerge from unintegrated worldsheet correlators is at odds with the idea that z-integration plays a role.

Requested changes

1. The abstract mentions a series of papers in several installments, without specifying what the next installment will be about or whether it is even planned. The conclusion lists a number of open questions, without saying which are the most difficult or important ones. I would be interested to know more about the authors' opinions and plans, in order to properly understand the place of this article in their work and more generally in the subject.

2. The abundance of footnotes, and references to appendices, makes the reading less smooth than it could be. There are even two references to previous footnotes: a sure sign that material therein was not so accessory. I am not sure anything can be done about the appendices without overburdening the main text. But most footnotes could be integrated into the text.

3. In the partial list [1-25] in the introduction, many of the cited articles are outdated or irrelevant. When citing an article it would be helpful to indicate why the reader might be interested to consult it.

4. I am not sure I understand what is meant by the "times dependency" of the background, given that it is invariant under time translations.

5. "cross-ratio" seems more standard than "cross ratio" or "crossratio". At least, the authors should choose one convention.

6. The claim on page 3 that CFT correlators usually only have singularities in z when vertex operators collide is dubious. Having extra singularities seems to be the norm when the symmetry algebra is larger than just conformal symmetry, as was realized in the case of the algebra ^sl2 as early as [61]. (For an example with the W3 algebra see arXiv:1109.6764.) A more cautious formulation of this interesting discussion would be welcome.

7. Is Pw (2.9) integer?

8. Is the prefactor Sw (2.10) symmetric under permutations? Does is really not have a simpler expression?

9. A short discussion of the behaviour of the four-point function (2.13a) under permutations of the four fields might be interesting. In fact, it might be useful to make permutation symmetry manifest by restoring arbitrary positions and isospins z1,,z4 and x1,,x4.

10. Right after (2.14), it would be useful to write X explicitly, rather than (or in addition to) having it in (3.2).

11. I do not understand why we need chiral correlators in (3.7). The suggestion of taking one chiral term in a conformal block expansion apparently makes the determination of singularities dependent on the choice of an expansion (s-channel, t-channel or u-channel). But such choices are apparently not needed in subsequent calculations.

12. "lower codimension loci as" "lower codimension loci than"?

13. Section 3.1 is hard to understand in the absence of a precise definition of an exceptional intersection. The main statement of this section about the existence of exceptional intersections (3.8), is implied to be "trivial" (compared to the statement that there are no other exceptional intersections).
However, the present text does not convey this simplicity to the reader.
If I understand correctly, the idea is that XI is a polynomial in the yis, and that there is an exceptional intersection whenever one of its coefficients vanishes. And this point does not depend on the specific form of the Pws.
It would be nice to have a simple explanation, together with a more explicit treatment of one example.
By now the authors seem to consider Pw(x;z) and XI as elementary objects just like the sine function, but they should remember that readers may not be so familiar with these objects.

14. To make (3.8) precise, it is necessary to explicitly replace I with {i|ϵi0} or something of the sort.
15. I am puzzled that the left-hand side of (3.8) only depends on the sum w+ϵ, whereas the right-hand side seems to depend on w,ϵ independently. A comment would be welcome.

16. On page 13, the discussion of the singularity Pw(x;z)=0 could benefit from reminders that X=Pw(x;z)cz. (See (3.4).)

17. On page 14 I do not see the point of mentioning that (3.12) gives the leading term and that there are subleading terms: this seems obvious enough.

18. On page 15, it is nice to show explicitly that the second critical exponent vanishes. It would be even better to say what this means and whether this was expected. This seems to be a common (if not universal) occurrence for the singularities where operators do not coincide.

19. The formula (3.20) is a free-floating expression, not an equation. It forces the authors to later refer to "(3.16) and (3.20)". It would be better to write the full result. One option is to insert (3.20) into (3.16), and later to write that the factor (3.20) results from the integral over yi.

20. Typo in (3.26): ϵ=ijik+1 misses 0.

21. Footnote 12 is not very enlightening, in the absence of a more detailed explanation.

22. On page 18, the expression "external variables" is not clear.

23. Section 4.1 is a rare case of an explanation that is overly detailed and complicated. It is in principle a good idea to give the general mechanism that leads to (4.7), but why not write Fi=ρfi where ρ is the distance to the codimension m subvariety, and quickly conclude? It is not clear to me that the case of multiple zeros needs to be mentioned, since the Fis need not be distinct.

24. On page 14, we have one of many mentions of reflection symmetry: it would be useful to properly define reflection symmetry with the reflection equation for vertex operators, rather than drop the reflection coefficient in (3.27) without explanation. This would also help in the run-up to Eq. (5.4), which is a bit obscure as it is now.

25. It would be nice to explain in more detail why only two of the conditions (4.12) are independent, and whether this relies on properties of Pw(x;z).

26. In Section 4.2, if we were to impose the vanishing of an arbitrary subset of the XIs, would we always obtain a singularity that is related to the cases under study by a reflection? There are 216 such subsets, and it is not quite clear why it is enough to focus on the few cases under study.

27. In the codimension 4 case page 20, more details on "going to zero at twice the speed" would be welcome.

28. Page 21, the trivialization of X under condition (4.25) is not obvious: its geometrical origin could be explained.

29. In Section 4.4, two statements that would deserve more explanations are "the corresponding codimension 4 singularity does not exist" and "X=0 in this case".

30. In Section 5.2, it would be helpful to review the screening operators and their dimensions, possibly by reproducing relevant formulas from [25], for instance (2.34) and (2.47). The dimension of σ2 and the fact that eϕb has the momentum 12b (which shows up in Eq. (5.8)) should be made clearer. Is it true that the dual screening has momentum Q+12b with Q=b1b, and that the term in (5.8) has contributions from both screenings? Is the integer positive? If not, what is the -th order in perturbation theory?

31. In Appendix A.6, the limit to be computed "will turn out to be useful in the main text": give a more precise reference? More generally, references from the appendices to the main text could be useful.

32. In Equation (E.3), it would be preferable to have more explicit notations, rather than apologizing that the notations are misleading. Actually, the notations make it seem that the dependence on wi is simple, whereas (if I understand correctly) there is a non-trivial dependence hidden in the second line.

  • validity: -
  • significance: -
  • originality: -
  • clarity: -
  • formatting: -
  • grammar: -

Author:  Andrea Dei  on 2022-07-25  [id 2685]

(in reply to Report 1 by Sylvain Ribault on 2022-05-10)

We would like to thank the referee for his careful reading of the manuscript.

We think that the suggestion of the referee for the derivation of spacetime poles in the correlators is a good one and we did not think of it previously. It is true that what is suggested here is much simpler than what we did (assuming of course the highly non-trivial result of [19]). However, we do not think that it captures all the singularities and we think that one needs additional arguments to turn this into a full derivation. We see some problems:

The H3^+/ Liouville correspondence gives the worldsheet singularities in the m-basis. As the referee says, the x-basis correlators could conceivably be obtained from the m-basis correlators by considering also correlators of descendants in the m-basis and summing an infinite number of them. This sum may or may not converge. Conversely, the m-basis correlators can be obtained from the x-basis correlators by sending N1 xi’s to zero and one xi to . Generically this limit is singular and hence there is not always an m-basis correlator to each x-basis correlator. An m-basis correlator can be recovered precisely if the old `spectral flow violation rules’ are satisfied, i.e. when |iwi|N2. We discussed this limit in detail in our previous paper 2107.01481, see Section 4.5. We also want to mention that this feature of more non-vanishing x-basis correlators was not a mysterious feature of our previous papers, but a phenomenon that is already known for a long time. The spectral flow violation rules were established in hep-th/0111180, see their Appendix D for a precise derivation. It is also a feature in their derivation that the x-basis correlators satisfy weaker bounds than the m-basis correlators that are precisely consistent with what we find. The main point to mention this again here is that the x-basis correlators are more general than the m-basis correlators. It can (and does) happen that the x-basis correlators have more poles than the m-basis correlators which are just a particular limit of the x-basis correlators. Thus the formula suggested by the referee captures only a subset of poles in the x-basis worldsheet correlators that survive the limit described above. Moreover it of course only applies whenever the m-basis correlator is non-zero, i.e. when the m-basis spectral flow violation bound is satisfied. This argument says nothing about the x-basis correlator with, say, w1=100, w2=110, w3=120, w4=140.

The second problem with the suggested derivation is that it misses the integral over z. This integral clearly leads to more poles, which can already be seen at the simplest example. For the correlation function of four unflowed vertex operators, the worldsheet correlator behaves as (xz)kj1j2j3j4×c.c. as the two cross-ratios approach. This behaviour clearly leads to additional poles when we integrate over z. Thus we maintain that it is very important to the problem that we integrate over z and this changes the set of poles.

One should be careful when predicting the poles from the H3+/Liouville correspondence and also take into account that the integrals over y_i in e.g. equation (3.29) of [19] may introduce additional zeroes. We would of course be more than happy to find a simpler derivation of our result and also think that this should probably be possible. Unfortunately we were not able to find such a derivation. Perhaps it is possible to turn this suggested line of attack into a proof, but we do not immediately see how this would work.

In order to highlight the fact that the integration over z does matter for the set of poles in the string correlator, we also included the result for the poles of the worldsheet correlators right before Section 4.1. Hence the reader can now easily see which poles arise from integration over z and which don’t.

  1. We did not specify whether there will be a next installment in this series, because we do not know ourselves. We would of course be happy to follow this article up with another one, but don’t have specific plans at the moment and hence did not want to commit to a future paper. We also think that it is a matter of taste and opinion which open problems are the most interesting or difficult and hence left it on purpose to the reader to decide whether they would like to tackle one of them. We would of course be happy to communicate privately with anyone interested about various rough ideas we have.
  2. We moved various footnotes to the main text and removed in particular the cross references to footnotes.
  3. When compiling the list of references [1-25], we tried to mention the most important works on the subject. We agree that some of the papers are not directly relevant to the present paper, but at this level we want to keep the discussion general and put our work into a broader framework. The committed reader can easily find the references directly relevant to our paper since they are cited multiple times throughout the text when needed.
  4. We clarified what we mean by time-dependent in the introduction.
  5. We changed it to cross-ratio throughout.
  6. We included a reference to standard CFT axioms, where this is one of the basic assumptions. We also clarified that we mean local vertex operators.
  7. Yes, even though this is not obvious from the formula. This property never plays a role in the text.
  8. They are not invariant under permutations of 1, 2 and 3. However, the full three-point function is invariant under permutation of the labels 1, 2 and 3, which is discussed at the end of section 5.1 of 2105.12130. Unfortunately, we were not able to find a simpler formula for Sw. We should however mention that these formulas have some conventions built in that ultimately do not show up in the three-point functions, since some signs cancel out when combining left- and right-movers. It might hence be very well be possible that simpler expressions could be discovered.
  9. This property was discussed in our paper 2107.01481. See section 4.2. There are also many other further properties discussed there that are not directly relevant to the present paper and we did not mention them again in the text.
  10. This is a special case of eq. (2.14) and we explicitly mention this example now after the equation (2.14).
  11. We agree that this was confusing and significantly rewrote this explanation. The reasoning is basically the same as in Section 4.1: For the purposes of the singularity discussion one can assume that the worldsheet integrand is homomorphically factorized since this is true near singularities (but not in general). It is much better to just say this and not use the conformal block expansion since it is not needed. We changed the discussion so that it is much more similar to the one in Section 4.1.
  12. Changed according to the suggestion.
  13. We made the definition of exceptional intersection very precise and added more examples to clarify the notion. We agree that our use of the quantities XI is ubiquitous, but any statement we make about these quantities can be easily checked with the help of the ancillary Mathematica notebook that we provided with the arXiv submission of this paper.
  14. Changed according to the suggestion.
  15. w determines the set of hypersurfaces and should hence be considered fixed in the whole section. It is correct that different values of (\boldsymbol{w},\boldsymbol{\varepsilon}) with the same \boldsymbol{w}+\boldsymbol{\varepsilon} share these singularities. The left-hand-side of (3.8) makes this manifest, but it is more difficult to see on the right-hand-side. This overlap of singularities is related to the fact that representations are identified according to [D_j^+]^w=[D_{k/2-j}^-]^{w+1}.
  16. We thought that equation (3.9) already serves as such a reminder.
  17. We agree that this is perhaps obvious, but sometimes it is useful to state the obvious, since it may not be obvious to all readers. We hence kept the sentence there.
  18. We agree that it is a nice feature of the computation that there is always one critical exponent that vanishes. We do not understand whether this has a deeper meaning.
  19. We added the correct left-hand-side to eq. (3.20).
  20. We added \sim 0.
  21. We significantly expanded footnote 12 and incorporated it into the main text.
  22. We replaced the `external variables’ with a precise explanation of what we mean.
  23. We changed the explanation of the case with higher order zeros according to the suggestion. We feel that our explanation of change of variables is more complete than what the referee proposes (he only suggests how to replace one of the 10 integration variables and it is unclear whether the others have any influence). We hence feel that even though our explanation is slightly longer, it is more complete.
  24. We already included a reference to [25] for the discussion of reflection symmetry in the flowed sector, where this is explained in detail. The referee probably has in mind the reflection symmetry in the unflowed sector, where it involves an integral over x. In the flowed sector, the role of x is played by y and hence reflection symmetry in the y-space also involves such an integral, see eq. (2.30) of 2105.12130. However, after transforming back to h, the reflection symmetry just amounts to a multiplication by the reflection coefficient, see eq. (2.14) of 2105.12130.
  25. We agree that this is not clear from our explanations and as always with these identities, we only observe them experimentally for a large collection of w_i’s. They do depend on properties of the P_w’s. We provided a notebook with the definition of X_I in the ancillary file of 2107.01481 and the reader can easily convince themselves of the correctness of these identities. Some of the identities needed to show e.g. that not all of (4.14) are independent appeared earlier in the paper. For example, the identity mentioned in footnote 4 ensures that if 5 of the X_{ij}’s are zero, then the same is also true for the last one.
  26. First of all, depending on the parity of \sum_i w_i, either X_I with |I| even or odd is defined. There is thus only 8 different X_I’s. It is then true that there are combinatorially 2^8 choices of subsets that can go to zero. This number of choices is greatly reduced by our assumption that we only consider singularities that are symmetric in all the four spins and hence lead to a condition on \sum_i j_i. Within this subset one can also only choose some possibilities. For example, it would not be possible to set all X_{ij}’s to zero without setting X_{1234} also to zero, since vanishing of the former already implies vanishing of the latter, see eq. (3.4). The cases that we discuss are all those that are symmetric and respect the identities between the X_I’s. Cases that are not symmetric in the spins are related by reflection symmetry.
  27. Replaced "twice the speed" by "double zero" and included the reference to Section 4.1.
  28. We added more details on the triviality of X_\emptyset. It follows directly from the definition.
  29. We added more details why X_\emptyset=0. It follows again from the definition. We are not sure which singularity the referee is referring to since there is no codimension 4 singularity here. But as we stated, absence of singularities always directly follows from X_\emptyset=0, since the definition of those singularities does involve X_\emptyset.
  30. We explained that the linear dilaton momentum is -\frac{1}{2b} and the conformal weight is indeed equal to 1. We do not know whether there is a notion of a dual screening as in Liouville theory. Liouville theory is invariant under b \to 1/b, but this property is clearly not true in this case. One might of course hope that the vertex operator proposed by the referee could access other poles of the string correlator, but we don’t know whether this is true and we don’t see any physical reason why this should be true. Nonetheless we agree that it would be a natural thing to try. We also clarified that we can at the moment only compute residues of poles with m=0 in eq. (5.8). \ell is by definition a positive integer since it is the order of conformal perturbation theory, which we now stated explicitly.
  31. We added a reference to the main text.
  32. We added additional labels to distinguish the fields. We also added a reference to the definition of a_i and \Pi, which makes it obvious that the w_i-dependence is in fact very complicated.

---

## Round 1 · Referee Report · Anonymous (Referee 2) · 2022-7-23

Strengths

1- Impressive technical results on worldsheet correlators
2- Detailed match with the boundary CFT2
3- Very good exposition in spite of the nature of the material

Weaknesses

1- Probably not so easy to follow for an outsider, in spite of the author's efforts

Report

This paper is part of a series on the AdS3/CFT2 correspondence. The string background under consideration only has NSNS flux, and can be described using worldsheet techniques. Most of the paper is dedicated to computing certain worldsheet correlators; at the end the results are compared with those expected from the CFT2.

The worldsheet computations are an impressive tour de force, and lead to beautiful agreement with AdS/CFT. The use of the worldsheet (rather than supergravity) allows for a more thorough match than would be possible in most instances of the correspondence. While unfortunately these techniques cannot be easily exported to other, perhaps more famous dual pairs, they are likely to provide insights that could be useful more broadly.

The authors have done their best to explain their very technical results in such a way that they can be followed to a reasonable degree. Overall the paper is very impressive.

Requested changes

Two typos:
- p. 11, "lower codimension loci as"
- p. 18, "literately"

---

## Editorial Decision

resubmitted